# Dynamical modulation of hippocampal replay through firing rate adaptation

Zilong Ji [1,2,3,9], Tianhao Chu [1,2,9], Xingsi Dong [2,9], Changmin Yu[4], Daniel Bush [5], Neil Burgess [3,6] ✉ & Si Wu [1,2,7,8] ✉

During periods of immobility and sleep, the hippocampus generates diverse self-sustaining sequences of "replay" activity, which exhibit stationary, diffusive, and super-diffusive dynamical patterns. However, the neural mechanisms underlying this diversity in hippocampal sequential dynamics remain largely unknown. Here, we propose a unifying mechanism by showing that modulation of firing-rate adaptation strength within a continuous attractor model of place cells gives rise to these distinct forms of replay. Our model accounts for empirical data and yields several testable predictions. First, more diffusive replay sequences should positively correlate with longer theta sequences, both reflecting stronger adaptation. Second, increased neural activity combined with firing-rate adaptation should reduce the step size of decoded trajectories during replay. Third, the framework is consistent with previous work showing that replay diffusivity can vary within an animal across behavioural states that may influence adaptation (such as wake and sleep). Together, these results suggest that the diverse replay dynamics observed in the hippocampus can be understood through a simple yet powerful neural mechanism, providing insight into the computational role of replay in hippocampal-dependent cognition and its relationship to other electrophysiological phenomena.

The hippocampus plays a pivotal role in various cognitive processes, including navigational learning[1,2], goal-directed decision-making[3–5], and episodic memory[6–8]. These cognitive processes require the temporal coding of relationships between events and/or locations, with hippocampal sequential activity hypothesised to support these computations[9–14]. Empirical studies have identified two distinct yet interrelated types of hippocampal sequences. First, during active exploration, place cells fire in sequences within individual theta cycles detected from local field potential (LFP), termed theta sequences (Fig. 1a, b)[10,11,15]. Nested within longer behavioural timescales, theta sequences compress spiking intervals between successive place cells to within dozens of milliseconds, thereby facilitating

Hebbian synaptic plasticity[16–18], which is essential for the initial formation of memory traces[19–21]. Furthermore, their forward-directed nature suggests a potential role in spatial planning and decision-making[3,4,22,23], as well as in spatial sampling[5,24–26]. Second, during immobility and sleep, behavioural sequences encoded in the hippocampus are reactivated as "replay" events (Fig. 1c, d). Similar to theta sequences, replay sequences occur in temporally compressed form, lasting tens to hundreds of milliseconds, and are associated with hippocampal sharp-wave ripples (SWRs)[27]. Replay sequences manifest in diverse forms, including forward or reverse replay of recent experiences[12,28], remote replay of past experiences[13], and preplay of anticipated trajectories[29]. This diversity has been interpreted as

[1]School of Psychological and Cognitive Sciences, Peking University, Beijing, China. [2]Peking-Tsinghua Center for Life Sciences, Academy for Advanced Interdisciplinary Studies, Peking University, Beijing, China. [3]Institute of Cognitive Neuroscience, University College London, London, UK. [4]Computational and Biological Learning Lab, Department of Engineering, University of Cambridge, Cambridge, UK. [5]Department of Neuroscience, Physiology and Pharmacology, University of College London, London, UK. [6]UCL Queen Square Institute of Neurology, University College London, London, UK. [7]PKU-IDG/McGovern Institute for Brain Research, Peking University, Beijing, China. [8]Center of Quantitative Biology, Peking University, Beijing, China. [9]These authors contributed equally: Zilong Ji, Tianhao Chu, Xingsi Dong. ✉e-mail: n.burgess@ucl.ac.uk; siwu@pku.edu.cn

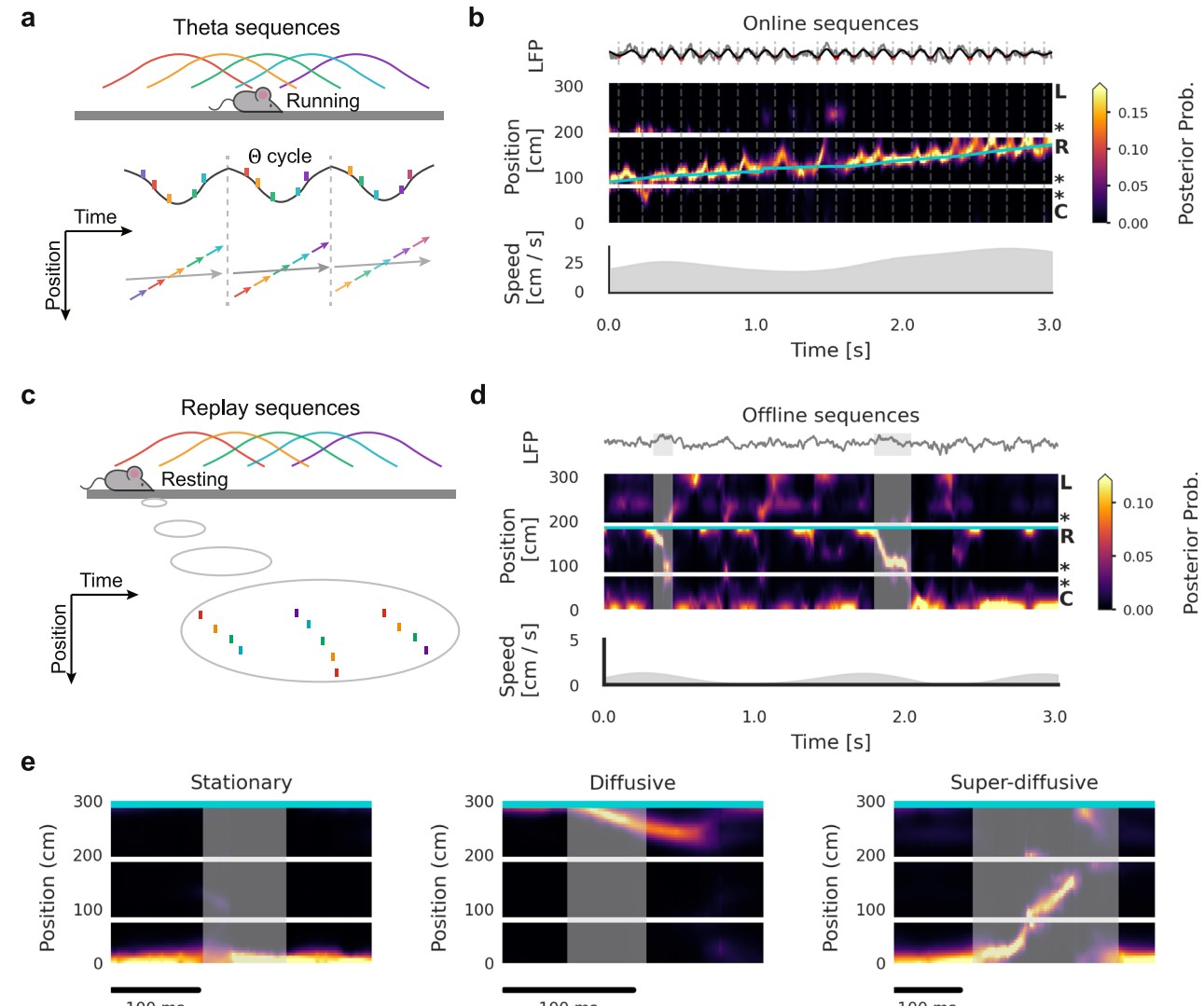

**Fig. 1 | Sequential dynamics in the hippocampus. a** Schematic of theta phase precession at the single-neuron level and theta sequences at the population level. **b** Example theta sequences in empirical data when an animal performed a W-track spatial alternation task. Top panel: theta-band (5–11 Hz) filtered LFP. Middle panel: decoded position; linearised from the central arm (C) to the right arm (R) and then to the left arm (L). The posterior probability is plotted as colour values with the blue line representing the actual location of the animal. Bottom panel: running speed of the animal. **c** Schematic of replay sequences. **d** Example replay sequences in empirical data when the animal stopped at the end of the right arm (marked by the blue line). Grey shading indicates periods of detected SWRs. **e** Diverse replay dynamics. From left to right: stationary replay (decoded positions stay at a single location), diffusive replay (decoded positions slowly propagate), and superdiffusive replay (decoded positions make abrupt transitions to distant locations). Grey shading indicates periods of detected SWRs.

reflecting distinct underlying mechanisms and cognitive roles, such as memory consolidation and goal-directed planning[30,31].

While theta sequences regularly sweep through the animal's current location during active movement, many replay sequences lack steady propagation and instead display diverse dynamical patterns (Fig. 1e). For example, some consistently represent single remote locations, potentially corresponding to reward wells or decision points[32]. In spatial memory tasks, some replay sequences exhibit "jumping" transitions[33], a phenomenon termed superdiffusive dynamics. Conversely, during post-task sleep following random foraging, replay trajectories display Brownian diffusion[34], which do not directly recapitulate behavioural paths. Beyond task context, developmental studies show that replay dynamics evolve from representing isolated locations in pre-weaning stages to trajectory-like sequences after weaning[20,35]. Together, these findings indicate that multiple factors shape the detailed dynamics of replay, highlighting unresolved questions about the mechanisms governing their statistical structure. What conditions are necessary and sufficient for the generation of

replay sequences? Which factors determine the emergence of specific dynamical regimes, and how are these mechanisms modulated across behavioural and physiological states? Finally, how are different hippocampal sequences—particularly theta sequences during active exploration and replay during rest—functionally interconnected?

Recent studies have proposed that the generation of sequential activity may arise from single-cell firing-rate adaptation (FRA), which penalises repeated activation of the same firing patterns (Fig. S1) and therefore causes different cells with lateral connections to fire sequentially[36–38]. FRA is a widespread neurobiological phenomenon, exhibited by almost any type of neuron that generates action potentials, such as rodent motoneurons[39], hippocampal CA1 pyramidal cells[40], pyramidal cells of the piriform cortex[41], and most pyramidal neurons in rodent neocortex[42]. Biologically, FRA is a complex and multifaceted phenomenon that arises from multiple mechanisms, including intrinsic neuronal properties such as afterhyperpolarization (AHP)[43]; synaptic processes such as short-term depression[44]; and neuromodulatory influences such as acetylcholine (Ach)[45].

In this study, we identify a neural mechanism capable of generating replay dynamics that follow a diverse range of movement statistics. To this end, we develop a theoretical framework that conceptualises the hippocampal place-cell assembly as a continuous attractor network (CAN)—a model extensively used to describe the firing properties of spatially tuned cells in the entorhinal-hippocampal system[46], including head-direction cells[47], place cells[48] and grid cells[25,49]. In this framework, spatial representations in the hippocampus are encoded as an activity "bump" within the CAN, and replay sequences arise as spontaneous movements of this bump. We show, both theoretically and empirically, that FRA provides an intrinsic mechanism that destabilises the activity bump, with variation in adaptation strength giving rise to distinct replay-like dynamics—ranging from stationary and Brownian-diffusive to superdiffusive sequences, consistent with empirical observations. Beyond offering a unified theoretical explanation for the diverse movement statistics observed during replay, our model also captures several key features reported in experimental data.

First, prior studies have demonstrated the concurrent development of theta and replay sequences[19,20,35]. Building on this, we examined how replay relates to theta sequences, as theta sequences can also be reproduced within our model[38]. The model predicts that more diffusive replay should correlate with longer theta sequences, both reflecting stronger adaptation. We confirm this prediction in empirical data. Second, previous work has identified systematic differences in replay dynamics between waking[33] and sleep[34] states, with awake replay exhibiting greater diffusivity. Our model attributes these state-dependent differences to variations in FRA strength, and further predicts that the same pattern should be evident within individual animals. This prediction was supported by our analyses and consistent with recent findings[50]. Third, long-jump transitions during replay have been shown to correlate negatively with population activity levels[33]. Our model provides a mechanistic explanation for this phenomenon, suggesting it arises from an interaction between FRA and external oscillatory inputs.

## Results

### A continuous attractor network with FRA for the hippocampal place-cell system

We modelled the hippocampal place-cell network as a continuous attractor network (CAN)[46,48] ("Methods"). Neurons were arranged according to the locations of their firing fields along a linear track (Fig. 2a; this framework is expected to generalise to two-dimensional environments), and the network dynamics were defined as

$$\tau_u \frac{\partial U(x,t)}{\partial t} = -U(x,t) + \rho \int_{x'} J(x,x') r(x',t) \mathrm{d}x' - V(x,t) \\ + I_{ext}(x,t) + \sigma_U \xi_U(x,t), \tag{1}$$

where $U(x,t)$ denotes the synaptic input to the cell at location $x$, and $r(x,t)$ the corresponding firing rate. Each place cell forms recurrent excitatory connections with all other cells through synapses $J(x,x')$ (scaled by $\rho$), where the connection strength decays with the distance between the place fields of the two cells. This translation-invariant connectivity can arise naturally from a Gaussian random walk over environmental locations[51,52]. In addition, the network is subject to global feedback inhibition (Eq. (9) in "Methods"), which constrains total neural activity. This network configuration leads to the emergence of an activity"bump" as a stable attractor state, accompanied by localised firing fields of individual neurons (Fig. 2b; see "Methods" for theoretical derivation). The CAN model also receives a location-dependent sensory input $I_{ext}(x,t)$, which is active during animals' running but absent during rest. Finally, the network experiences internally generated noise $\sigma_U \xi_U(x,t)$.

Importantly, neurons in the CAN exhibit firing-rate adaptation (see Fig. 2a), a general feature of neural responses across the brain[53–56]. Although this phenomenon can arise from diverse biophysical mechanisms[43–45], it universally involves a slow negative feedback $V(x,t)$ acting on neuronal excitability, expressed as

$$\tau_v \frac{\partial V(x,t)}{\partial t} = -V(x,t) + mU(x,t) + \sigma_m U(x,t)\xi_V(x,t), \tag{2}$$

where $\tau_v$ is the FRA time constant ($\tau_v \gg \tau_u$), $m$ denotes the FRA strength, and $\xi_V(x,t)$ is noise scaled by $\sigma_m$. This negative feedback destabilises the activity-bump state: when the FRA strength exceeds a threshold, the bump begins to drift, and its intrinsic speed increases with the FRA strength (Fig. 2c and "Methods")[36,57]. Specifically: (1) weak adaptation (below threshold): active cells maintain constant firing rates, and the activity bump remains stationary (Fig. 2d); (2) moderate adaptation (just above threshold): active-cell firing rates gradually decline, recruiting nearby cells via recurrent excitation and global inhibition, resulting in slow bump movement (Fig. 2e); (3) strong adaptation (well above threshold): firing rates drop rapidly, triggering fast recruitment of neighbouring cells and rapid bump propagation (Fig. 2f).

### FRA strength determines replay diffusivity in the CAN model

Empirical data showed that hippocampal replay exhibits diverse movement patterns, including stationary, Brownian, superdiffusive ("jumping") movements[32–34,58,59]. We next investigated how the CAN model with FRA can account for this diversity by complementary theoretical analyses.

**Dynamical system analysis for generating diverse replay from a mechanistic viewpoint.** Our first analytical approach quantifies the step size of bump movements in the CAN model by solving the system dynamics and identifying the conditions that give rise to distinct replay regimes under different combinations of adaptation strength. Replay diffusivity is characterised by a power-law distribution of step sizes:

$$p(\|\Delta z\|) \sim \|\Delta z\|^{-1-\alpha}, \tag{3}$$

where $\Delta z$ denotes the displacement of the activity bump over a time interval $\Delta t$, and $\alpha$ is the power-law exponent. Mathematically, when $\alpha \geq 2$, the bump trajectory approximates Brownian diffusion, whereas for $0 < \alpha < 2$, it follows a superdiffusive process (values $\alpha \leq 0$ are not considered, as they imply divergence of both the mean and variance of the step size). Theoretical analysis ("Methods") revealed that the exponent $\alpha$ depends on two key parameters of the model-adaptation strength and noise level-with the functional form:

$$\alpha = 1 + \frac{2\mu}{\gamma^2}. \tag{4}$$

Here, $\mu$ represents the difference between the FRA strength and the threshold above which the bump starts to move (Fig. 2c), whereas $\gamma$ denotes the normalised noise level (proportional to the noise level $\sigma_m$ in FRA; see "Methods"). The relationship between the exponent $\alpha$ and these two parameters is summarised in a phase diagram (Fig. 3a, b), revealing three characteristic regimes: (1) super-diffusive dynamics occur when $\mu$ is small (FRA strength near the threshold) and $\gamma$ is large (high FRA noise); (2) Brownian-diffusive dynamics arise when $\mu$ is moderate (FRA strength moderately below the threshold) and $\gamma$ is moderate (intermediate FRA noise); (3) stationary dynamics emerge when $\mu$ is large (far below the threshold) and $\gamma$ is small (low FRA noise).

**Spectral analysis for generating diverse replay from a normative viewpoint.** The second analytical approach treats the CAN model as a

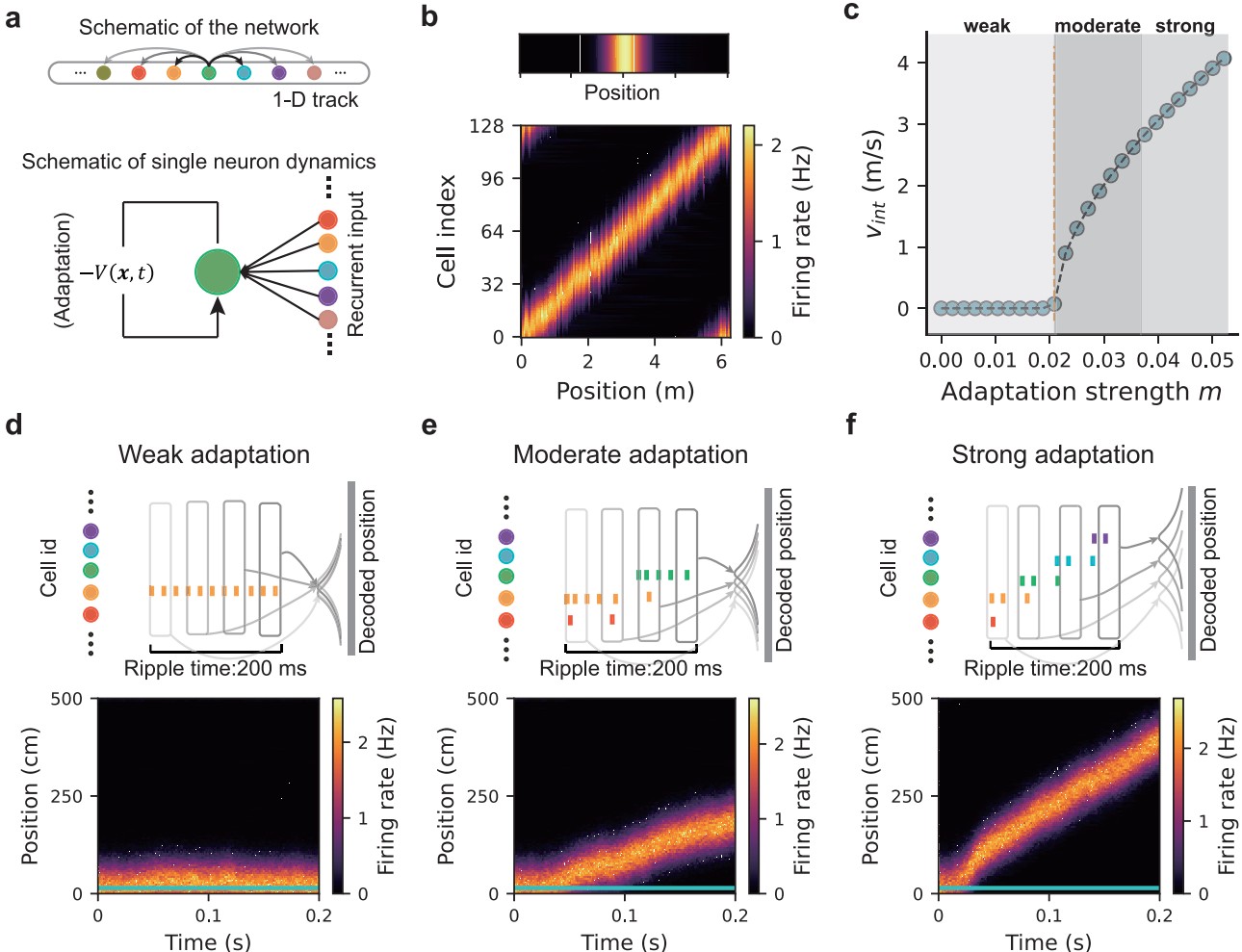

**Fig. 2 | Generating diverse replay sequences in a hippocampal place-cell network modelled by a continuous attractor network (CAN) with FRA. a** Top: schematic of the CAN used to model the place-cell network during navigation on a linear track. Neurons, shown as coloured dots, are arranged according to the spatial locations of their firing fields. Bottom: single-neuron dynamics, with recurrent input from other neurons and FRA illustrated as negative feedback to the neuron. **b** Top: activity bump within the CAN. Bottom: spatially localised firing fields for all neurons in the network. **c** Intrinsic speed of the activity bump as a function of adaptation strength. The orange line indicates the FRA strength threshold below which the bump remains stationary. Grey shading highlights example regimes of weak, moderate, and strong FRA strength. **d** Replay-like sequences under weak adaptation. Top: schematic showing stationary sequences; each vertical bar represents a spike event, coloured by place-cell identity. Each box denotes a time bin, with decoded position shown on the right. Bottom: population activity in the CAN. The simulated animal remains at the bottom of the linear track (horizontal blue line). **e** Replay-like sequences under moderate adaptation. **f** Replay-like sequences under strong adaptation.

sequence generator[50] by vectorising its dynamics into a master equation of the form

$$\tau \dot{U} = (O - C)U. \qquad (5)$$

Here, $U$ denotes the state distribution defined by population activity across place cells, $O$ is the transition matrix describing recurrent connections among states, and $C$ represents the perturbation matrix reflecting the negative feedback effect of adaptation. In this formulation, replay generation can be viewed as a sampling process from $U_t$, which evolves over time under the influence of the perturbed sequence generator (Fig. 3c). From Eq. (5), analysing the spectrum of the perturbed transition matrix provides normative insights into how adaptation-induced perturbations shape the diffusivity profiles of generated sequences (Fig. 3d–f; see "Methods").

Specifically, the perturbation matrix $C$ is a sub-diagonal matrix capturing the influence of the adaptation signal $V$ on the CAN activity bump $U$. The adaptation input has a similar Gaussian profile as $U$ but lags behind it by a displacement $s$ (Fig. 3d, inset). Increasing FRA

strength enlarges this displacement $s$ (Fig. 3d), thereby producing a greater perturbation offset in $C$ and shifting its elements further from the main diagonal. Mathematically, the perturbed transition matrix retains the same set of eigenvectors irrespective of the offset value[52,60], which are the Fourier modes (Fig. 3e; see "Methods"). Consequently, to assess how adaptation-induced perturbations affect the temporal evolution of the state distribution, it is sufficient to analyse how the spectrum (eigenvalues) of the transition matrix varies with the diagonal offset, which reflects different FRA strengths.

With a small offset ($s = 5$), corresponding to weak FRA, eigenvalues associated with large spatial scales (lower-frequency Fourier modes; Fig. 3e) are suppressed, whereas those of finer spatial scales are amplified (Fig. 3f, g). This rescaling enhances sampling within local regions, giving rise to Brownian-like diffusion. In contrast, with a large offset ($s = 50$), representing strong FRA, increases in eigenvalues of larger spatial scales always precedes decrease in eigenvalues of smaller spatial scales (the oscillatory pattern in Fig. 3f, h). Such rescaling amplifies transition probabilities between distant states—enhancing sampling of remote locations—and promotes super-diffusive

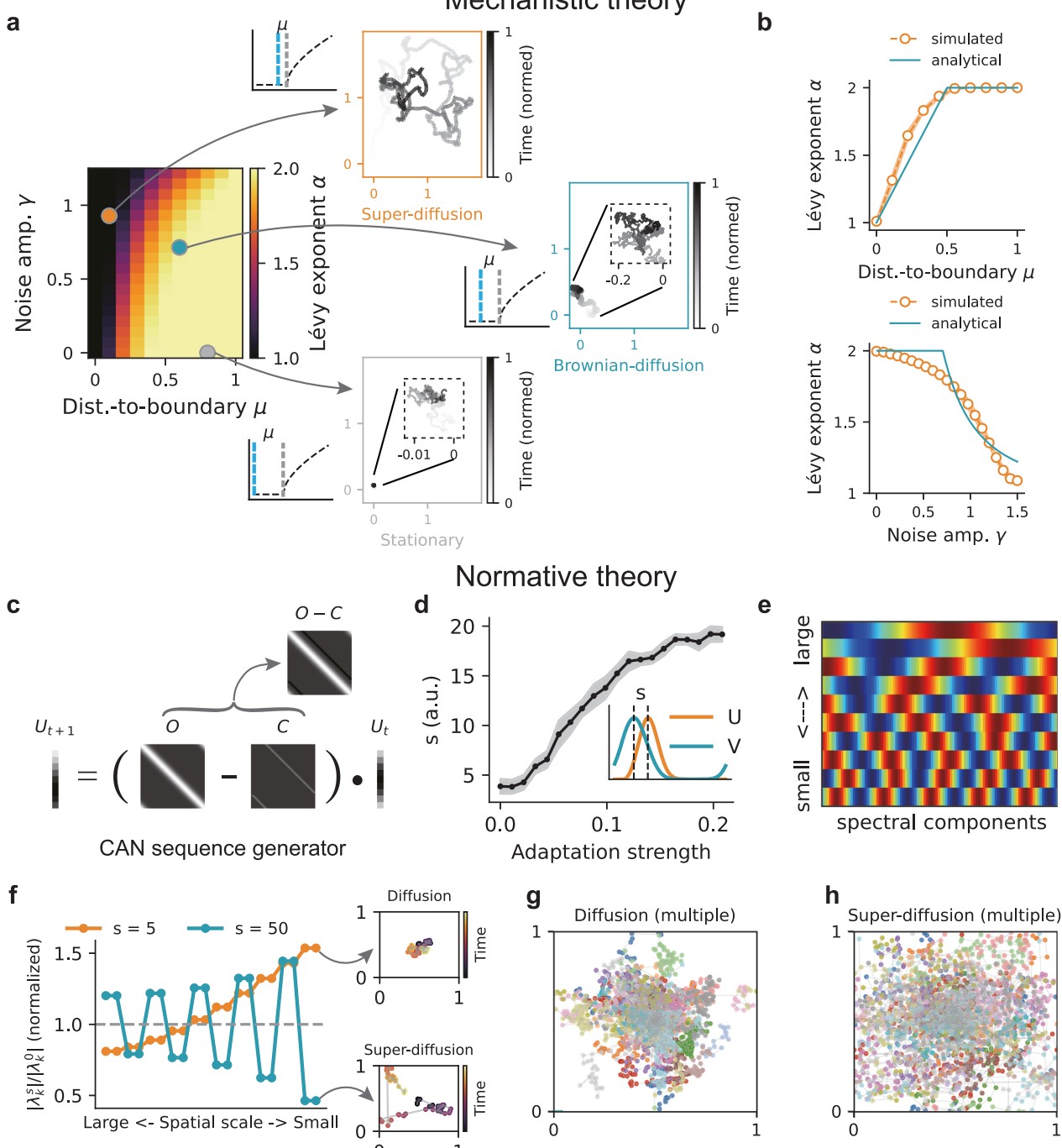

**Fig. 3 | Mechanistic and normative theoretical analyses of diverse replay generation in the CAN with FRA. a** Phase diagram of replay diffusivity (quantified by the power-law exponent $\alpha$) as a function of FRA strength and noise level. Larger values of $\alpha$ indicate more super-diffusive dynamics. Orange, blue, and grey dots denote parameter combinations producing super-diffusive, diffusive, and stationary dynamics, respectively. In the three insets, blue dashed lines indicate the FRA strength values, and grey dashed lines indicate the threshold shown in Fig. 2c. For clarity, all values $\alpha > 2$ were clipped to 2 as no heavy-tailed distribution exists in this regime, and the sum of step sizes converges to a Gaussian distribution under the Central Limit Theorem, corresponding to Brownian motion. **b** Top: replay diffusivity as a function of FRA strength. Bottom: replay diffusivity as a function of noise level. **c** Schematic of the sequence generator in the CAN. $U_t$ represents the state distribution, $O$ the transition matrix, and $C$ the perturbation matrix arising from the influence of FRA. **d** Perturbation offset $s$ as a function of FRA strength. Inset: example offset between the activity bump $U$ and the FRA bump $V$. **e** Example eigenvectors (Fourier modes) of the perturbed transition matrix, represented by wave vectors of increasing frequencies when eigenvalue magnitude decreases from large to small. **f** Left: rescaled eigenvalues (normalised by their unperturbed values) after perturbation at different offsets (orange: s = 5; blue: s = 50; top 20 eigenvalues shown). Right: sampled trajectories from the state sequences generated by the perturbed generator. **g** Multiple diffusion trajectories with s = 5. **h** Multiple super-diffusion trajectories with s = 50.

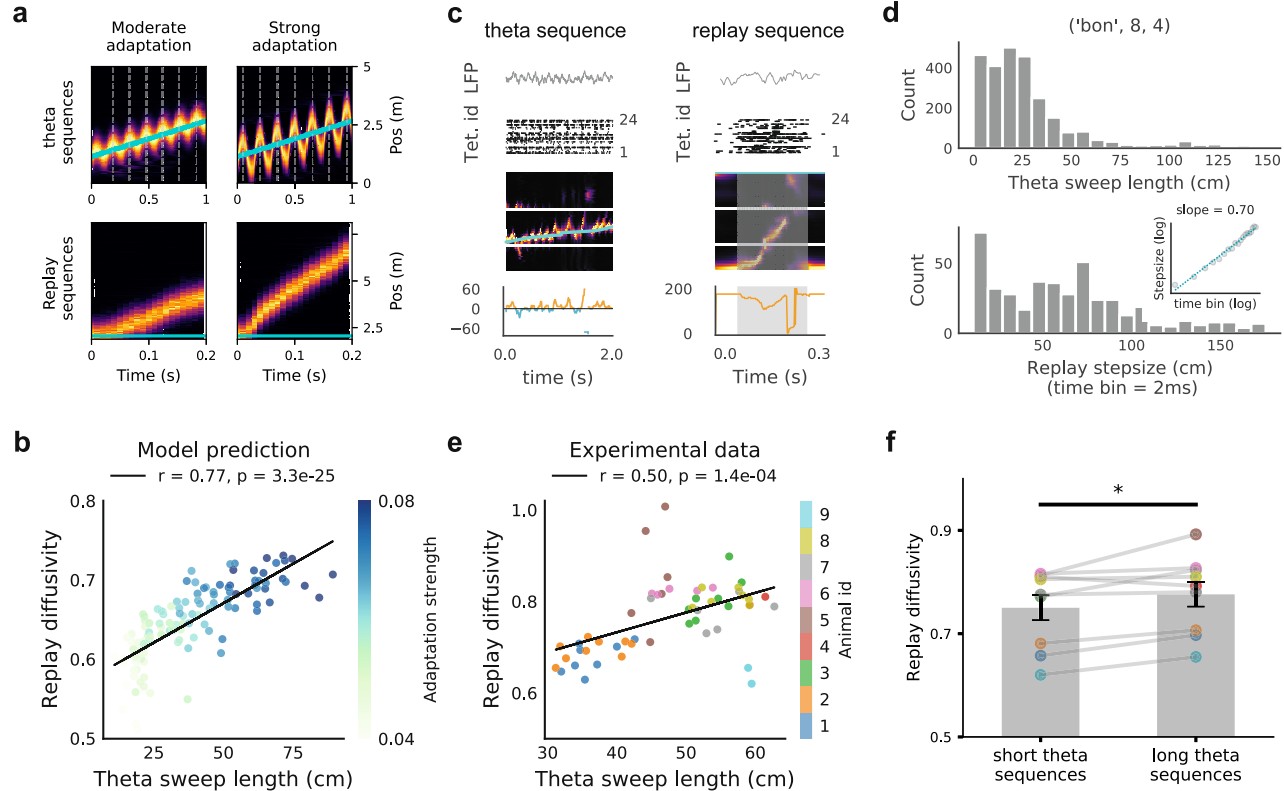

**Fig. 4 | More diffusive replay correlates with longer theta sequences. a** Theta (top) and replay sequences (bottom) generated in the CAN under moderate and strong FRA conditions, respectively. Blue lines indicate the agent's position over time, and heat maps represent network activity. **b** Scatter plot showing the positive correlation between replay diffusivity and theta sweep length. Each dot represents results from a simulated CAN network with varying FRA strength; darker blue denotes stronger adaptation. **c** Example theta and replay sequences from empirical data. From top to bottom: LFP trace from a CA1 tetrode, multiunit activity, posterior probability map, and offset distance over time. In theta sequences (left), blue and orange indicate look-back and look-ahead distances, respectively. In the replay (right), the offset between decoded and actual positions is shown. **d** Distributions of theta-sequence lengths and awake replay step sizes (2 ms time bin) from one recording session. Inset: relationship between replay step size and time bin on a log–log scale. **e** Correlation between replay diffusivity and theta-sequence length in empirical data. Each point represents the diffusion exponent and mean theta-sequence length from a single recording session; different colours indicate individual animals. **f** Within each animal, replay diffusivity is higher during sessions with long theta sequences compared with short ones (Wilcoxon signed-rank test with $P = 0.027$). Short and long theta sequences were defined as those below or above the mean theta-sequence length across sessions for each animal. Different colours represent individual animals.

dynamics. By rigorously deriving the eigenvalue rescaling spectrum under perturbation across offset values, we identified the precise boundary at which sampling behaviour transitions from Brownian to super-diffusive (Fig. S3; see "Methods" for more details).

## More diffusive replay correlates with longer theta sequences

During movement, the hippocampus generates theta sequences, whereas during rest it produces replay sequences using the same underlying network. We next investigated the relationship between these two forms of sequential activity. Theta sequences were reproduced in the CAN model by introducing location-dependent sensory input, implemented as an external bump input (Fig. 4a, and see "Methods" for more details)[25,38]. As adaptation strength increases, the activity bump sweeps further from the animal's location (Fig. 4a). Notably, higher adaptation strength also increases replay diffusivity in the absence of sensory input. From these results, we predicted a positive correlation between replay diffusivity and theta-sweep length (Fig. 4b; Pearson correlation $r = 0.77$, $P = 3.3 \times 10^{-25}$), indicating that both phenomena are jointly modulated by FRA strength.

We next tested this prediction using electrophysiology data from rodents performing a spatial memory task[61] (Fig. S2). A clusterless state-space decoding algorithm[59,62] was used to decode both theta sequences and replay sequences in the data (Fig. 4c, d and Fig. 4; see "Methods"). For each recording session, theta-sequence length was

calculated as the mean of the look-ahead and look-back distances across all LFP theta cycles, while replay diffusivity was estimated as the slope of replay step size versus time duration on a log-log scale[34] ("Methods"). Across sessions, replay diffusivity correlated significantly with theta-sequence length (Fig. 4e; Pearson correlation $r = 0.50$, $P = 1.4 \times 10^{-4}$), indicating that more diffusive replay is associated with longer theta sequences. Although this relationship was most evident in the aggregated data across animals, individual analyses revealed similar trends within each animal (Fig. S5). To further assess the within-animal effect, we divided recording sessions for each animal into two groups: those with short theta sequences and those with long theta sequences. Replay diffusivity was significantly higher in sessions with long theta sequences compared with short ones (Wilcoxon signed-rank test with $P = 0.027$; Fig. 4f). We finally compared observed correlations to those obtained after randomly shuffling theta-sequence lengths across recording days within each animal. Shuffling markedly reduced the correlation coefficients (Fig. S6; $P = 0.014$, Kolmogorov–Smirnov test), further supporting a genuine within-animal relationship between theta-sweep length and replay diffusivity.

Importantly, it is possible that place field size could confound the observed correlation between replay diffusivity and theta-sequence length. Specifically, smaller place fields may constrain the network activity, limiting its spatial spread and leading to shorter theta sequences and less diffusive replay. To rule out this possibility, we

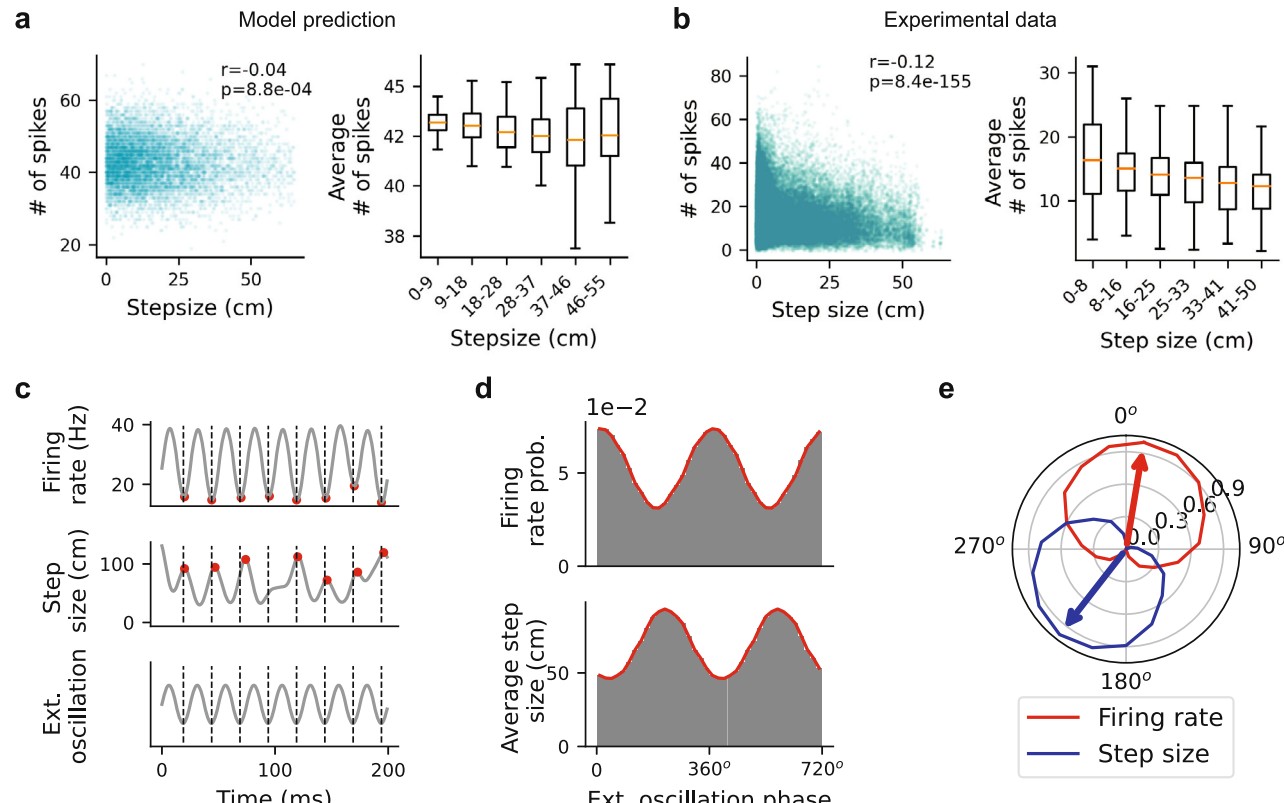

**Fig. 5 | Negative correlation between population activity and replay step size, and their relationship with slow-gamma oscillations. a** The CAN model predicts a negative correlation between neural activity and replay step size. Left: scatter plot showing summed Poisson spike counts (across all neurons within the time window used to calculate each step size) versus replay step size. Each dot represents a sampled step from simulated replay trajectories across different FRA strengths. Right: box plot of population firing rate versus binned step size. Orange lines indicate mean values, boxes represent the inter-quartile range, and whiskers denote non-outlier extremes. **b** Negative correlation between replay step size (time

bin = 20 ms) and neural activity in empirical data. **c** Anti-phase locking of population firing rate (top) and replay step size (middle) with an external oscillation signal (bottom) in the CAN. Red dots mark troughs and peaks of firing rate and step size, respectively; dashed lines mark oscillation troughs. **d** Population activity (top) and step size (bottom) plotted as a function of oscillation phase in the CAN (two repeated cycles shown). **e** Normalised contour plots with circular weighted means (arrows) for neural activity (red) and step size (blue) as functions of oscillation phase.

quantified place-field size using the population vector correlation method[63] (Fig. S7; see "Methods"). We found only a marginal correlation between place-field size and theta-sequence length (Pearson correlation $r = 0.26$, $P = 0.055$; Fig. S8), indicating that larger place fields tend to be associated with longer theta sequences[64]. In contrast, place-field size showed no significant correlation with replay diffusivity (Pearson correlation $P = 0.650$), suggesting that it is not a confounding factor. We also tested whether decoding accuracy might explain the observed relationship. Sessions with fewer spike events could yield noisier decoding of both theta and replay sequences, potentially inflating theta-sequence length and replay diffusivity. However, the correlation between theta-sequence length and the number of spikes used for decoding was weak (Pearson correlation $r = 0.31$, $P = 0.023$), and replay diffusivity was not significantly correlated with spike count (Pearson correlation $r = 0.14$, $P = 0.317$) (Fig. S9). These results indicate that the relationship between replay diffusivity and theta-sequence length cannot be attributed to variations in place-field size or decoding accuracy.

**Replay step size negatively correlates with population activity**

Individual awake replay sequences have been shown to exhibit interleaved local and long-jump movements in a spatial memory task[33]. However, the mechanism that modulates replay step-size structure remains unclear. To address this, we examined the factors influencing replay step size in the CAN. We observed a negative correlation between replay step size and population neural activity (Pearson's

$r = -0.04$, $P = 8.8 \times 10^{-4}$; Fig. 5a). This relationship arises from FRA: increases in neural excitability (driven by noise fluctuations in the CAN) transiently counteract the negative feedback imposed by FRA, resulting in smaller step sizes of bump movement. Conversely, reduced excitability fails to oppose—or can even amplify—the influence of FRA, leading to stronger adaptation-driven suppression of activity and larger step sizes, producing more diffusive trajectories.

To test this prediction empirically, we analysed the same empirical dataset[61]. Across all animals, replay events during awake immobility showed a significant negative correlation between replay step size and aggregate spike count (Pearson's $r = -0.12$, $P = 8.4 \times 10^{-155}$; Fig. 5b). Specifically, during periods of high neural activity, replays tend to remain localised, whereas during low-activity periods they are more likely to transition between locations. A potential concern is that reduced spike counts within decoding windows could degrade decoding accuracy and artificially inflate step size. However, this was not supported by the data: replay diffusivity remained stable across varying spike counts during ripple events (Fig. S9).

It has also been shown that both replay step size and neural activity are modulated by opposite phases of slow-gamma rhythms (25–50 Hz)[33]. Specifically, large replay step sizes occur near the trough of the slow-gamma cycle, when neural activity is low, while local movements align with the peak, when activity is high. Our model provides a mechanistic account of this phenomenon. To mimic slow-gamma oscillations during replay events, we introduced an external oscillatory input to the CAN (see "Methods"). In this configuration,

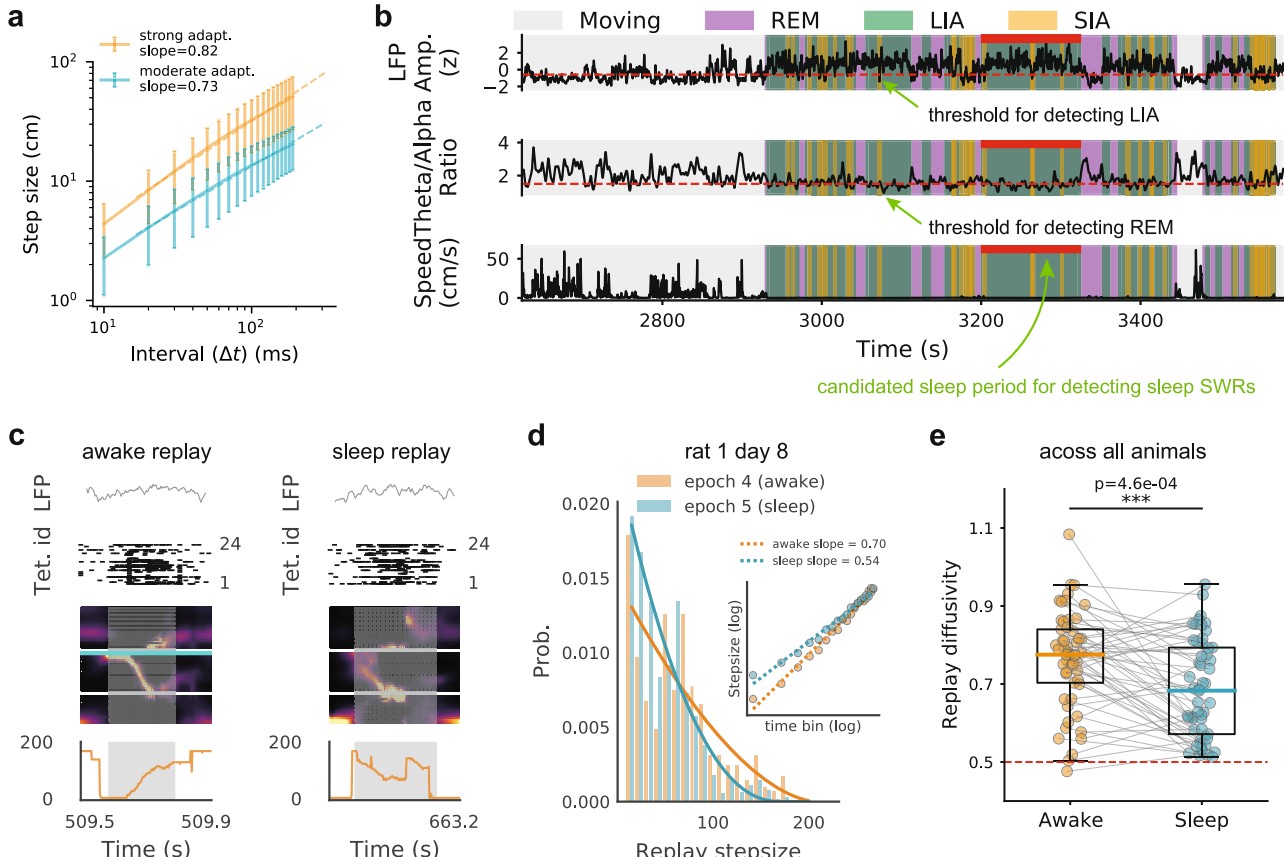

**Fig. 6 | Replay sequences exhibit greater diffusivity in the awake state compared to the subsequent sleep state. a** Replay diffusivity can be changed by tuning adaptation strength in the CAN model. **b** Identifying SWRS during the sleep state. Top, aggregated LFP from CA1, CA2, and CA3 tetrodes for low/high-amplitude LFP (SIA/LIA) period detection; middle, theta to delta ratio for REM period detection; bottom, running speed for immobile period detection. Red bars mark candidate sleep periods lasting at least 90 s with extended (>5 s) continuous LIA periods ("Methods"). Only ripple events within these periods were analysed further. **c** Examples of an awake replay (left) when an animal performed the W-track spatial alternation task and a subsequent sleep replay (right) when the animal was in the resting box. **d** Replay step size distribution (time bin = 2 ms) for awake (orange) and subsequent sleep (blue) states from two successive sessions for one animal. The inset panel illustrates step size versus time bins on a log-log scale for both awake and sleep replays, showing greater diffusivity in awake replay. **e** Comparison of diffusivity of awake replay (orange) and subsequent sleep replay (blue) across all recording sessions and animals. Each dot represents a replay diffusion exponent calculated from a recording session, with grey lines connecting values from successive running and sleep sessions. Awake replays show significantly higher diffusivity than sleep replays (Wilcoxon signed-rank test, $P = 4.6 \times 10^{-4}$).

network activity is driven by periodic modulation rather than stochastic noise fluctuations. At the peaks of these oscillations—corresponding to maximal excitatory drive—population activity is elevated, which counteracts the negative feedback from FRA and results in smaller replay step sizes (Fig. 5c, d). Conversely, at the troughs—where excitatory drive is minimal—population activity decreases, strengthening the effect of FRA and allowing the activity bump to shift between locations, producing larger step sizes. This anti-phase relationship between replay step size and population activity relative to external oscillations (Fig. 5e), particularly slow-gamma rhythms that intensify during replay events[33], demonstrates how FRA and rhythmic input interact to shape the temporal organisation of hippocampal replay.

**Awake replay is more diffusive than subsequent sleep replay**

Empirical studies have reported differences in replay diffusivity between awake and sleep states: awake replay during spatial memory tasks exhibits super-diffusive dynamics[33,58], whereas sleep replay following random foraging shows Brownian-like diffusion[34]. This difference can be accounted for in the CAN model by varying FRA strength (Fig. 6a). However, it remains unclear whether these differences persist within the same animal across behavioural states, or whether they

simply reflect differences in task type (spatial memory versus random foraging). To address this question, we identified sleep replay events (Fig. 6b; see "Methods") and analysed their diffusivity during post-task sleep following the spatial memory task[61]. If a difference in replay diffusivity is also observed within the same animal across awake and sleep periods, it would indicate that replay dynamics are modulated by behavioural state rather than by task type.

Specifically, sleep replay trajectories were decoded using the spatial tuning properties of cells recorded during the preceding running session (Fig. 6c), and one diffusion exponent were computed for one sleep session (Fig. 6d). The analysis revealed that sleep replay was significantly less diffusive than the preceding awake replay (53 paired awake-sleep recording epochs in total; Wilcoxon signed-rank test with $P = 4.6 \times 10^{-4}$) (Fig. 6e). These findings are consistent with a recent study[50], who reported similar results using an independent dataset[34]. Unlike our analysis, however, their study aggregated data across animals rather than performing a within-animal paired comparison. Notably, the diffusivity of sleep replay remained significantly greater than 0.5 (mean ± s.d.: 0.69 ± 0.12; one-sample Wilcoxon signed-rank test, $P = 2.4 \times 10^{-10}$; $n = 53$), indicating a deviation from the Brownian-like diffusion reported previously[34] (Fig. S10).

## Discussion

In this study, we introduced a theoretical framework to explain the diverse diffusivity of hippocampal replay by a CAN with FRA. Using a combination of theoretical and empirical analysis, our model reconciles a range of empirically observed replay behaviours-including stationary, Brownian, and super-diffusive patterns. It also generates testable predictions and provides mechanistic explanations for previously unexplained empirical observations. These findings indicate that flexible modulation of FRA might be a key mechanism governing hippocampal sequence dynamics. They also raise important questions about the computational role of replay and the broader implications for hippocampal function, which are explored further below.

There are two main classes of computational models that seek to explain hippocampal sequence generation. The first class focuses on intrinsic hippocampal circuit dynamics, incorporating mechanisms such as asymmetric synaptic connectivity[65], cholinergic modulation[66,67], firing-rate adaptation[36,53,54], and short-term plasticity[37,68,69]. Despite differences in biological implementations, these mechanisms all rely on symmetry breaking in attractor network dynamics, temporarily destabilising the activity bump and introducing autonomous transitions between network states. The second class focuses on the entorhinal-hippocampal system (EHC model)[50,70], which describes a linear feedback network between the two brain regions. In this framework, spectral modulation along the dorsoventral axis of medial entorhinal cortex (mEC) grid cells regulates the generative sampling of hippocampal state sequences. Several empirical evidence supports this view, showing that MEC input modulates the temporal organisation of hippocampal activity[71,72]. While sequence generation in our model does not depend on systematic modulation of mEC grid populations, it nevertheless shares important principles with the EHC framework, which can be seen from our theoretical analysis. Both models rely on feedback-based modulation to recirculate state updates: in the EHC model, this modulation arises from slow dynamics along a large spatial feedback loop between the two regions, whereas in our model it emerges from slow temporal dynamics due to FRA. Furthermore, whereas the EHC model generates distinct sequences by varying the spectral power of its generator−analogous to gain modulation across grid-cell modules −our model does so by varying adaptation strength. Mathematically, these differences could be reconciled through spectral analysis of CAN dynamics, which reveals a form of spectral modulation underlying sequence generation similar to the EHC model (Fig. 3 and Fig. S3). Computationally, these mechanisms are not mutually exclusive: both implement a form of slow feedback modulation− exemplified as FRA in our framework−and may jointly contribute to a unified explanation of hippocampal sequence generation.

The coordination between theta and replay sequences (Fig. 4) is consistent with their parallel development in the hippocampus[20,35]. Replays are initially stationary in pre-weaning animals and become progressively more sequential after weaning, coinciding with the maturation of theta sequences. This development likely reflects the gradual emergence of attractor dynamics in the hippocampus[73], supported by increased co-firing among cells with overlapping place fields during SWRs[20]. In our model, a more mature attractor network corresponds to stronger recurrent connectivity, within which FRA drives smoother bump movement, resulting in more diffusive replay sequences and longer theta sweeps. Previous experimental work has also identified a causal relationship between theta and replay sequences, showing degraded replay after disruption of theta sequences[19]. While this model does not directly establish causality, we propose that perturbing theta sequences may weaken network connectivity via Hebbian plasticity[10,17,74]. Weakened connectivity would reduce the ability of the FRA to propagate the activity bump, thereby explaining the causal dependence of replay on theta sequences. Incorporating synaptic plasticity into future

computational models can provide a richer understanding of how hippocampal circuits coordinate theta and replay sequences across development and behaviour.

The higher diffusivity of awake replay, compared with sleep replay, more closely resembles the diffusivity of behavioural trajectories. This observation suggests that awake replay shares greater similarity with the animal's movement dynamics, for example, reflecting goal-directed planning that supports future navigation towards a goal[31]. In contrast, sleep replay may contribute more strongly to generalisation across multiple environments, extracting and recombining information to support the formation of novel cognitive maps[34,75]. Consequently, it exhibits less diffusive dynamics that deviate from preceding behavioural trajectories. From a modelling perspective, this difference can be explained by the dynamic modulation of adaptation strength across brain states. Although direct evidence for such state-dependent variation in adaptation is currently lacking, in vitro whole-cell recordings suggest that neuromodulators−such as acetylcholine−can modulate adaptation strength differently during awake and sleep states[41,43], which could be tested in future experiments.

FRA in biological systems is hypothesised to serve roles extending beyond the initiation of hippocampal replay[36,76]. For instance, FRA has been proposed to facilitate mental exploration[53], generate theta sequences during movement[25,26,38], and enable efficient sampling-based Bayesian inference of sensory information[77]. It may also function as a high-pass filter, separating transient signals from slower oscillatory signals to improve signal processing[78]. These potential roles suggest that FRA is not merely a by-product of intrinsic dynamics that limit over-excitation, but rather a mechanism supporting specific neural computations and energy-efficient coding strategies. Although FRA is ubiquitous across neural systems, its hypothesised computational functions remain to be directly tested. Moreover, whether cognitive processes can exert top-down control over FRA represents an open question for future investigation.

## Methods

### Experimental data

The dataset used in this study has been described in detail in ref. [61]. On each experimental day, each animal (9 male Long Evans rats in total) performed a 15-min running epoch on a W-shaped track, flanked by 20-min rest sessions in a rest box. The W-shaped track had one reward site at the endpoint of each arm, and the animals were rewarded for performing a continuous alternation task (Fig. S2)[79]. Neural recordings were obtained via a microdrive array containing 30 independently movable tetrodes targeting CA1, CA2, CA3, MEC, Subiculum, and DG, depending on the animal. For analysis in the current study, we only included tetrodes in CA1, CA2, and CA3. Each running epoch consisted of 10−24 tetrodes (140 valid epochs in total; 2.6 ± 0.7 epochs per day, 5.9 ± 2.4 recording days per animal). Multiunit spikes were then obtained as any potential exceeding a 60 $\mu$V threshold on any one of the four tetrode wires. The waveform was identified as the electrical potential value on each wire of the tetrode at the time of maximum potential of any of the four wires. All interneurons were excluded from analysis when the spike widths were less than 0.3 ms based on the waveform feature.

### Identifying SWRs during awake

Detection of sharp wave and ripple events (SWRs) was performed only when at least three CA1 cell layer recordings were available, following the method described in ref. [80]. Specifically, LFPs from all available CA1 cell layer tetrodes were band-pass filtered between 150 and 250 Hz, then squared and summed across tetrodes. This sum forms a single population trace over time, which was then smoothed with a Gaussian kernel ($\sigma$ = 4 ms) and square-rooted. Candidate SWR times were detected when the z-scored signal exceeded 2 standard

deviations for at least 15 ms and the animal's speed was less than 4 cm/s. The detected SWR periods were then extended before and after the threshold crossings to include the time until the population trace returned to the mean value. Only SWRs with spikes from at least two tetrodes were included in replay analysis.

### Identifying SWRs during sleep

To identify SWRs in sleep states in the rest box, we followed the method described in ref. 80. First, we extracted periods when the animals' speed was <4 cm/s for longer than 60 s. Second, we detected REM periods following the method in ref. 81. Specifically, for all CA1 tetrodes, the ratio of Hilbert amplitudes (smoothed with a Gaussian kernel, $\sigma$ = 1 s) of theta (5–11 Hz) to delta (1–4 Hz) filtered LFP was calculated, and then the mean was taken over tetrodes. REM periods were identified as sustained periods (10 s minimum duration) in which the theta to delta ratio was elevated above a manually set threshold (range: 1.2–1.8). Third, after excluding REM periods, LFPs from all available CA1, CA2, and CA3 tetrodes were squared, smoothed with a Gaussian kernel ($\sigma$ = 0.3 s), and then z-scored and summed across all tetrodes. The sum trace was again z-scored to obtain an aggregate hippocampal LFP amplitude. A bimodality was observed in the histogram distribution of the aggregated LFP amplitude[80]. Fourth, SIA periods were defined as non-REM times in which the aggregate LFP amplitude was below the threshold separating the two modes on the histogram, and LIA periods otherwise. LIA periods were therefore high-amplitude LFP, corresponding to a hippocampal sleep state dominated by SWRs, frequently interrupted by periods of low-amplitude LFP, i.e., SIA. Only SWRs within those candidate sleep periods at least 90 s in duration, which contained extended (>5 s) continuous LIA periods, were included in sleep replay analysis.

### The state space decoder

The state space model was described in detail in ref. 59. One of the advantages of the state space decoder is its use of very small temporal time bins (2 ms) to perform a moment-by-moment estimation of the position representation (compared to the standard "Bayesian" decoder with >20 ms time bins). This property allows for the detection of rapid representational movement and therefore captures a range of movement dynamics, including super-diffusion (trajectories with interleaved slow progression and large jumps), stationary (unchanging representations), and diffusion (trajectories that progress at variable or constant speeds).

The state space model takes two inputs: the multiunit spikes described above and the linearised position of the animal. The linearisation was done by converting the 2D position of the animal into a single line extending from the home well (central arm) to the edge of the left and right arms (central arm, right arm, and then left arm; see Fig. S2b). The linearised position was then binned into 3 cm spatial bins within each arm. Compared to decoding in a 2D scenario, such linearisation significantly speeds up the decoding process. We adopted the "clusterless" version of the state-space decoder, which directly uses multiunit spikes and their spike waveform features to decode position without spike sorting[62]. This clusterless decoder allows access to a larger pool of data than the spike-sorted decoder, providing more information about the extent of the event and greater certainty in the latent position representation[59].

We then ran the state space model independently for each session. Before decoding replay sequences and theta sequences, we first validated the model with the decoding accuracy of the animals' locations during running (with animals' speed >4 cm/s). We found that the decoded position closely tracks the animal's actual position (fivefold cross-validation; with the distance difference between the actual and decoded location having a median of 9.0 cm, and 95% CI of 8.2 – 9.8 cm), which is comparable to previous studies[35,82].

### Decoding of replay sequences

For the state space model of replay decoding, we built the encoding model using all multiunit spikes when the animal was moving >4 cm/s and a 2-ms temporal bin. Three movement dynamics were included in the encoding model: (1) continuous movement dynamic—with an equal probability of moving one position bin forward or back; (2) stationary movement dynamic—staying in the same position; (3) fragment movement dynamic—with an equal probability of moving to any other possible position bin. The transition matrix, which defines how likely the movement state is to switch to another state versus persist in the same state, is set to have a 0.98 probability of persisting in the same state and 0.01 probability of switching to one of the other two states. After training the encoding model, we applied it to SWRs during both the awake immobile state and the subsequent sleep state. The main output of the model is the posterior probability of position, which is estimated by marginalising the joint probability over the dynamics. This indicates the most probable "mental" positions of the animal based on the data. For replay diffusivity, we fit the slope between replay step size and time duration on a log–log scale.

### Decoding of theta sequences

For the state space model of theta sequence decoding, we again built the encoding model using a 2-ms temporal bin. However, we only considered two movement dynamics: continuous and fragmented. The continuous dynamic was modelled by a random-walk transition matrix with a 6-cm standard deviation, and the fragmented dynamic was modelled by a uniform transition matrix. The probability of staying in either the continuous or the fragmented movement dynamic was set to 0.968, which corresponds to 62.5 ms of staying in the same movement dynamic on average, or roughly the duration of half a theta cycle. The probability of transitioning to the other movement dynamic is 0.032.

We used fivefold cross-validation for decoding, in which we built the encoding model on 4 folds of the data and then decoded the sequences on the remaining fifth fold of the data. This ensures that the data used for constructing a given encoding model are not also used for decoding the representation. We repeated this for each fifth of the data. It is also noteworthy that we didn't perform cross-validation for replay decoding since the training data and the testing data are naturally separated from each other. To calculate the average theta sequence length for each recording session, we averaged the theta sweep lengths (look-ahead distance plus look-back distance) across all LFP theta cycles during running.

### Diffusion exponent analysis of replay trajectories

The power-law distribution of step size $p(\|\Delta z\|) \sim \|\Delta z\|^{-1-\alpha}$ (Eq. (3)) rigorously characterises the diffusivity of a movement trajectory. However, the condition of obtaining the probability distribution is as $\Delta t \rightarrow 0$. This implies that, to quantify the diffusivity of replay trajectories, we need to fit a power-law distribution of the step size in infinitesimal time bins. However, in experimental data analysis, the decoding time bin is typically set between 2 and 20 ms (depending on the decoding algorithm), posing a challenge for the fitting process. Therefore, we followed previous work[34,58] to quantify the diffusivity of a movement trajectory as the relationship between distance and time, which is expressed as:

$$\langle d(t)^2 \rangle^{\frac{1}{2}} = Gt^\eta, \tag{6}$$

where $d$ is the distance between two points in time within a replay trajectory, $t$ is the time elapsed between those two time points, $G$ is a constant describing the scale of diffusion, and $\eta$ is the diffusion exponent. To quantify the diffusivity of replay trajectories in experimental data, we plot the relationship between $d(t)$ and $t$ in log-log space, and use linear regression applied to this log-log plot to find the

slope, which corresponds to the diffusion exponent $\eta$. A slope (or $\eta$) of 0.5 corresponds to Brownian diffusion, whereas a slope (or $\eta$) greater than 0.5 corresponds to superdiffusion.

Instead of quantifying the diffusivity of each replay trajectory (which contains only a few time bins, and therefore does not have enough data points for fitting), we quantify the overall diffusivity of replay trajectories in one recording session by merging all step sizes and their corresponding time bins from all replay trajectories in the recording epoch, and calculate one diffusion exponent by linearly fitting these two variables in the log scale. This allows us to quantify the overall diffusivity of replay dynamics from one recording epoch. Specifically, we calculate the distance $d^k(\Delta t_j)$ between all pairs of decoded positions separated by all multiples of the time bin used for trajectory decoding $\Delta t_j / \delta t$ where $\Delta t_j$ is the time elapsed between decoded positions and $\delta t$ is the unit decoding time bin. $k$ is the $k_{th}$ replay trajectory. This resulted in a set of distance-time pairs in one recording session:

$$\left( d^k(\Delta t_j), \Delta t_j/\delta t \right)_{j=1:J, \, k=1:K}, \tag{7}$$

with $J$ the number of elapsed time bins and $K$ the number of replay trajectories in one recording session. We then calculate the mean distance $\bar{d}(\Delta t_j)$ for each multiple $j$ of the time bin by taking the average over all trajectories and fit a linear regression model to $(\log \bar{d}(\Delta t_j), \log(\Delta t_j/\delta t))$, which gives the estimate of the diffusion exponent $\eta$ for the recording session as the slope of the regression.

The two descriptions of diffusivity (Eqs. (3) and (6)) are interrelated to each other. Movement trajectories with the diffusion exponent $\eta = 0.5$ or the power-law exponent $\alpha \geq 2$ follow Brownian diffusion, while movement trajectories with the diffusion exponent $\eta > 0.5$ or the power-law exponent $1 < \alpha < 2$ follow super-diffusion. A slight difference is that Eq. (3) with $1 < \alpha < 2$ describes a sub-type of super-diffusion, that is, Lévy flights (see our previous work[83]), which is composed of frequent local motion and intermittent long-jump motion; while Eq. (6) with $\eta > 0.5$ describes a more general super-diffusion, which includes not only Lévy flights, but also replay dynamics with a constant fast speed. Since both cases have been observed in previous works[33,58,59], we use the diffusion exponent (Eq. (6)) to describe experimental results while use the power exponent (Eq. (3)) to describe the model results.

**Measuring the place field size index (population vector correlation analysis)**
To check the confounding factor of place field size in contributing to the positive correlation of replay diffusivity and theta sweep length, we calculate the place field size index for each recording session via the population vector correlation analysis (PVC)[63]. The PVC is a measure of how quickly the spatial firing patterns change with distance (i.e., how quickly the vectors decorrelate), and therefore implies the average size of place fields. Since we used the clusterless version of the state space model, we treat each recording unit as a "cell". For each recording session, the rate matrix was constructed by arranging all the recording units into a two-dimensional matrix $M_n^l$, where the unit identities ($n$ in total) are represented on the first dimension and the spatial intensity of unit activity ($l$ bins in total) on the second dimension. As in building the encoding model for the state space decoder, the spatial intensity is calculated using the data when the animal's speed is more than 4 cm/s. To provide an estimate of the similarity of the hippocampal neuronal ensemble code, we calculate the Pearson correlation coefficient for each pair of $M$-dimension population vectors at two spatial bins, and obtain a $l \times l$ correlation coefficient matrix (Fig. S7b). Since the animals perform a spatial alternation task between the left and right arm, we further split the correlation coefficient matrix into two matrices, one represents the spatial similarity from the central to the left arm and the other represents the spatial similarity from the central to the right arm (Fig. S7b). The mean similarity value along the diagonal is calculated and further smoothed with a Gaussian kernel with $\sigma = 5$ for both arms, and then averaged across two arms. This results in a population vector correlation curve as a function of the diagonal offset (Fig. S7c). Finally, the place field index is calculated as the slope from the peak to the trough value on the curve, which measures how quickly the PVC decays among the spatial bins. A large place field index (large slope) represents a quick decay of the PVC value along the spatial bins, and hence indicates smaller place fields in the current recording session.

**The continuous attractor network (CAN) model with FRA**
The dynamics of the CAN model with FRA is written as:

$$\tau_u \frac{\partial U(x,t)}{\partial t} = -U(x,t) + \rho \int_{x'} J(x,x')r(x',t)\mathrm{d}x' - V(x,t) + I_{ext}(x,t) + \sigma_U \xi_U(x,t), \tag{8}$$

$$r(x,t) = \frac{U^2(x,t)}{1 + k\rho \int_{x'} U^2(x',t)\mathrm{d}x'}, \tag{9}$$

$$\tau_v \frac{\partial V(x,t)}{\partial t} = -V(x,t) + mU(x,t) + \sigma_m U(x,t)\xi_V(x,t). \tag{10}$$

Equation (8) describes the dynamics of the presynaptic current of a neuron in the CAN, with the firing rate (Eq. (9)) implemented as a global inhibition. Eq. (10) describes the dynamics of FRA, which is negatively fed back to the dynamics of the presynaptic current. $m$ is the adaptation strength. $\rho$ represents the place cell density covering the environment, and $k$ represents the strength of the global inhibition. $\sigma_U$ and $\sigma_m$ represent the noise amplitude applied to $U$ and $V$, respectively. $\tau_u$ is the neuronal time constant, and $\tau_v$ is the adaptation time constant, which is much larger than $\tau_u$, highlighting the feature of slow feedback inhibition. FRA (whether reflecting e.g., spike frequency adaptation or short-term depression) is considered as a slow dynamic process because it involves mechanisms that operate on timescales longer than the action potentials themselves. For instance, spike frequency adaptation is mediated by ion channels that open or close more slowly than the voltage-gated sodium and potassium channels responsible for the rapid upstroke and repolarization of action potentials. Examples include calcium-activated potassium channels and slow voltage-gated potassium channels. Additionally, calcium accumulation and subsequent clearance (via pumps or buffers) occur on much slower timescales compared to the millisecond duration of individual spikes, further contributing to the slow nature of adaptation. Similarly, short-term depression, another form of FRA, is caused by the depletion of neurotransmitter vesicles at the presynaptic terminal during high-frequency activity. The replenishment of vesicle pools depends on processes such as vesicle docking, priming, and mobilisation from reserve pools, which typically take tens to hundreds of milliseconds or longer.

The synaptic connection $J(x,x')$ is translation-invariant, which is written as:

$$J(x,x') = \frac{J_0}{2\pi a^2} \exp\left[ -\frac{\| x - x' \|^2}{2a^2} \right], \tag{11}$$

with $J_0$ controlling the strength of the recurrent connection and $a$ controlling the range of neuronal interaction. This translation-invariant form indicates that the synaptic strength of two place cells only depends on the distance between the locations of their place fields.

**Table 1 | Common parameter settings of the one-dimensional CAN model**

| Parameters | Values |
|---|---|
| **Common parameter setting** | |
| Number of neurons: $N$ | 128 |
| Neuron density: $\rho$ | 20 |
| Recurrent connection range (Gaussian width): $a$ | 0.4 m |
| Synaptic connection strength: $J_0$ | 4 |
| Global inhibition strength: $k$ | 20 |
| Time constant of neural firing: $\tau_u$ | 1 ms |
| Time constant of FRA: $\tau_v$ | 48 ms |
| Simulation time interval: $\delta t$ | 0.1 ms |
| **Common parameter setting for replay sequences** | |
| Location-dependent input strength: $\beta$ | 0 |
| Animals' running speed: $v_{ext}$ | 0 m/s |
| **Common parameter setting for theta sequences** | |
| Location-dependent input strength: $\beta$ | 0.01 |
| Animals' running speed: $v_{ext}$ | 1.5 m/s |

**Table 2 | Parameter settings of the adaptation strength and the noise level in generating a replay sequence with different diffusivity and theta sequences with different amplitudes**

| Parameters | Values |
|---|---|
| Replay with weak adaptation (Fig. 2d) | $m = 0, \gamma = 0$ |
| Replay with moderate adaptation (Fig. 2e) | $m = 0.03, \gamma = 0$ |
| Replay with strong adaptation (Fig. 2f) | $m = 0.07, \gamma = 0$ |
| Stationary replay (Fig. 3a) | $m = 0.004, \gamma = 0.05$ |
| Brownian-diffusive replay (Fig. 3a) | $m = 0.008, \gamma = 0.6$ |
| Superdiffusive replay (Fig. 3a) | $m = 0.019, \gamma = 0.95$ |
| Short theta sequences (Fig. 4a) | $m = 0.12, \gamma = 0$ |
| Long theta sequences (Fig. 4a) | $m = 0.19, \gamma = 0$ |
| Theta vs. replay (Fig. 4b) | $m \in [0.04, 0.08], \gamma = 0.1$ |

The location-dependent sensory-motor input $I^{ext}(x, t)$ is modelled as a bump input conveying the information of the animal's physical location to the CAN. It is expressed as:

$$I^{ext}(x,t) = \beta \exp\left[-\frac{(x - v_{ext}t)^2}{2a^2}\right], \tag{12}$$

where $\beta$ controls the input strength, and $v_{ext}$ represents the moving speed of the artificial animal. For simplicity, we modelled the animal's movement with a constant speed. To generate theta sequences in the model, we set $\beta > 0$ to mimic the interplay between external sensory input and the intrinsic dynamics from FRA in the hippocampus (see "Methods" below). Conversely, to generate replay sequences, we set $\beta = 0$, simulating the absence of location-dependent sensory input during resting states, allowing the network state to evolve solely based on intrinsic dynamics.

Common parameters used to simulate replay-like sequences and theta sequences in the CAN model are summarised in Table 1. For the key parameters, i.e., the adaptation strength $m$ and the noise level $\gamma$ (see Eq. (4)) for generating replay sequences with different diffusivity and theta sequences with different amplitudes are summarised in Table 2. All simulations were conducted using the first-order Euler method.

**Parameter conditions to generate bump activity in the CAN model.** The global feedback inhibition (Eq. (9)) and the distance-dependent

recurrent synaptic connection (Eq. (11)) prevent the neural activity from spreading in the CAN, and hence can result in a bump-like activity profile as the network state. However, from a mathematical perspective, this bump-like network state is not a trivial solution of the network dynamics. For instance, when the global inhibition is strong (large value of $k$), the activity bump may not survive. Following the theoretical analysis in ref. 84, we now derive the parameter conditions necessary to ensure the emergence of the bump state.

For simplicity, we consider that in the CAN model, there are only the recurrent connections and the global feedback inhibition, allowing us to investigate how these two factors affect the emergence of bump activity in the network. The network dynamics are then expressed as:

$$\tau_u \frac{\partial U(x,t)}{\partial t} = -U(x,t) + \rho \int_{x'} J(x,x')r(x',t)\mathrm{d}x', \tag{13}$$

$$r(x,t) = \frac{U^2(x,t)}{1 + k\rho \int_{x'} U^2(x',t)\mathrm{d}x'}. \tag{14}$$

We assume the activity bump has the following profile (if it exists in the simple network):

$$\overline{U}(x,t) = A_u \exp\left\{-\frac{[x - z(t)]^2}{4a^2}\right\}, \tag{15}$$

$$\overline{r}(x,t) = A_r \exp\left\{-\frac{[x - z(t)]^2}{2a^2}\right\}, \tag{16}$$

where $A_u$, $A_r$ represent the bump heights of $U(x, t)$ and $r(x, t)$, respectively. $z(t)$ is the bump location, and $a$ is the range of neuronal interaction. We then substitute Eqs. (15) and (16) into Eqs. (13) and (14) which gives:

$$\tau_u \frac{dA_u}{dt} = -A_u + \frac{\rho J_0}{\sqrt{2}} A_r, \tag{17}$$

$$A_r = \frac{A_u^2}{1 + \sqrt{2\pi} k\rho a A_u^2}, \tag{18}$$

For the activity bump to exist in the CAN, the bump height should have a fixed positive value, which means $dA_u/dt = 0$, i.e., $A_u = \rho J_0 A_r/\sqrt{2}$ in Eq. (17). Combining it with Eq. (18), we obtain the solutions of $A_u$ and $A_r$, which are:

$$A_u = \frac{\rho J_0 \pm \sqrt{\rho^2 J_0^2 - 8\sqrt{2\pi}k\rho a}}{4\sqrt{\pi}k\rho a}, \tag{19}$$

$$A_r = \frac{\sqrt{2}}{\rho J_0} A_u. \tag{20}$$

For $A_u$ to exist, "$\rho^2 J_0^2 - 8\sqrt{2\pi}k\rho a$" should be non-negative, which means $k < \rho J_0^2/(8\sqrt{2\pi}a)$ should be met. In summary, the condition that the CAN generates bump activity as its network state is that the global inhibition strength $k$ is set smaller than a threshold determined by three other parameters in the CAN model, that is, the neuronal density $\rho$, the recurrent connection strength $J_0$ and the neuronal interaction range $a$. In order to obtain a meaningful representation of the environment in the hippocampal place cell network (equivalent to localised bump activity in the CAN), we always choose $k$ below the threshold throughout the paper.

**Deriving the relationship between bump intrinsic speed and adaptation strength.** FRA introduces instability to the activity bump, thereby causing intrinsic movement of the activity bump when there is no external input drive (Fig. 2c–f). A typical feature in Fig. 2c highlights that there exists a state transition boundary in the adaptation strength, below which the bump stays stationary, and above which the bump moves faster under stronger adaptation strength. We here theoretically analyse how the adaptation strength affects the movement of the activity bump.

To simplify the analysis, we consider a noise-free network where $\sigma_U$ and $\sigma_V$ are all zero. We again assume the network activity has the following bump profile (now including the $V(x, t)$):

$$\overline{U}(x, t) = A_u \exp\left\{ -\frac{[x - z(t)]^2}{4a^2} \right\}, \tag{21}$$

$$\overline{r}(x, t) = A_r \exp\left\{ -\frac{[x - z(t)]^2}{2a^2} \right\}, \tag{22}$$

$$\overline{V}(x, t) = A_v \exp\left\{ -\frac{[x - (z(t) - s(t))]^2}{4a^2} \right\}. \tag{23}$$

Here $A_v$ is the bump height of the adaptation effect, and $z(t)$ is the bump centre. The intrinsic speed of the bump under FRA is then described as $dz(t)/dt$ (marked as $v_{int}$ below). $s(t)$ in Eq. (23) indicates the displacement between the position of the $U$ bump and the $V$ bump. Without loss of generality, we assume that the intrinsic movement of the bump is from left to right on the linear track, with 0 located at the left side. Therefore, $dz(t)/dt > 0$ always holds, indicating the bump travels to the right, and $s(t) > 0$ holds, indicating $V(x, t)$ lags behind $U(x, t)$ due to the slow dynamics in FRA ($\tau_v \gg \tau_u$).

Following the analysis in ref. 57, we can solve the network dynamics by utilising an important property of the CAN, that is, the dynamics of a CAN are dominated by a few motion modes corresponding to different distortions in the shape of a bump. Specifically, we can project the network dynamics onto these dominant modes and simplify the network dynamics significantly. The first two dominant motion modes used in the present study correspond to the distortions in the height and position of the Gaussian bump, which are given by

$$u_0(x|z) = \frac{1}{a\sqrt{2\pi}} \exp\left\{ -\frac{[x - z(t)]^2}{4a^2} \right\}, \tag{24}$$

$$u_1(x|z) = \frac{1}{a^2\sqrt{2\pi}} [x - z(t)] \exp\left\{ -\frac{[x - z(t)]^2}{4a^2} \right\}. \tag{25}$$

Projecting a function $f(x)$ on a mode $u(x)$ means computing $\int_x f(x)u(x) dx$. We first substitute the assumed network states (Eqs. (21)–(23)) into the network dynamics (Eqs. (8)–(10)) and then apply the projection method to simplify the dynamics, and then we obtain the intrinsic speed of the bump under FRA $v_{int}$, with

$$v_{int} = \frac{2a}{\tau_v} \sqrt{\frac{m\tau_v}{\tau_u} - \sqrt{\frac{m\tau_v}{\tau_u}}}. \tag{26}$$

When $m > \tau_u/\tau_v$, the intrinsic speed $v_{int}$ is positive. We denote $\tau_u/\tau_v \equiv m_0$ as the transition boundary, below which the activity bump stays stationary and above which the activity bump moves intrinsically on the linear track. The intrinsic dynamics of the network depend on three factors in the network, i.e., the neuronal interaction range $a$ (also controls the place field size), the time constant $\tau_u$ and $\tau_v$ and the adaptation strength $m$. For instance, when the adaptation strength

increases, the bump moves faster (Fig. 2c); when neurons have a larger place field size, i.e., interacting more with each other, the bump also moves faster.

**Deriving the position dynamics from the CAN model.** We have shown that without noise, the activity bump exhibits intrinsic dynamics under the destabilization of FRA. Now we investigate the bump dynamics under the joint effects of FRA and network noise. The key idea behind the derivation is that the CAN dynamics (in the N-dimensional neural space) can be simplified into the bump position dynamics and bump height dynamics over time (both of which are in one-dimensional space) by a projection method[85] (similar to the analysis in "Methods" above). For the one-dimensional dynamics, it is much easier to get the theoretical solution regarding how the position evolves over time, i.e., the quantification of replay diffusivity in the main text.

Specifically, we first substitute the assumed network states (Eqs. (21)–(23)) into the network dynamics (Eqs. (8)–(10)), and then project the network dynamics onto the bump height model (Eq. (24)). This operation gives us the dynamics of bump heights which are:

$$\tau_u \frac{dA_u}{dt} = -A_u - A_v + \frac{J_0 \rho A_r}{2} + \frac{\sigma_U}{a\sqrt{2\pi}} \xi_{U,0}(t), \tag{27}$$

$$\tau_v \frac{dA_v}{dt} = -A_v + mA_u + \frac{\sigma_m A_u}{2a\sqrt{\pi}} \xi_{V,0}(t), \tag{28}$$

where $\xi_{U,0}(t)$ and $\xi_{V,0}(t)$ denote, respectively, the projected noises of $\xi_U(t)$ and $\xi_V(t)$ on the height mode, which are still Gaussian white noises of zero mean and unit variance.

Second, we project the network dynamics onto the position mode (Eq. (25)), and obtain the dynamics of bump positions which are:

$$\tau_u \frac{dz}{dt} = \frac{A_v}{A_u} s + \frac{\sigma_U}{A_u} \sqrt{\frac{2}{\pi}} \xi_{U,1}(t), \tag{29}$$

$$\tau_v \frac{ds}{dt} = \left( \frac{\tau_v A_v}{\tau_u A_u} - \frac{mA_u}{A_v} - \frac{\sigma_m A_u \xi_{V,0}}{2A_v\sqrt{\pi}a} \right) s + \frac{\tau_v \sigma_U}{\tau_u A_u} \sqrt{\frac{2}{\pi}} \xi_{U,1}(t) - \frac{\sigma_m A_u}{A_v} \sqrt{\frac{1}{2\pi}} \xi_{v,1}, \tag{30}$$

where $\xi_{U,1}(t)$ and $\xi_{V,1}(t)$ denote, respectively, the projected noises of $\xi_U(t)$ and $\xi_V(t)$ on the position mode, which are also Gaussian white noises of zero mean and unit variance.

Equations (27) and (28) can be described by the Fokker-Planck equations, which when solved, give the stationary distributions of $A_u$ and $A_v$. In addition, since $\sigma_U$ and $\sigma_m$ are relatively small, we ignore the variances of $A_u$ and $A_v$ and keep their mean values. Therefore, Eqs. (29) and (30) can be further simplified by replacing $A_u$ and $A_v$ with their mean values $\widetilde{A}_u$ and $\widetilde{A}_v$. Together with the approximation of $\widetilde{A}_v = m\widetilde{A}_u$ (according to Eq. (28)), Eqs. (29) and (30) can be written as:

$$\tau_u \frac{dz}{dt} = ms + \frac{\sigma_U}{\widetilde{A}_u} \sqrt{\frac{2}{\pi}} \xi_{U,1}(t), \tag{31}$$

$$\tau_v \frac{ds}{dt} = -\left[ 1 - \frac{\tau_v}{\tau_u} m + \frac{\sigma_m}{2\sqrt{\pi}am} \xi_{V,0}(t) \right] s + \sqrt{\frac{2}{\pi} \left( \frac{\tau_v \sigma_U}{\tau_u \widetilde{A}_u} \right)^2 + \frac{1}{2\pi} \left( \frac{\sigma_m}{m} \right)^2} \xi_s(t), \tag{32}$$

where $\xi_s(t)$ is a Gaussian white noise of zero mean and unit variance (by combining the last two noise terms in Eq. (30)). We define $1 - \tau_v m/\tau_u \equiv \mu$, which quantifies the normalised distance of the adaptation strength to the transition boundary $m_0$, and define $\sigma_m/(2\sqrt{\pi}am) \equiv \gamma$, which quantifies the normalised noise amplitude.

We also rewrite the noise terms in Eqs. (31) and (32). After these operations, we obtain the position dynamics under the drive of both FRA and noise fluctuations, which is:

$$\tau_u \frac{dz}{dt} = ms + a_z \xi_z(t) \tag{33}$$

$$\tau_v \frac{ds}{dt} = -\left[\mu + \gamma \xi_s^1(t)\right]s + a_s \xi_s^2(t). \tag{34}$$

Equations (33) and (34) are typical Langevin dynamics, showing that the position dynamics $z(t)$ is determined by a drift term reflecting the contribution of FRA and a diffusion term reflecting the contribution of network noise. In fact, the position dynamics $z(t)$ is a second-order variable which depends on the dynamics of $s$, i.e., the displacement of bump $U(x, t)$ and bump $V(x, t)$. For instance, when the adaptation strength is set far below the transition boundary (small $m$ and large $\mu$), $s$ decays quickly to zero, and $z$ is determined only by the noise diffusion term $a_z \xi_z(t)$, and hence exhibit the dynamics of Brownian motion; when the adaptation strength is set near the transition boundary (large $m$ and small $\mu$), $z$ is determined by both the drift and the noise diffusion term, and hence exhibits super-diffusive dynamics.

It is noteworthy that the noise term $\gamma \xi_s^1(t)$ in Eq. (34) is necessary for generating the super-diffusive dynamics. If $\gamma = 0$, Eq. (34) becomes an Ornstein–Uhlenbeck (OU) process, and the stationary distribution of $s(t)$ has a Gaussian form, which leads to two additive noises in Eq. (33), and the position dynamics only exhibits Brownian motion. We will quantify the diffusivity in a power-law expression of the step size below.

**Obtaining the probability distribution of the step size from the position dynamics.** The position dynamics (Eqs. (33) and (34)) are a second-order process. Therefore, to solve the position dynamics of $z(t)$, we first solve the dynamics of $s(t)$. Following the analysis in ref. 83, we can describe the dynamics of $s(t)$ as a Fokker-Planck equation and obtain the probability distribution of $s(t)$ which has the form of a power law:

$$p(s) = c_0 \left(\sigma_s^2 + \gamma^2 s^2\right)^{-(1+\mu/\gamma^2)}, \tag{35}$$

where $\sigma_s^2 = (\sqrt{2}\tau_v \sigma_U/(\sqrt{\pi}\tau_u \widetilde{A}_u))^2 + (\sqrt{2}a\gamma)^2$ and $c_0$ is a normalisation constant. The dynamics of $z(t)$ (Eq. (33)) shows that the step size of the activity bump in $\Delta t$ is:

$$\| \Delta z \| = \| ms\Delta t/\tau + \sqrt{2\Delta t/(\pi\tau)}\sigma_U/\widetilde{A}_u \xi_{U,1} \|, \tag{36}$$

with $\Delta t \rightarrow 0$. Therefore, by replacing $s$ with its stationary distribution given by Eq. (35), we obtain the power-law distribution of the step size $\|\Delta z\|$, which is written as,

$$p(\| \Delta z \|) \sim \| \Delta z \|^{-1-(1+2\mu/\gamma^2)}. \tag{37}$$

Equation (37) shows that increasing the adaptation strength (decreasing the value of $\mu$) and/or increasing the noise level (increasing the value of $\gamma$) can increase the probability of a large step size, that is, the probability of long-jump movements of the activity bump on the linear track. This power-law distribution corresponds to Levy walks where the activity bump traverses the intervening positions before stabilising at the final position, rather than Levy flights where the activity bump "jumps" to the final position without traversing the intervening positions (see ref. 50 for more details). This is because we assumed the absence of external input when modelling replay-like dynamics, the activity bump moves continuously through the space rather than making instantaneous "jumps" to a new location. In empirical data, distinguishing between Levy walks and Levy flights is challenging, with the difficulty arising partly because current neural recordings are limited to hundreds of cells

during navigation in open fields. This limited coverage can result in unevenly distributed place fields, introducing noise into the decoded locations and creating apparent "jumps".

### Generating theta sequences in the CAN

Theta sequences have been hypothesised to result from the interaction of external location-dependent sensory input and the intrinsic network dynamics[21,65]. Therefore, following previous work[38], we generated theta sequences in the CAN model by activating the location-dependent sensory input $I_{ext}(x, t)$ (Eq. (12)). The external input is modelled as an activity bump travelling with a constant speed, simulating the update of the animal's physical location in the environment. The interaction of external input and the intrinsic dynamics creates a push-pull effect on the activity bump: as the animal advances, the external input exerts a constant pull effect on the activity bump, attracting it back to the current physical location, while slow feedback inhibition (adaptation) pushes it away from the current physical location. This results in theta-like sweeps of the location representation as the animal explores the environment (see Fig. 4a and Table 1 for parameter settings). Intriguingly, akin to how the adaptation strength governs replay diffusivity in the CAN model, it also regulates the sweep amplitude of generated theta sequences. Specifically, stronger adaptation results in a larger sweep amplitude, as illustrated by comparing Fig. 4a, left and right. This phenomenon arises from the increased intrinsic mobility in the activity bump associated with stronger adaptation, causing it to sweep further during the push effect.

### CAN model with external slow-gamma oscillation

To show the phase-locking phenomenon of movement step sizes and neural activity to the slow-gamma oscillation during sleep SWRs, we consider a CAN with an external input oscillating at the slow-gamma rhythm, which is written as:

$$\tau_u \frac{\partial U(x, t)}{\partial t} = -U(x, t) + \rho \int_{x'} J(x, x')r(x', t)dx' \\ - V(x, t) + I_\gamma(x, t) + \sigma_U \xi_U(x, t), \tag{38}$$

where $I_\gamma(x, t)$ has a sinusoidal waveform given by:

$$I_\gamma(x, t) = A\sin(\omega t + \phi), \tag{39}$$

with $A$ representing the amplitude, $\omega$ the angular frequency, and $\phi$ the initial phase of the slow-gamma rhythm. Without loss of generality, we simply set $A = 0.5$, $\omega = 30$, and $\phi = 0$ during the simulation. For other parameters, see Table 1.

### Spectral analysis of the CAN sequence generator

**Convert the CAN dynamics into a sequence generator.** In the CAN, the activity bump vector $U$ containing a finite population of N neurons evolves according to Eqs. (8)–(10), and can be interpreted as the distribution of current state estimate under idealistic setup (e.g., given spikes simulated from the network dynamics, Eq. (2)). Hence the CAN can be conceptualised as a continuous-time Markov process (e.g., a generator model) with specific constraints so that the state vector represents a valid probability distribution which evolves over time (Fig. 3c)[50]. This analogy allows us to perform spectral analysis of the evolution matrix (i.e., the generator) and derive the condition for generating distinct sequential dynamics.

To derive the generator model, we first vectorise the network dynamics in Eqs. (8)–(10) as follows:

$$\tau_u \dot{U} = -U + W \cdot f(U) - V \tag{40}$$

$$\tau_v \dot{V} = -V + m \cdot U. \tag{41}$$

Here, $U$, $r$, $V$ are now population vectors with each dimension representing the states of a place cell. Since the global inhibition (Eq. (9)) restricts the activity bump from spreading without changing the location of the bump, $f(U)$ can be treated as an identical mapping as $f(U) = IU$. Furthermore, since the adaptation bump V lags behind the activity bump U with a displacement $s$ (see Eq. (34) and Fig. 3d) but shares the same Gaussian profile as $U$, it can be treated as a shift version of $U$ with $V = CU$, where $C$ is a sub-diagonal matrix with zeros except the $s$-th upper diagonal with value of $\epsilon$. Hence, from a normative account, we can interpret $V$ as an exponential moving average of past state distributions for constraining the smoothness of the dynamics of $U$, which lead to constant sub-diagonal perturbation to the canonical evolution matrix (without feedback inhibition). The detailed value of $\epsilon$ depends on the adaptation strength $m$ and the time scale $\tau_v$, with larger adaptation strength/smaller time scale leading to larger values of $\epsilon$. But for simplicity, we consider a fixed value of $\epsilon$ below. With these assumptions, the evolution of the states can be simplified as:

$$\tau \dot{U} = (O - C)U, \tag{42}$$

with $O = W - I$. We will focus our analysis on the 1D ring track environment for simplicity, where the transition matrix $O$ is a circulant matrix.

**Spectral analysis of the perturbed transition matrix.** For notational simplicity, we define $O^{s,\epsilon} = O - C^{s,\epsilon}$ as the perturbed transition operator, where as above, $C^{s,\epsilon}$ represents the sub-diagonal matrix with constant $s$-th upper diagonal (of value $\epsilon$) and zero everywhere else. Our key observation is that both the original and perturbed transition matrices are circulant matrices, hence sharing the same set of eigenvectors, with perturbed eigenvalues[52,60]. Specifically, the (normalised) eigenvectors for the circulant transition matrix over a circular state space of $N$ states are the set of Fourier modes (Fig. 3e).

$$\vec{v}_k = \frac{1}{\sqrt{N}}(1, \omega_k, \omega_k^2, \ldots, \omega_k^{N-1}), \text{ where } \omega_k$$
$$= \exp\left(-\frac{2\pi i}{N}k\right), \text{ for } k = 0, \ldots, N-1, \tag{43}$$

And the corresponding eigenvalues are the discrete Fourier transforms for the first row of $O$.

$$\lambda_k^0 = \sum_{n=0}^{N-1} a_n \omega_k^n, \text{ for } k = 0, \ldots, N-1, \tag{44}$$

where $\{a_n\}_{n=0}^{N-1}$ is the set of elements that uniquely define the circulant matrix (first row or column of the matrix).

Following the above Fourier analysis, we can analytically compute the perturbation to individual eigenvalues following the $(s, \epsilon)$-perturbation to the transition operator.

$$\lambda_k^{s,\epsilon} = \sum_{n=0}^{N-1} (a_n + \epsilon \mathbb{1}_s(n)) \omega_k^n, \text{ for } k = 0, \ldots, N-1, \tag{45}$$

where $\mathbb{1}_s(n)$ is the indicator function that equals to 1 when $n = s$ and 0 otherwise. Hence the $(s, \epsilon)$-perturbation to the $k$-th eigenvalue is as following.

$$\Delta_k^{s,\epsilon} = \lambda_k^{s,\epsilon} - \lambda_k^0 = \epsilon \exp\left(-\frac{2\pi i \cdot s}{N}k\right), \tag{46}$$

As a reminder, the eigenvectors remain unchanged. Hence, the perturbation effects on the transition dynamics are fully characterised by perturbations in eigenvalues (Fig. 3f and Fig. S3). Increase/decrease in (absolute values of) eigenvalues lead to amplifying/damping effects in the spatial distribution modelled by the corresponding

eigenvectors. Hence, an increase in eigenvalues corresponding to eigenvectors of larger spatial scales lead to increased tail probabilities in the transition dynamics, leading to higher probability for sampling distant locations (super-diffusivity). The precise effect can be qualitatively illustrated via examining the ratio between absolute values of perturbed and original eigenvalues (Fig. S3a, b). Depending on the offset associated with the perturbation, there exists a spectrum of oscillatory patterns dependent on the perturbation offset. Here we take two extreme cases for demonstration and perform full analysis of a simplified case – "Lazy" random walk below. With a small offset ($s = 5$), eigenvalues of larger spatial scales (corresponding to lower-frequency Fourier modes) are dampened, and the ratios in absolute value with respect to original eigenvalues increase with the decrease in spatial scales, leading to a decrease in tail probabilities; hence, local diffusion is more likely. On the other hand, with a larger offset, an increase in eigenvalues of larger spatial scales precedes a decrease in eigenvalues of smaller spatial scales, amplifying transition probabilities associated with the distant states, hence it becomes more likely for super-diffusive behaviour to emerge. Intermediately small perturbation offsets (e.g., $s = 15$, Fig. S3b) lead to a reversed oscillatory pattern comparing to that of $s = 50$. In addition, we could analytically derive the precise boundary values for the offset at which the animal switches from local- to super-diffusion (see below), hence providing the ability to precisely track the nature and dynamics of replay sequences.

**A simple case of "Lazy"-random walk for further analysis of the eigenvalue rescaling problem.** For the simplicity of quantitative analysis, we consider a "Lazy"-random walk transition matrix (for analytical computation of the perturbed eigenvalues, Fig. S3c).

$$O^{\text{lazy}} = \begin{bmatrix} 0.5 & 0.25 & 0 & \cdots & 0.25 \\ 0.25 & 0.5 & 0.25 & \cdots & 0 \\ 0 & 0.25 & 0.5 & \cdots & 0 \\ \vdots & \vdots & \vdots & \vdots & \vdots \\ 0.25 & 0 & 0 & \cdots & 0.5 \end{bmatrix} \tag{47}$$

We could analytically compute the eigenvalues of $O^{\text{lazy}}$.

$$\lambda_k^{\text{lazy}} = 0.5 + 0.5\cos(i\theta_k), \text{ where } \theta_k = \frac{2k\pi}{N}, \tag{48}$$

Perturbations to the principal eigenvector (second principal due to the constant Fourier mode corresponding to $k = 0$) has the most dominant effect on the resulting dynamics, we hence focus our analysis on $\Delta_1^{s,\epsilon}$ for all $s$.

$$\lambda_1^{s,\epsilon} = 0.5 + (0.5 + \epsilon)\cos(\theta_1 s) - i\epsilon\sin(\theta_1 s), \text{ for } s = 0, \ldots, N-1, \tag{49}$$

Examining the set of perturbed eigenvalues corresponding to different offsets reveals that the transition from dampened to amplified principal eigenvalue happens as $s = 25$ (and $s = 75$ given the symmetry, Fig. S3d). We additionally verify this under the Gaussian diffusion policy studied in the main text, which conforms well with our theoretical analysis (Fig. S3d). Note that the precise boundary is dependent on a number of factors, including the value of $\epsilon$, which we have assumed to be constant in the current instance.

As discussed in Results, varying the perturbation offset leads to a spectrum of oscillatory patterns on the ratio between perturbed and original eigenvalues. To maximally demonstrate our intuition, we only studied two extreme cases in the main text. However, the complete range of oscillatory patterns are much more heterogeneous and harder to analyse (Fig. S3e), involving complicated interplay amongst $s$, $\epsilon$, and spatial scale of eigenvalues ($k$, Eq. (44)). The main intuition is that for intermediately small offsets (up to symmetry), the oscillatory

patterns are reversed comparing to larger offsets (Fig. S3b, e), i.e., decrease in eigenvalues of larger spatial scales lead increase in eigenvalues of smaller spatial scales, hence the resulting dynamics will more likely be local diffusion. We leave a more concrete analysis of such spectral behaviour for future studies.

Note that for simplicity, we assume a symmetric transition matrix (for both Gaussian diffusion and lazy random walk), but note that our analysis generalises to arbitrary asymmetric and circulant transition matrices (e.g., corresponding to non-trivial translation) so long as the circulant symmetry is preserved given the perturbation.

### Reporting summary

Further information on research design is available in the Nature Portfolio Reporting Summary linked to this article.

## Data availability

All experimental data are taken from the Collaborative Research in Computational Neuroscience (CRCNS) hc-6 dataset contributed by Loren Frank and colleagues[61]. They are publicly available at: https://crcns.org/data-sets/hc/hc-6. Source data are provided with this paper.

## Code availability

Code for reproducing all the results in the main text is available at https://doi.org/10.5281/zenodo.17488344.

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

## Acknowledgements

We thank Eric Denovellis for sharing the code of the state space decoder. We thank Loren Frank and colleagues for making the experimental data available online. We thank Kenneth Kay, Thomas Wills, Mattias Horan, Wentao Qiu, and Wenhao Zhang for valuable discussions. This work was supported by: a National Key Research and Development Program of China (2024YFF1206500, S.W.), a Wellcome Principal Research Fellowship (NB), a UKRI Frontier Research Grant (EP/X023060/1, D.B.), and an International Postdoctoral Exchange Fellowship Program (PC2021005, Z.J.).

## Author contributions

Z.J., T.C., X.D., N.B., and S.W. conceptualised and designed the research. Z.J. analysed the experimental data with the input from N.B. Z.J., T.C., X.D., and C.Y. performed theoretical analysis and simulations. D.B. and N.B. supervised the analysis of experimental data, and S.W. supervised the analysis of theoretical modelling. Z.J., N.B., and S.W. wrote the manuscript with the input from all the other authors.

## Competing interests

The authors declare no competing interests.
