## [Transparent Peer Review file · Nature Communications]

Dynamical modulation of hippocampal replay through firing rate adaptation

Corresponding Author: Professor Neil Burgess

Version 1:

Reviewer comments:

Reviewer #1

(Remarks to the Author)

Summary:

The authors study the phenomena of replays and theta sequences in the hippocampus, focusing in particular on the states of spatio-temporal dynamics that characterize them (stationary / diffusive / fragmented) - and on the conditions/constraints that need to be met to evoke such dynamical states. They propose a model based on a continuous attractor network, and combine the study of the system's dynamics with spectral theory to offer both a mechanistic and normative view of how and why replays might occur. Overall, the framework designed by the authors to answer these questions is convincing: the theory is developed rigorously in its mathematical aspects, and the validation on the empirical data is solid. Linking the features of theta sequences and diffusive replay appears to be a particularly strong prediction of the modeling in the context of the literature given that theta sequences and replay are often studied separately.

Some aspects of this work might be further developed as described below, although they are not of primary importance given the focus of the paper, and might still be explored in future studies. Furthermore, some claims of novelty need to be relaxed and better contextualized in the literature on this topic.

1. The authors do a good job in explaining which conditions create a favorable environment in their system for replays to appear. It is not sufficiently clear though from their study whether replays are then just a "side effect" of the network functioning (i.e. oscillatory noise due to the recurrent structure of the system) or if they have a computational purpose. Several studies suggest (and the authors also mention this in their introduction) that replays might be involved in memory consolidation, spatial planning and decision making. Nevertheless, in this model replays/theta sequences emerge even when the network is just designed to encode position, so it is not clear if they would play a role - and, if so, which role - during planning/inference or other types of computation.

2. In this model the trajectories of the network bump - and hence those of replays and theta sequences - are assumed to follow Brownian motion: as shown by the authors, this is a consequence of modeling the recurrent excitatory connections $J(x,x')$ with a translationally invariant connectivity map. Nevertheless, replays are known to feature specific spatio-temporal patterns that reflect both the structure of the environment and/or of the current task, which cannot be captured by brownian motion alone (e.g. Widloski and Foster, 2022). It would be interesting to discuss how to extend this mechanistic model to alternative connection schemes that would allow for more complex replay spatial patterns (while still preserving the bump stability and the results illustrated in 4.3-4.7).

3. The Stachenfeld 2017 study proposes that hippocampal place cells encode a predictive map of the environment based on the successor representation (SR). It would be interesting to know if the model presented by the authors can be reconciled with the findings in Stachenfeld 2017. The authors could speculate on whether the SR matrix could be embedded in the excitatory recurrent connections of the CAN model. Would the results shown in 4.3-4.6 still hold?

4. Authors should discuss how neural adaptation may serve as an extrinsically controllable network mechanism as is required by the normative / cognitive theory. Furthermore, it is claimed:

"While the mechanism that allows individual hippocampal networks to switch between

these replay dynamics during awake and sleep states is still lacking, our model suggests that such alternation could be achieved by simply adjusting the adaptation strength (see Discussion for more details)."

It has already been proposed in McNamee et al that cortical input via entorhinal cortex switches between different hippocampal replay regimes across the awake and sleep states. The authors, could critically evaluate/compare this proposal with their model in this respect.

5. The references linking neuromodulation to neural adaptation do not specifically investigate neuromodulation in hippocampus. In fact, Yoshida et al establish neuromodulation of adaptation in the entorhinal cortex (which is more consistent with the entorhinal generator model (McNamee et al)). The Madison & Nicoll study is the only one that studies hippocampus however in vitro and the causal manipulations in this investigation do not include neuromodulators. I think neuromodulation makes sense as a possible modifier of neural adaptation however I wonder if the authors could strengthen the case here since this is critical to model plausibility.

6. Seems this CAN model generates Levy walks rather than Levy flights in the superdiffusive regime. That is, when making a large step, a Levy walk has that the activity bump traverses the intervening positions between stabilising at the final position. In contrast, a Levy flight "jumps" to the final position without traversing the intervening positions. Is this correct? If so, it would be useful to elaborate this subtlety and examine this distinction with respect to empirical phenomena i.e. does the data look more like Levy walks or flights. Possibly, there is a separation of timescales to be considered here.

7. The text claims that more superdiffusive replay during the awake state and more diffusive replay during the sleep state has not been established in the literature:

"While these findings are based on different animals engaged in different tasks, it remains unclear whether such a difference holds in the same animal with different behavioural states, or it is simply due to the difference of the engaged tasks (the spatial memory task versus the random foraging task)."

Actually, this was established in McNamee et al 2021 - see Fig 6a there which re-analyzed Stella et al 2019 data in order to show, using the MSD measure, more superdiffusive replay in the awake state vs the sleep state. Here, Figure 6e, f is consistent with McNamee et als analysis of Stella2019. And establishes this effect in a new dataset (which is nice). Furthermore, Fig 6f here adds a paired analysis across behavioral states within animals (McNamee et als analysis collapsed data across animals).

Minor Comments:

Introduction

Might be useful to provide some definition/background/reference for firing rate adaptation in the hippocampus.

Might be useful to provide some context concerning the use and achievements of Attractor Networks to model the hippocampus

Figure 1

1.B,D Change time stamps into time interval (3889.5 - 3892.5 \Rightarrow 0 - 3) to improve readability

Figure 2

2.D "the animal stays at the bottom of the linear track..." isn't this a simulation? Saying "the animal" might be misleading.

Results 2.1

"Fire adaptation [...] a general feature of neural responses in the brain [hopfield 2010..]". It would be useful to have these references - and a brief background on adaptation - in the introduction.

Figure 3

3.B Why do simulated values differ from analytical?

Figure 5

5.B,C Change time stamps into time interval to improve readability

Figure 6d

Please label y-axes.

Figure 7

7.A,B why not use the same measures (firing rate over spike count) and same binning in model and experimental data for easier comparison?

7.E,H the firing rate peaks in experimental data seem to have a delayed phase with respect to the model prediction, why? Also, maybe it would improve readability to have the same color code for firing rate and step size in panels e and h.

Results 2.4.3

"...decreased activity might affect decoding accuracy and potentially inflate replay step size...". Design a control to assess the effect of this bias? What if we binarize spiking activity?

Methods 4.2.1

You are using an encoder/decoder that “assumes” that the neural trajectories are structured according to three distinct dynamical states (with fixed transition probabilities to move from one state to another). You are then using these same decoded trajectories to show that indeed their dynamics can be modeled with three dynamical states with different statistics. Isn't this circular?

Methods 4.5

There is a typo in the first sentence of the second paragraph:

“To simplify the analysis, we consider a noise free network with where σ and σ are all zero”

(Remarks on code availability)

The code repo seems very well-organized and interpretable, and contains clear instructions on how to run the code. The results of the paper appear to be reproducible from the code (with a clear demonstration of such in the ipython notebooks provided) although I did not run the code myself.

Reviewer #2

(Remarks to the Author)

In this manuscript, Ji et al study sequence generation in the hippocampus using modeling and data analysis. They analyze a neural network model based on CAN and adaptation using two different and convergent approaches. Then they derive three predictions from their model how different adaptation strengths affects correlations between different aspects of theta and replay sequences. Finally, they confirmed the three model predictions through analysis of experimental data collected previously by another group (Karlsson et al., 2015). Overall, the manuscript is very well written and presents a nice combination of modeling and experimental results. My main concern is about the novelty of the predictions and the lack of support for the central claim that the three key observations are the result of differences in adaptation.

Major concerns:

1. As the authors acknowledge, there has been much modeling and experimental work on sequences in the hippocampus. The central appeal of this manuscript is the integration of and match between the modeling and experimental parts, which are linked through three predictions. Unfortunately, the three predictions that the authors derive from their model are not novel and have been demonstrated in previous studies (which the authors have cited in their manuscript). The first prediction has been suggested by Azizi et al. (2013; Fig. 4) who showed that sequences propagate faster if adaptation is stronger. Given the same time interval for propagation, hence, sequences will be longer for stronger adaptation. The second prediction that sleep replay should be different from awake replay was already discussed by McNamee et al. (2021) who posited that awake replay should be super-diffusive whereas sleep replay should be diffusive. They match their model to data from Stella et al (2019), see Fig. 6a in McNamee et al. The final prediction that spiking probability and step size are anticorrelated matches results by Pfeiffer and Foster (2015; Fig. 3).

While it is correct that the three predictions follow from their model and hence can be considered “predictions” of their model, they are not predictions in the sense that they represent something new that we didn't know before.

2. The central claim is that the different observations are caused by variations in adaptation. This argument seems very intuitive and is supported by modelling results. However, on the one hand, it is not so clear that this is the only theoretical mechanism that can account for the three main observed differences. On the other hand, no experimental evidence is presented that there are difference in adaptation in the experimental conditions between which differences in sequence properties are observed. Therefore, there is a missing link between the three observed differences in sequential activity and adaptation.

3. Specifically, there is a hint that there is more, or something else, going on. For instance, while the correlation between theta sweep lengths with replay diffusivities are significantly smaller for data shuffled within animals than for the real data (section 2.4.1), the shuffled correlations are still significantly different from zero. Since none of the correlations for individual animals are significant (Fig. S4), the results suggest that there are differences between animals and not within animals. However, if differences in adaptation were the primary driver of differences in sequences, one would expect within-animals differences, too.

4. I didn't understand why the authors suggest that the $\eta > 0.5$ ($\eta = 0.69$) in their study could be explained by the fact that random exploration was more constraint in the W-track than in the open field that Stella et al used (end of section 2.4.2). Brownian motion in 1-d can be mathematically shown to lead to $\eta = 0.5$, which is the same as in 2-d.

Minor concerns:

Fig. 3a+b: The value of alpha seems to be clipped at 2, but this isn't made explicit anywhere.

p. 12: When discussing the correlation between place field size and theta sweep length, the authors should cite Parra-Barrero et al. (2021) who showed that both are governed by running speed.

p. 19 "However, future research highlights the need for..." How can you know what the future will show the need for?

p. 20 "First, we extracted periods when the animals' speed was < 4 cm/s, preceded by 60 s with no movement > 4 cm/s." I have a hard time making sense of this sentence. The phrase "no movement > 4 cm/s" means "speed < 4 cm/s". If that was indeed the condition, it's much more understandable to just say "we extracted periods when the animals' speed was < 4 cm/s for longer than 60s, but the first 60s were not included in the analysis" or something like that.

Fig. S9: In the caption, the reference should be to Stella et al (2019), not Steele and Morris (1999).

(Remarks on code availability)

There are instructions for running the code and the code seems to be complete. I have not tried running the code.

Reviewer #3

(Remarks to the Author)

This manuscript extends the group's 2024 Elife article where they explained hippocampal theta sequences and theta cycling with the continuous attractor network (CAN) model. In this manuscript, the authors studied the properties of the CAN model with a slow damping effect, which they called adaptation. The authors found that with different adaptation strengths and input noise strengths, the model can generate stationary, diffusive, or super-diffusive replay-like dynamic patterns. Based on this model, they made three predictions related to the fine features of hippocampal replay and verified those predictions with experimental data. Combined with their previous work on theta sequences, the authors aimed to provide a comprehensive theoretical framework that incorporates two to three major hippocampal compressed sequential coding across brain states. They can explain many interesting phenomena which were previously observed but less understood in terms of their cause. The study provides rigid derivation for the replay features based on the model parameters. The analytical derivation is elegant, and this is an important step to enhance our understanding of replay quantitatively.

The major issue here is that the concept of adaptation was not well-explained, which restricts the reader's understanding of these findings and makes the audience feel some observed phenomena may be caused by other technical effects. To enhance the impact of their work in the field, especially to audiences without strong modeling background, the authors need to provide more biological explanations of the model and adaptation.

Major concerns:

1. Adaptation is the key part in the model and requires a more detailed introduction or discussion. In the right hand side of equation 1, the restoring term is split into $-U$ and $-V$ which are assumed to have different time scales. I can imagine the fast restoring $-U$ might be related to refractory period, then what is the biological mechanism that supports the slow adaptation ($-V$)? Does the slow adaptation actually exist in the hippocampus? Wouldn't that lead to a more uniform firing rate after a long duration while in fact hippocampal cells' firing rates are highly skewed? Is being able to trigger replay the only purpose of having adaptation? The authors should address all these issues.
2. By making three predictions in the manuscript, the authors assume a hierarchy related to the adaptation stability, where the stability increases from the level of replay events, to across brain states, and then across days and animals. Because the authors predict the diffusive replay is correlated with longer theta sequences even within the animals, they assumed the adaptation strength should be more stable across these brain states than across days during a familiar task. I don't think everyone will expect this to happen, especially without an understanding of what is adaptation and what is the underlying biological mechanism. The authors should provide additional evidence and discussion to support their claims.
3. I don't think showing waking rest replays are more diffusive than sleep replay is a valid prediction and validation process, unless the authors can show direct or indirect evidence explaining why waking rest should have stronger adaptation strength compared with sleep. Many technical issues may also lead to this phenomenon, such as cells having different firing rates and spike amplitudes across states, and during waking rest, the recoding may suffer stronger from the EMG noise, which could impact the decoding. These issues will need to be addressed thoroughly.
4. My feeling is this model is an (abnormal) diffusion equation with damping. For a traditional diffusive process, we will observe a Gaussian distribution with increasing standard deviation over time. The reason why we observe a traveling front in this model seems to be because the center of this Gaussian distribution is strongly suppressed and what is left is the tail of the distribution. The fact that the bump traveling speed depends on adaptation strength is also consistent with this hypothesis, where the strong damping suppresses all the activities except for the extreme tail of the distribution. This leads to two issues: 1) the tail should be very small in terms of amplitude, the reason why we can see a prominent traveling bump is because of the normalization step in equation 8. Then this step needs to be rationalized. What does this mean, and does it actually happen in the hippocampus? 2) The diffusion process doesn't have a preferred direction, meaning we could observe two bumps travel to opposite directions simultaneously. The authors state "Without loss of generality, we assume that the intrinsic movement of the bump is from left to right on the linear track", this is true when the propagation has a preferred direction. In reality, replay does have preferred directions, but I cannot see why it should happen in the model based on equations 1-2. The authors should talk about the potential mechanism that could break the symmetry of the propagation. Also, hippocampal replay can start from the middle of the track rather than the end of the track. When authors run numerical simulations at the middle of the track, does the probability bump propagate in one direction rather than two?

5. The authors use a cluster less method to decode neural activity from combined spikes from areas CA1, CA2, and CA3. These 3 brain areas are known to behave differently across theta phases (CA1 versus CA3) and during sharp-wave ripples (CA2). Notably, CA2 can show stationarity during ripples. Additionally, decoded stationarity was shown for whole detected events during development in CA1 (Farooq and Dragoi, 2019 and Muessig et al., 2019, both should be referred to). An additional confound could be brought up by pyramidal spike-like activity that may not pass the criterion of a well clustered pyramidal neuronal activity, specifically spikes from putative inhibitory neurons (that escaped the waveshape criterion) and EMG activity (or other non-neuronal activity). These activities could increase diffusivity as well as stationarity in an uncontrollable fashion. The authors should cross-validate their biological claims using clustered CA1 putative pyramidal neurons from this or other datasets available to them.
6. The decoding error during animal exploration of the maze has a median of 9 cm for 1-2 m-long segments being decoded. This error is very large and will most likely impact the reliability and interpretation of the posterior probabilities. The authors compared this error with errors reported in 2 previous studies. However, one of those studies used a 10 m-long track and large bins and the other used immature place cells during development, both of which increase the decoding error. The reported decoding error in 1-3 m-long track of adult rats is usually less than half of that reported here. The large error here could be contributed by the cluster less decoding where 'noise' combined with genuine spiking activity of pyramidal (CA1) neurons. The authors should also compute the decoding error obtained from CA1 clustered putative pyramidal neurons from this or other available datasets (see comment #6).

Minor concerns

1. If I understand the model (equation 1) correctly, cells are arranged based on their place field in the environment. In that way, we have a one-to-one mapping between space and cells. Then the assumption that the recurrent connection strength diminishes with distance can be very interesting. In other words, we can say that cells with stronger functional connectivity are more likely to represent adjacent locations. Does it mean that by this arrangement the model indicates a very strong pre-configured spatial representation (preplay)?
2. If there is a one-to-one mapping, I would suggest changing the xlabel in Fig. 2b top panel into position. We care more about activity strength across spatial bins. Also, what are those off-diagonal high values in Fig. 2b bottom panel? Are they related to some boundary conditions?
3. I don't understand this statement in section 2.2.2 "This type of eigenvalue rescaling amplifies transition probabilities associated with distant states...". I assume the spectral analysis decomposes the time series into spectral components, and, in the absence of nonlinearity, Fourier modes are mutually independent, with their temporal dynamics illustrated by eigenvalues. Large (>1) eigenvalues indicate their strength will grow over time while small (<1) eigenvalues indicate they will be reduced. In that case, at large offset, some modes (large or small) will grow, and some will decay, those modes will change in their power/amplitude rather than become other modes. So, what does "amplifies transition probabilities associated with distant states" mean?
4. The cluster less decoding method (Denovellis et al., 2021) assumes three states (stationary, continuous, and fragmented), and the initial transition matrix is heavily concentrated on the diagonal. I don't know how this strong tendency of keeping the current state (inertia) will impact the result as the concepts of stationary, diffusive, or super-diffusive seem to overlap with these states in the cluster less decoding method. How about using a memoryless Bayesian decoding method with all the spiking events before clustering?
5. The discussion of the relationship between theta sequences and replay needs to be re-evaluated. Firstly, Drieu et al., 2018 showed a correlation between lack of theta sequences in a non-natural condition and a degraded replay, not a definitive causal relationship between them. Secondly, Farooq and Dragoi, 2019 also showed a correlation between the emergence of theta sequences and replay during development (should be referred to together with Drieu et al. 2018 and Muessig et al., 2019). Thirdly, Dragoi and Buzsaki, 2006 showed the temporally-compressed organization of place cells sequences during theta oscillations (theta sequences) and discussed their relationship to replay (should be cited). Figure 1a in the current study seems to be adapted after the Dragoi and Buzsaki study and this should also be acknowledged.
6. I think in Fig. 5c maybe a 2d joint distribution showing density/counts can better illustrate the relation between theta sequence length and replay step size. Original plots can be still included as marginal distributions.
7. Fig. 5e seems to be bad examples, the differences between animals are not obvious. Why in Fig. 5d rat 3 has an average sweep length >50 cm while in Fig. 5e the mean value is clearly <50 cm?
8. I don't agree with the statement at the last line of page 18 "awake replay might directly reflect the animal's movements." Even though waking rest replay has higher ratio than sleep replay, only less than 15% are replay of current trajectories by the traditional measure (with a 5% chance level). Especially in Pfeiffer and Foster, 2013, less than 1% of replays are goal-directed.
9. There is a typo in 4.3, " τ_u is the neuronal time constant, and τ_v is the adaptation time constant which is much larger than τ_v ". Last one should be τ_u .
10. Adding line numbers can make the reviewing process easier.

(Remarks on code availability)

Version 2:

Reviewer comments:

Reviewer #1

(Remarks to the Author)

Thanks to the authors for their extensive revisions and comments. I have a few remaining queries/comments which I hope

will be useful.

The authors performed a fresh set of analyses in order to provide evidence for adaptation mechanisms in awake behavior animals. This seems to me to be a significant contribution in this manuscript and I would suggest adding this figure and potentially even referencing it in the abstract.

However, the authors say "In addition, we have highlighted a very recent paper from the Foster lab (Mallory et al, Science, 2025) showed that firing rate adaptation is a key factor controlling hippocampal replay. This is a direct support of our firing rate adaptation model which we have added to the references. " As far as I can see, Mallory et al just model the predominance of forward replay post-stop as a neural adaptation/fatigue mechanism in a CAN model. They don't provide empirical evidence that that is how the brain is generating such replay dynamics which I interpret is what "direct support" means. Its the similar issue that was highlighted in the previous round of reviews from my pov. There are many CAN-type models invoking neural adaptation in neural replay. Here is another study along the same lines from the same authors: [https://www.cell.com/cell-reports/fulltext/S2211-1247\(25\)00246-3](https://www.cell.com/cell-reports/fulltext/S2211-1247(25)00246-3)

"Our focus is on illustrating how diverse replay dynamics observed in the hippocampus can be reconciled through a simple and effective neural mechanism of firing rate adaptation, rather than on whether controlling adaptation strength can directly induce a replay regime switch. These are two distinct questions."

I don't understand. It seems the model is that you get diverse replay dynamics if the adaptation strength is changed. Changing the adaptation strength gives you diversity so I don't see how "diverse replay dynamics" can be divorced from changing the adaptation strength. Maybe I'm misinterpreting the word "diverse" but I understand it to mean diffusive vs superdiffusive for example.

In the abstract, it is still claimed that this model predicts that "replay diffusivity varies within an animal across behavioural states" As discussed in the previous round of reviews, this is not a (novel) prediction, since it was already normatively predicted and evidenced in data in McNamee et al 2021 and also forms a key element of Krause & Drugowitsch 2022. The contribution here is to repeat such analyses on a per-animal basis rather than at the group-level. Furthermore, I don't see how it is a prediction at all in this ms. What is shown here is that it is possible for adaptation mechanisms to generate both diffusive and superdiffusive sequencing but doesn't give any reason why these might be associated with e.g. wake vs sleep. Why not superdiffusive during sleep? This seems beyond the scope of this study which focuses more on circuit mechanisms.

Regarding point 6 - the distinction between Levy flights and Levy walks. Thanks for clarifying, it makes sense to me. Interesting comment from Brad Pfeiffer, I think the control analyses in Pfeiffer & Foster should account for such concerns but no matter. In order to sharpen the predictive distinction between models, I suggest explicitly citing McNamee et al as predicting Levy flights in the superdiffusive regime (based on cortical input) in contrast to the Levy walks of the model considered here.

(Remarks on code availability)

Reviewer #2

(Remarks to the Author)

The authors have comprehensively and successfully addressed all concerns that I raised in the first round.

(Remarks on code availability)

There are instructions for running the code and the code seems to be complete. I have not tried running the code.

Reviewer #3

(Remarks to the Author)

In the revised manuscript, the authors have addressed most of our previous concerns. By incorporating patch-clamp data analysis and discussing potential neural mechanisms underlying hippocampal adaptation, the manuscript has improved in quality and now has stronger biological support. I do not have concerns requiring further analysis, but I have two minor questions that need clarification. I believe there may have been some misunderstanding, possibly due to a lack of clarity in my previous comments, so I will attempt to clarify them here.

1. Related to previous major concern 2: The authors support their model with the following arguments:

- 1.1 Their model predicts that adaptation strength is positively correlated with theta sweep extent during running.
- 1.2 Their model predicts that adaptation strength is positively correlated with replay diffusivity during waking rest.
- 1.3 Experimental data show that theta sweep extent is positively correlated with replay diffusivity.

For this argument to be valid, do we need to assume that the hippocampal circuit exhibits similar adaptation strength between running and waking rest states? The authors demonstrated that adaptation strength can differ between rest and sleep states. If adaptation strength also differs between waking rest and running, could that undermine their argument? My intuition is that a common factor invariant across both waking rest and running states (such as recording quality or place map length) may better explain the observed correlation between theta sequences and replay.

2. Related to previous major concern 4: The authors have addressed most of my questions, and I appreciate their patience. My remaining question pertains to Eqn 9 (global inhibition), particularly with the large k value used in their study. To me, this term looks like a normalization term that makes sure overall firing rates remain stable over time. While this may be a commonly used technique in CAN models, I would like further biological explanation on this. My question consists of two parts:

2.1 With a small k value, the firing rate of neurons would be more directly dependent on their own presynaptic inputs, which seems more biologically plausible to me. In this case, would the overall activity level of the circuit (or the amplitude of the traveling bump) experience rapid growth or decay over time?

2.2 If so, then a relatively large k value is crucial to reconcile the model's predictions with experimental data. In that case, what is the biological interpretation of this strong global inhibition (or alternatively, global excitation, if the overall activity tends to decay over time)? This equation implies that at each time step, a neuron's firing rate depends not only on its own presynaptic inputs but also on the presynaptic inputs of all neurons in the network. How could neurons instantaneously know presynaptic inputs of other neurons in the network? I would expect neuronal homeostasis to occur on a relatively longer timescale, making this instantaneous adjustment seem biologically implausible. A discussion or explanation of this mechanism would be valuable.

(Remarks on code availability)

Version 3:

Reviewer comments:

Reviewer #1

(Remarks to the Author)

Thanks for your work on this revision and clear descriptions of the updates.

Regarding the Mallory et al work, I also spoke with John on his recent European tour :) some take-aways:

I remain steadfast in my point that there is no "direct evidence" for the neural adaptation model in the context of the Mallory et al data. As you say "...it is not direct in the sense of experimentally manipulating adaptation and observing its effect on replay." I'm quite confident that "direct evidence", in general parlance, does indeed correspond to the experimental manipulation of the cause in order to test for the predicted effect. One could say that your model (which is essentially the same concept of John's in modeling retrospective replay suppression in the Mallory paper) is consistent with this data. However, it is also consistent with replay suppression via MEC input to HC as modeled here (Figure 2) albeit in a different scenario with an explicit cognitive function (<https://www.mdpi.com/1099-4300/24/12/1791>).

It is also notable that a major part of the Mallory work is to demonstrate the manipulation of hippocampal replay as a function of MEC input, thus I think it worth emphasizing how the Mallory work points to some form of cooperative interaction between HC-endogenous neural adaptation and MEC input as two distinct mechanisms controlling the timecourse of replay (i.e. adaptation suppresses retrospective replay initially then MEC input subsequently enhances it) e.g. towards the end of the second paragraph in your Discussion (fyi, an attempt at such a point was made here in Figure 2 <https://www.sciencedirect.com/science/article/pii/S0959438824000175> but not as eloquently elaborated as you have it in your Discussion points).

In the discussion, you have the line "This idea is supported by empirical data showing that MEC input controls the temporal organization of hippocampal activity [Schlesiger et al., 2015, Yamamoto and Tonegawa, 2017], but possibly not by Ormond and McNaughton [2015]."

Can you elaborate re the Ormond/McNaughton point? The point is that the Ormond/McNaughton data is inconsistent with the MEC theory? I always thought it quite consistent in the sense that the grid code in MEC constitutes a spectral decomposition. In the McNamee et al paper, there is some modeling where if you suppress high-frequency components (via tau modulation) then the place receptive fields in HC expand/spread.

Really comprehensive and interesting work, thank you.

(Remarks on code availability)

Reviewer #3

(Remarks to the Author)

In this improved revised manuscript, the authors have comprehensively addressed all my questions and I have no additional queries. I recommend this interesting study for publication.

(Remarks on code availability)

Dear Referees,

We extend our sincere gratitude to the reviewers for their detailed, thoughtful, and constructive feedback on our manuscript titled "Dynamical Modulation of Hippocampal Replay Sequences through Firing Rate Adaptation" (NCOMMS-24-50968A-Z). We deeply appreciate the time and effort devoted to reviewing our work, as well as the valuable insights provided, which have greatly contributed to improving the quality and clarity of the manuscript. In this letter, we first address common questions raised by all the reviewers before providing a point-by-point response to individual comments. For clarity, we have reproduced the reviewers' comments in green, followed by our responses in plain text. Where appropriate, we have also indicated specific revisions made in the manuscript. We hope that the revisions and explanations provided meet the reviewers' expectations and enhance the manuscript's contribution to the field.

Yours sincerely,

Dr. Zilong Ji & Prof Neil Burgess

UCL Institute of Cognitive Neuroscience & UCL Queen Square Institute of Neurology,
University College London, London, UK

General reply:

1: All referees raised concerns about the biological basis and potential control of firing rate adaptation (major point 4 and 5 from referee 1; major point 2 from referee 2; major point 2 from referee 3). Biologically, firing rate adaptation is a complex and multifaceted phenomenon that arises from multiple mechanisms, including:

- a) Intrinsic neuronal properties (e.g., afterhyperpolarization (AHP) currents, activation of voltage-dependent potassium currents (M-type potassium channels, and accumulation or depletion of intracellular or extracellular ion concentrations).
- b) Synaptic processes (e.g., short-term depression (depletion of neurotransmitter vesicles).
- c) Network interactions (e.g., recurrent inhibition from inhibitory circuits)
- d) Neuromodulatory influences (e.g., Acetylcholine (ACh), serotonin and dopamine)

Given the diversity of these mechanisms, the detailed form of firing rate adaptation can vary significantly depending on the specific biological context. For a comprehensive review, see Benda and Herz (2003). In our study, we utilized a mean-field model, which simplifies firing rate adaptation by focusing on its most critical feature: **slow negative feedback inhibition**. This abstraction is a necessary simplification to make the problem mathematically tractable while retaining the core dynamics of firing rate adaptation. However, we agree that the representation of firing rate adaptation in our work was deliberately simplified.

In response to the reviewers' concern about the existence of firing rate adaptation in behaving animals, we showed here the evidence of neural adaptation when rodents navigate in open fields (see the figure below). Note that firing rate adaptation has been extensively studied, but the majority of research was conducted using patch-clamp experiments on brain slices. If neural adaptation is present in neurons in behaving animals, one would expect more spikes as the animal is in the first half of the firing field of a cell compared to when it is in the second half of the field (panel c). To test this, we analyzed two separate datasets: one focusing on grid cells (1,330 cells in total, panel d) and the other on place cells (633 cells in total, panel e). Our findings provide robust evidence of neural adaptation, with particularly strong effects observed in place cells.

We agree that, due to the simplifications in our mean-field model with few parameters, it is challenging to directly attribute potential control to the adaptation strength parameter. However, the model effectively predicts and reproduces key phenomena found in empirical data. While we lack direct evidence for how this parameter changes between sleep and wake states (a limitation we have addressed in the revised paper), the model still serves as a valuable framework for understanding the underlying dynamics.

We have revised the manuscript, particularly the introduction, to clarify these points and better address the referees' concerns. Please see Lines 55-65 and 78-95.

(Unpublished data) Evidence of neural adaptation in grid cells and place cells in behaving animals. (a) Spike events (red dots) of an example grid cell with yellow circles representing grid fields. (b) The pdc values (see Jeewajee et al., 2014 for detailed computation) which is the distance to the field center projected on the current moving direction, with -1 representing entry to the field (red) and 1 representing leaving (blue). (c) The histogram of spike numbers of the example grid cell as a function of pdc. The total number of spikes of the example grid cell in the entry half and the leaving half are shown above the plot. The asymmetry (AS) index is calculated as the difference in the number of spikes between the entry half and the leaving half, normalized by the total number of spikes. Positive values mean more firing on the entry half and negative values mean more firing on the leaving half. (d) The histogram of the AS index for all grid cells (1330 in total). (e) The histogram of the AS index for all place cells (633 in total).

In addition, we have highlighted a very recent paper from the Foster lab (Mallory et al, Science, 2025) showed that firing rate adaptation is a key factor controlling hippocampal replay. This is a direct support of our firing rate adaptation model which we have added to the references.

2: As the referees pointed out, some of the predictions in our manuscript are more accurately described as theoretical explanations of previous empirical findings (referee 2, major point 1), findings already demonstrated

in prior work that we overlooked (referee 1, major point 7), or findings that may not be fully valid (referee 3, major point 3). In response, we have carefully revised our use of the terms “predictions” and “explanations” for greater clarity. Specifically:

- We retained the correlation between replay diffusivity and theta sequences as a **prediction**.
- We reframed the “anti-phase locking between step size and neural activity to slow-gamma phase” as a **theoretical explanation** of empirical observations, which were previously observed but less well understood.
- We described the “difference in replay diffusivity between awake and sleep states” as an **interesting finding**, which is consistent with differences that can occur within our simple model, but we do not have direct evidence that altered firing rate adaptation causes observed differences in replay diffusivity between awake and sleep states.

To reflect these changes, we removed the dedicated “predictions” section and reorganised the manuscript. The three points listed above are now presented as separate results sections, each starting with the modelling results followed by the corresponding empirical findings.

Reviewer #1 (Remarks to the Author):

Summary:

The authors study the phenomena of replays and theta sequences in the hippocampus, focusing in particular on the states of spatio-temporal dynamics that characterize them (stationary / diffusive / fragmented) - and on the conditions/constraints that need to be met to evoke such dynamical states. They propose a model based on a continuous attractor network, and combine the study of the system’s dynamics with spectral theory to offer both a mechanistic and normative view of how and why replays might occur. Overall, the framework designed by the authors to answer these questions is convincing: the theory is developed rigorously in its mathematical aspects, and the validation on the empirical data is solid. Linking the features of theta sequences and diffusive replay appears to be a particularly strong prediction of the modeling in the context of the literature given that theta sequences and replay are often studied separately.

Some aspects of this work might be further developed as described below, although they are not of primary importance given the focus of the paper, and might still be explored in future studies. Furthermore, some claims of novelty need to be relaxed and better contextualized in the literature on this topic.

1. The authors do a good job in explaining which conditions create a favorable environment in their system for replays to appear. It is not sufficiently clear though from their study whether replays are then just a “side effect”

of the network functioning (i.e. oscillatory noise due to the recurrent structure of the system) or if they have a computational purpose. Several studies suggest (and the authors also mention this in their introduction) that replays might be involved in memory consolidation, spatial planning and decision making. Nevertheless, in this model replays/theta sequences emerge even when the network is just designed to encode position, so it is not clear if they would play a role - and, if so, which role - during planning/inference or other types of computation.

Thank you for pointing this out. Our model does not directly address the eventual role of theta sweeps or replay, it provides a mechanism for their occurrence, which could be used in either a passive or an externally-directed way.

First, in this study, we only investigate replay diffusivity under a simple scenario where synaptic connections are translationally invariant. In this context, replays may appear as a "side effect" of non-local random drifting dynamics in the network. However, replays can also be directed by external inputs (e.g., via top-down modulation from reward cells) or shaped by synaptic connectivity. For instance, if the connection map is not translationally invariant but instead biased towards neurons encoding a goal or reward location, both theta sequences and replay sequences can be directed towards this goal or reward.

Second, from a broader perspective, we think that replays contribute to consolidation, via stabilizing the map learned during active behaviors through synaptic plasticity during rest, and/or by 'reading out' map structure to transfer this information to neocortex. As such it contributes to the information transformation between hippocampus and neocortex. The model is agnostic as to whether or not replay contributes to planning, and the literature is unclear, e.g. some experiments suggest that it might (Pfeiffer and Foster, 2013; Widloski and Foster, 2022) while others suggest that it might not (Frank replay at choice points paper).

2. In this model the trajectories of the network bump - and hence those of replays and theta sequences - are assumed to follow Brownian motion: as shown by the authors, this is a consequence of modeling the recurrent excitatory connections $J(x,x')$ with a translationally invariant connectivity map. Nevertheless, replays are known to feature specific spatio-temporal patterns that reflect both the structure of the environment and/or of the current task, which cannot be captured by brownian motion alone (e.g. Widloski and Foster, 2022). It would be interesting to discuss how to extend this mechanistic model to alternative connection schemes that would allow for more complex replay spatial patterns (while still preserving the bump stability and the results illustrated in 4.3-4.7).

This point relates to the second part of our response to comment 1 (see above) and to the discussion section where we briefly addressed the formation of synaptic connections. We agree that replays can exhibit spatio-temporal patterns that reflect both the structure of the environment and the requirements of the current task, such as goal-directed navigation. These patterns are likely shaped by synaptic modifications among neurons during active behaviours, potentially driven by long-term potentiation (LTP)-like synaptic plasticity. This process

may be facilitated by theta sequences which sequentially activate neurons with firing fields along short trajectories near the animal's current location, and do so within a timescale that affords plasticity. Under this condition, replays would deviate from simple Brownian motion. Instead, they could follow structured trajectories, such as navigating around barriers (Widloski and Foster, 2022) or tracing goal-directed paths (Pfeiffer and Foster, 2013). We have now expanded the discussion to address this point in greater detail (see Lines 379-390):

"...A potential mechanism for the development of attractor dynamics could involve synaptic modifications among neurons during active behaviour, possibly driven by synaptic plasticity mediated by theta sequences. Additionally, the learned synaptic configuration may not always exhibit translational invariance, depending on the structure of the environment (e.g., circular vs. square layouts) and/or the task design (e.g., random vs. goal-directed foraging). Such variations could lead to synaptic configurations that reflect the environmental or task-specific structure, potentially influencing the dynamics of theta and replay sequences to align with the underlying spatial or task context (Pfeiffer and Foster, 2013; Widloski and Foster 2022). Moreover, not only theta sequences but also offline replay sequences may contribute to synaptic modification between place cells, thereby aiding in the stabilization of attractor dynamics. In turn, these stabilized attractor dynamics generate useful sequential representations that support a variety of cognitive functions. They may also facilitate the consolidation of structural information from the hippocampus to other brain areas..."

3. The Stachenfeld 2017 study proposes that hippocampal place cells encode a predictive map of the environment based on the successor representation (SR). It would be interesting to know if the model presented by the authors can be reconciled with the findings in Stachenfeld 2017. The authors could speculate on whether the SR matrix could be embedded in the excitatory recurrent connections of the CAN model. Would the results shown in 4.3-4.6 still hold?

We thank the reviewer for pointing out the connection to the SR theory of hippocampal place cells. We note that the excitatory recurrent dynamics matrix is the one-step transition matrix under the prescribed behavioural policy (assumed to be the random walk policy for generality), and the SR matrix is, by definition, the infinite discounted sum of applications of such a transition matrix (and at the same time the stationary distribution corresponding to the Markov process defined by the transition matrix). Due to the fact that neural dynamics is computed given numerical integration of step-wise transitions within the neural space, the transition dynamics eventually converge to the transition structures specified by the corresponding SR matrix in the infinite limit. Hence, replacing the one-step transition matrix with the SR matrix in the excitatory recurrent connections should not change our theoretical analysis based on asymptotic conditions. Moreover, we wish to note that by definition, the transition matrix and the SR matrix have identical eigenvectors, and the eigenvalues are only different up to multiplicative scaling factors. Therefore, using either the transition matrix or the SR matrix would not change our current normative and mechanistic analysis.

One aspect we did not consider in our work is asymmetric synaptic connections. The asymmetric connections might correspond to an asymmetric SR matrix. For example, in the linear track environment when the animals

run from left to right, the one-step transition matrix is asymmetric, which leads to an asymmetric SR matrix. Embedding this asymmetric matrix into the excitatory recurrent connections of the CAN will lead to an activity bump moving ahead of the animal's position from left to right. This should be like the predictive coding aspect based on SR.

We have now discussed this in Line 390-394:

"...Additionally, Our model aligns with successor representation theory (Stachenfeld 2017) when the excitatory recurrent dynamics matrix is interpreted as the one-step transition matrix. By definition, the SR matrix is the infinite discounted sum of successive applications of this transition matrix. Consequently, replacing the one-step transition matrix with the SR matrix in the excitatory recurrent connections should not alter our theoretical analysis..."

4. Authors should discuss how neural adaptation may serve as an extrinsically controllable network mechanism as is required by the normative / cognitive theory. Furthermore, it is claimed:

"While the mechanism that allows individual hippocampal networks to switch between these replay dynamics during awake and sleep states is still lacking, our model suggests that such alternation could be achieved by simply adjusting the adaptation strength (see Discussion for more details)."

It has already been proposed in McNamee et al that cortical input via entorhinal cortex switches between different hippocampal replay regimes across the awake and sleep states. The authors, could critically evaluate/compare this proposal with their model in this respect.

Thank you for pointing this out. We first clarify that (1) the correlation between theta sweeps and replay diffusivity, and (2) the negative correlation between replay step size and neural activity do not require extrinsic control of firing rate adaptation. These two predictions are expected to hold in empirical data if firing rate adaptation is present. It is important to note that none of the previous experimental studies (e.g., Pfeiffer and Foster, 2015; Stella et al., 2019) have demonstrated evidence of a rapid switch between replay regimes within a single recording session. Our focus is on illustrating how diverse replay dynamics observed in the hippocampus can be reconciled through a simple and effective neural mechanism of firing rate adaptation, rather than on whether controlling adaptation strength can directly induce a replay regime switch. These are two distinct questions.

The only scenario where control of adaptation strength is relevant is in explaining transitions between replay regimes, specifically between awake and sleep states. As we discussed (Line 357), these transitions could be influenced by neuromodulation, such as Acetylcholine, which are known to modulate adaptation strength during awake and sleep states (Madison and Nicoll, 1984; Barkai and Hasselmo, 1994). However, direct evidence for differences in adaptation strength between these states is currently lacking (this might be similar to the case in McNamee et al justifying the rescaling of eigenvalues.) Referee 3 also raised this concern regarding extrinsic control of adaptation between awake and sleep states (Major Point 3).

In response to these concerns, we have reorganized the manuscript to present the "difference in replay diffusivity between awake and sleep states" as an interesting empirical finding. While this difference may not be directly explained by our model, it is consistent with the simple framework provided. This clarification is further elaborated under general replay point 2.

We hope these revisions address the reviewer's concerns.

5. The references linking neuromodulation to neural adaptation do not specifically investigate neuromodulation in hippocampus. In fact, Yoshida et al establish neuromodulation of adaptation in the entorhinal cortex (which is more consistent with the entorhinal generator model (McNamee et al)). The Madison & Nicoll study is the only one that studies hippocampus however in vitro and the causal manipulations in this investigation do not include neuromodulators. I think neuromodulation makes sense as a possible modifier of neural adaptation however I wonder if the authors could strengthen the case here since this is critical to model plausibility.

Thank you for raising this point. Despite an extensive literature search, we could not find studies explicitly investigating the causal link between neuromodulation and firing rate adaptation in the hippocampus. However, as highlighted in General Reply 1, we presented evidence of neural adaptation in both MEC and HPC cells, with adaptation being even stronger in HPC compared to MEC (as indicated by the asymmetry index, which is larger in HPC, see Figure panels d & e in the general reply).

While Yoshida et al. demonstrated neuromodulation of adaptation in the entorhinal cortex (EC), we hypothesise that similar mechanisms could extend to the hippocampus and other brain regions, given the observed stronger adaptation in HPC. To investigate this in greater depth, we are considering collaborating with teams specialising in patch-clamp recordings to explore the relationship between neuromodulation and firing rate adaptation more thoroughly.

6. Seems this CAN model generates Levy walks rather than Levy flights in the superdiffusive regime. That is, when making a large step, a Levy walk has that the activity bump traverses the intervening positions between stabilising at the final position. In contrast, a Levy flight "jumps" to the final position without traversing the intervening positions. Is this correct? If so, it would be useful to elaborate this subtlety and examine this distinction with respect to empirical phenomena i.e. does the data look more like Levy walks or flights. Possibly, there is a separation of timescales to be considered here.

Thank you for pointing this out. The CAN model indeed generates **Levy walks** rather than Levy flights (we apologise for overlooking this distinction in our original work). Since we assume the absence of external input when modelling replay-like dynamics, the activity bump moves continuously through the space rather than making instantaneous "jumps" to a new location.

In empirical data, distinguishing between Levy walks and Levy flights is challenging. We consulted Brad Pfeiffer for his opinion on this matter, and he also expressed uncertainty. This difficulty arises partly because current neural recordings are limited to hundreds of cells during navigation in open fields. This limited coverage can result in unevenly distributed place fields, introducing noise into the decoded locations and creating apparent "jumps."

If it were possible to record from many more hippocampal place cells and decode locations comprehensively, it would enable a more accurate determination of whether replay trajectories resemble Levy flights or Levy walks. For now, the distinction remains unclear due to these experimental limitations.

We have clarified this in the revised manuscript in Line 889-897:

"...This power law distribution corresponds to Levy walks where the activity bump traverses the intervening positions before stabilizing at the final position, rather than Levy flights where the activity bump "jumps" to the final position without traversing the intervening positions. This is because we assumed the absence of external input when modelling replay-like dynamics, the activity bump moves continuously through the space rather than making instantaneous "jumps" to a new location. In empirical data, distinguishing between Levy walks and Levy flights is challenging, with the difficulty arising partly because current neural recordings are limited to hundreds of cells during navigation in open fields. This limited coverage can result in unevenly distributed place fields, introducing noise into the decoded locations and creating apparent "jumps"...."

7. The text claims that more superdiffusive replay during the awake state and more diffusive replay during the sleep state has not been established in the literature:

"While these findings are based on different animals engaged in different tasks, it remains unclear whether such a difference holds in the same animal with different behavioural states, or it is simply due to the difference of the engaged tasks (the spatial memory task versus the random foraging task)."

Actually, this was established in McNamee et al 2021 - see Fig 6a there which re-analyzed Stella et al 2019 data in order to show, using the MSD measure, more superdiffusive replay in the awake state vs the sleep state. Here, Figure 6e, f is consistent with McNamee et al analysis of Stella2019. And establishes this effect in a new dataset (which is nice). Furthermore, Fig 6f here adds a paired analysis across behavioral states within animals (McNamee et al's analysis collapsed data across animals).

Thanks for pointing this out! We acknowledge overlooking this in the original work. We have now added McNamee et al. (2021) to the corresponding result section (Line 283) and included the clarification: *"...These findings align with results from McNamee et al. [2021], who used a separate dataset from Stella et al. [2019]. Unlike our study, their analysis merged data across animals rather than performing a pairwise test..."* It is encouraging to observe consistent results across different datasets, especially as the models are based on distinct assumptions.

Minor Comments:

Introduction

Might be useful to provide some definition/background/reference for firing rate adaptation in the hippocampus.

Updated. We added a background introduction for firing rate adaptation in **Line 55-65**:

"Recent studies have proposed that the generation of sequential activity may arise from single-cell firing rate adaptation, which penalises repeated activation of the same firing patterns and therefore causes different cells with lateral connections to fire sequentially (Romani and Tsodyks, 2015; Chu et al., 2024; Vollan et al., 2024). Firing rate adaptation is a widespread neurobiological phenomenon, exhibited by almost any type of neuron that generates action potentials, such as rodent motoneurons (Granit et al, 1963), hippocampal CA1 pyramidal cells (Lancaster and Nicoll, 1987), pyramidal cells of the piriform cortex (Barkai and Hasselmo, 1994), and in most pyramidal neurons in rodent neocortex (Connors and Gutnick, 1990). Biologically, firing rate adaptation is a complex and multifaceted phenomenon that arises from multiple mechanisms, including: intrinsic neuronal properties such as afterhyperpolarization (AHP) (Madison and Nicoll, 1984); synaptic processes such as short-term depression (Zucker, 1989); and neuromodulatory influences such as acetylcholine (Ach) (Liljenström and Hasselmo, 1993)"

Might be useful to provide some context concerning the use and achievements of Attractor Networks to model the hippocampus

Updated. We added some references about the used of attractor networks in modeling spatial tuning cells in **Line 67-70**:

"...we develop a theoretical framework conceptualizing the hippocampal place cell assembly as a continuous attractor network (CAN), which has been used extensively to model the firing features of spatial tuning cells in the entorhinal-hippocampal system (Mcnaughton et al, 2006), including head direction cells (Zhang, 1996), place cells (Tsodyks, 1999) and grid cells (Burak and Fiete, 2009; Ji et al., 2024)..."

Figure 1

1.B,D Change time stamps into time interval (3889.5 - 3892.5 \Rightarrow 0 - 3) to improve readability

Updated.

Figure 2

2.D "the animal stays at the bottom of the linear track..." isn't this a simulation? Saying "the animal" might be misleading.

Thank you for pointing this out. It is a simulation so we updated to "... The simulated rat stays at the bottom of the linear track..."

Results 2.1

"Fire adaptation [...] a general feature of neural responses in the brain [hopfield 2010..]". It would be useful to have these references - and a brief background on adaptation - in the introduction.

Thanks for pointing this out. We have updated the introduction (Line 55-65).

Figure 3

3.B Why do simulated values differ from analytical?

The differences between the simulated and analytical values in Figure 3b arise due to two main factors:

1. **Approximation in Theoretical Analysis:** During the derivation of Eqs. 25-28 from the original CANN dynamics, the right-hand side of the equations is not strictly linear with respect to s and z . Instead, it includes a factor $\exp\left(-\frac{s^2}{a^2}\right)$. Based on the assumption that s is much smaller than a , we approximated this factor with 1. This approximation simplifies the equations into the linear form of Eqs. 25-28. However, this also means that the dynamic equations used in the simulations (Eqs. 6-8) are not strictly equivalent to the theoretically derived Eqs. 25-28.
2. **Boundary Conditions:** In the theoretical analysis, z is assumed to range from negative infinity to positive infinity. However, since it is not feasible to simulate an infinite range of z , periodic boundary conditions were applied in the simulations. This also caused a mismatch between simulated results and analytical results.

Although these differences exist, they do not alter the main findings of the study from a qualitative perspective.

Figure 5

5.B,C Change time stamps into time interval to improve readability

Updated.

Figure 6d

Please label y-axes.

The y-axes are the same as Fig. 5b and the contents of y-axes were described in figure caption. Adding y-axes labels reduces the readability of the figure.

Figure 7

7.A,B why not use the same measures (firing rate over spike count) and same binning in model and experimental data for easier comparison?

Updated. We use the same measures (total number of spikes; as used in Pfeiffer and Foster (2015)) and increased the readability.

7.E,H the firing rate peaks in experimental data seem to have a delayed phase with respect to the model prediction, why? Also, maybe it would improve readability to have the same color code for firing rate and step size in panels e and h.

This might be simply due to noise and different decoding time bins used in neural data analysis (see SI fig 15-18 in Pfeiffer and Foster 2015). Color has been updated to improve readability.

Results 2.4.3

"...decreased activity might affect decoding accuracy and potentially inflate replay step size...". Design a control to assess the effect of this bias? What if we binarize spiking activity?

Thank you for pointing this out. Initially, we considered the possibility that decreased activity might contribute to inflated replay step sizes. However, upon re-evaluating this with the reviewer's suggestion in mind, we found no evidence to support this. As shown in Figure S9a, when the number of spikes participating in ripple events increases, replay diffusivity remains stable. If decreased activity were indeed affecting decoding accuracy and inflating replay step size, we would expect a negative relationship between replay diffusivity and the number of spikes in ripple events. However, such a relationship is not observed in our analysis.

We appreciate the reviewer's comment, which helped us to re-examine this assumption. We have now updated text as (Line 248-251):

"...One might expect that reduced activity, indicated by fewer spikes within the decoding window, could potentially affect decoding accuracy and inflate replay step size. However, we demonstrate that this is not the case, as replay diffusivity remains stable even as the number of spikes in ripple events increases (Fig. S9)..."

Methods 4.2.1

You are using an encoder/decoder that "assumes" that the neural trajectories are structured according to three distinct dynamical states (with fixed transition probabilities to move from one state to another). You are then

using these same decoded trajectories to show that indeed their dynamics can be modeled with three dynamical states with different statistics. Isn't this circular?

Thank you for raising this important point. We acknowledge that it would indeed constitute circular reasoning if the model explicitly imposed three latent dynamical states and we subsequently reported that the decoded trajectories strictly adhered to these states without independent validation. For instance, an extreme case would be if we imposed only stationary latent dynamics and concluded that hippocampal replay exclusively follows stationary dynamics.

However, our work does not make such a claim. We only illustrate three latent dynamics (stationary, continuous, fragmented) as examples Figure 1e to demonstrate the flexibility of the model. Alternatively, a simple Bayesian decoder can also have these three examples (as shown in Pfeiffer and Foster, 2015; Stella et al., 2019). Thus, the state-space model allows for, but does not enforce, these dynamics.

Crucially, in our analysis of neural data (Figures 5–7), we do not claim that replay dynamics must follow specific latent dynamics. Instead, these figures focus on comparisons:

- **Figure 5:** Replay diffusivity compared with theta sequence lengths.
- **Figure 6:** Diffusivity of awake replay compared to sleep replay.
- **Figure 7:** Detailed structure of jumping dynamics during replay against gamma oscillations.

The key advantage of using our state-space decoder lies in its ability to handle very small temporal bins, allowing for precise capture of rapid representational movement. Additionally, the decoder provides a robust statistical assessment of confidence for each latent dynamic, which is an improvement over classical methods. While we did not use this confidence information in the present work, it has been discussed extensively in the methods paper by Denovellis et al.

By employing a model that accommodates, rather than strictly enforces, the presence of these latent dynamics, we ensure that our analysis is guided by the structure naturally present in the neural data, rather than being limited by predefined model assumptions. Furthermore, we have tested the state-space decoder on open field and linear track data, as well as in both place cells and grid cells. It has demonstrated strong performance in decoding the animal's position during running periods (e.g., theta sequences), which gives us confidence in its ability to provide reliable decoding results during immobile states as well.

Methods 4.5

There is a typo in the first sentence of the second paragraph:

"To simplify the analysis, we consider a noise free network with where σ and σ are all zero"

Updated.

Reviewer #1 (Remarks on code availability):

The code repo seems very well-organized and interpretable, and contains clear instructions on how to run the code. The results of the paper appear to be reproducible from the code (with a clear demonstration of such in the ipython notebooks provided) although I did not run the code myself.

Reviewer #2 (Remarks to the Author):

In this manuscript, Ji et al study sequence generation in the hippocampus using modeling and data analysis. They analyze a neural network model based on CAN and adaptation using two different and convergent approaches. Then they derive three predictions from their model how different adaptation strengths affects correlations between different aspects of theta and replay sequences. Finally, they confirmed the three model predictions through analysis of experimental data collected previously by another group (Karlsson et al., 2015). Overall, the manuscript is very well written and presents a nice combination of modeling and experimental results. My main concern is about the novelty of the predictions and the lack of support for the central claim that the three key observations are the result of differences in adaptation.

Major concerns:

1. As the authors acknowledge, there has been much modeling and experimental work on sequences in the hippocampus. The central appeal of this manuscript is the integration of and match between the modeling and experimental parts, which are linked through three predictions. Unfortunately, the three predictions that the authors derive from their model are not novel and have been demonstrated in previous studies (which the authors have cited in their manuscript). The first prediction has been suggested by Azizi et al. (2013; Fig. 4) who showed that sequences propagate faster if adaptation is stronger. Given the same time interval for propagation, hence, sequences will be longer for stronger adaptation. The second prediction that sleep replay should be different from awake replay was already discussed by McNamee et al. (2021) who posited that awake replay should be super-diffusive whereas sleep replay should be diffusive. They match their model to data from Stella et al (2019), see Fig. 6a in McNamee et al. The final prediction that spiking probability and step size are anticorrelated matches results by Pfeiffer and Foster (2015; Fig. 3).

While it is correct that the three predictions follow from their model and hence can be considered “predictions” of their model, they are not predictions in the sense that they represent something new that we didn’t know before.

Thank you for your valuable feedback. We agree with the reviewer that some of the “predictions” in our manuscript are more accurately described as theoretical explanations of empirical data. To address this, we have carefully revised our use of the terms “predictions” and “explanations” for greater clarity. Specifically:

- We retained the correlation between replay diffusivity and theta sequences as a **prediction**.
- We reframed the “anti-phase locking between step size and neural activity to slow-gamma phase” as a **theoretical explanation** of empirical observations, which was previously observed but less understood.
- We described the “difference in replay diffusivity between awake and sleep states” as an **interesting finding**, which may not be directly explained by our model but is consistent with the simple framework it provides.

We added the citation of Azizi et al when we discussed about the bump propagation speed with respect to adaptation strength at:

Line 118: *“...This negative feedback destabilizes the bump state, causing the activity bump to move at a speed that increases with adaptation strength, when the strength is above a threshold (see Fig. 2c, Azizi et al. [2013], Mi et al. [2014] and Methods for details)...”*

We also acknowledged **McNamee et al. (2021)** for their empirical findings on replay diffusivity differences between awake and sleep states using data from the Csicsvari lab, which we had previously overlooked. The main difference between our analysis and McNamee’s approach is that, while McNamee et al. collapsed data across animals (as the first reviewer noted), we conducted a **pairwise analysis** between running sessions and subsequent sleep sessions for individual animals:

Line 283: *“...These findings align with results from McNamee et al. [2021], who used a separate dataset from Stella et al. [2019]. Unlike our study, their analysis merged data across animals rather than performing a pairwise test...”*

Finally, we appreciate the reviewer’s emphasis on the integration and alignment between the modelling and experimental components of our work. Additionally, we wish to underscore another key contribution: our detailed analysis of network dynamics from two different and convergent approaches, which offers analytical insights into the mechanisms driving diverse hippocampal sequential dynamics and serves as a bridge between theoretical and biological neuroscience.

2. The central claim is that the different observations are caused by variations in adaptation. This argument

seems very intuitive and is supported by modelling results. However, on the one hand, it is not so clear that this is the only theoretical mechanism that can account for the three main observed differences. On the other hand, no experimental evidence is presented that there are difference in adaptation in the experimental conditions between which differences in sequence properties are observed. Therefore, there is a missing link between the three observed differences in sequential activity and adaptation.

Adaptation is a well-established and ubiquitous phenomenon in neural firing, and we found evidence of firing rate adaptation in both grid cells and place cells in behaving animals (see the general reply). In our model, firing rate adaptation serves as a simplified representation of more detailed biophysical processes (such as spike frequency adaptation, short-term depression, and recurrent inhibition) ignored in the current work. The key feature of adaptation we focused on is the slow negative feedback dynamics inherent in these mechanisms. Demonstrating changes in adaptation would likely require in vivo patch-clamp recordings during different behavioural states, which is beyond the scope of this study. We hope future research will address this question further.

Below, we clarify the role of adaptation and whether it requires dynamical changes in strength to explain the three key empirical observations.

- 1. Replay and Theta Sequences**

Adaptation has been proposed as a mechanism for preplay (e.g., Azizi et al., 2013) and is likely involved in generating intrinsic theta sequences in the entorhinal-hippocampal regions (see the Discussion in Vollan et al., 2024, from the Moser lab, and our recent model in Chu et al., 2024). Given this background, it is intuitive to predict that adaptation underlies the correlation observed between these two types of sequential dynamics.

- 2. Replay Step Size and Neural Activity**

This phenomenon does not depend on changes in adaptation strength. Instead, it arises from the intrinsic interaction between extrinsic oscillatory input and intrinsic adaptation process, as explained in the manuscript.

- 3. Awake vs. Sleep Replay**

We acknowledge that our study does not provide direct evidence for changes in adaptation strength in the hippocampus of the same animal across awake and sleep states. To address this limitation, we have revised the corresponding sections of the manuscript to make this point clearer.

We appreciate the reviewer's insightful comments, which have helped us refine our manuscript. The revisions now clarify the role of adaptation in these phenomena and address the limitations in our approach.

3. Specifically, there is a hint that there is more, or something else, going on. For instance, while the correlation between theta sweep lengths with replay diffusivities are significantly smaller for data shuffled within animals than for the real data (section 2.4.1), the shuffled correlations are still significantly different from zero. Since none of the correlations for individual animals are significant (Fig. S4), the results suggest that there are

differences between animals and not within animals. However, if differences in adaptation were the primary driver of differences in sequences, one would expect within-animals differences, too.

Thank you for pointing this out. We would like to clarify that when the correlation value is significantly higher for data shuffled within animals (Fig. S5), this suggests a contribution of within-animal correlations. The non-significant correlations between replay diffusivity and theta sweep length within individual animals (Fig. S4) are likely due to the limited number of data points available for correlation analysis. For example, Rat 4 and Rat 9 each have only two recording sessions, while Rats 1, 2, and 3 have more sessions but still fewer than ten. This would limit the power of statistics performed here.

To further investigate within-animal correlations, we performed an additional analysis by dividing the recording sessions into two groups: one with short theta sequences and the other with long theta sequences. We then compared replay diffusivity between these two groups. Our analysis showed that replay diffusivity was significantly smaller during sessions with short theta sequences compared to sessions with long theta sequences across animals (as shown in Fig. 4f where points with different colors represent different animals):

We hope this additional analysis addresses the reviewer's concern.

4. I didn't understand why the authors suggest that the $\eta > 0.5$ ($\eta = 0.69$) in their study could be explained by the fact that random exploration was more constrained in the W-track than in the open field that Stella et al used (end of section 2.4.2). Brownian motion in 1-d can be mathematically shown to lead to $\eta = 0.5$, which is the same as in 2-d.

Thank you for pointing this out. The reviewer is correct that 1D Brownian motion should result in $\eta = 0.5$. In the W-track dataset, the sleep replay diffusivity is significantly greater than 0.5. To address this, we now state, "**Awake replay is more diffusive than subsequent sleep replay**," instead of emphasizing that sleep replay follows Brownian motion dynamics, as described in Stella et al. (2019). We have highlighted this clarification in the revised manuscript as follows (Line 285):

"...Notably, the diffusivity values of sleep replay in our study were significantly greater than 0.5 (diffusivity with

0.69 ± 0.12 ; Wilcoxon signed-rank test with $p < 0.001$), indicating a deviation from the Brownian-diffusive dynamics reported previously [Stella et al., 2019] (Fig. S9)..."

Minor concerns:

Fig. 3a+b: The value of alpha seems to be clipped at 2, but this isn't made explicit anywhere.

Thank you for pointing this out. The reason it was clipped at 2 is for better visualization. We now added the explanation in figure caption in Fig. 3:

"...Note that all $\alpha > 2$ were clipped to 2 for better visualization since there is no heavy tail in the distribution when $\alpha > 2$ and the sums of the step sizes converge to a Gaussian distribution due to the Central Limit Theorem, and thus always displaying Brownian motion..."

p. 12: When discussing the correlation between place field size and theta sweep length, the authors should cite Parra-Barrero et al. (2021) who showed that both are governed by running speed.

Updated. See Line 223:

"...Our analysis showed a marginal correlation between place field size and theta sweep length (Pearson correlation $r = 0.26$, $p = 0.055$; Fig. S7), indicating that larger place fields are associated with longer theta sequences [Parra-Barrero et al., 2021]..."

p. 19 "However, future research highlights the need for..." How can you know what the future will show the need for?

Thank you for pointing this out. We updated to (Line 358):

"...Future studies could benefit from more direct measurements of firing rate adaptation during different SWR-related awake and sleep states to experimentally validate this hypothesis. Additionally, manipulating adaptation strength in vivo could help determine its effects on replay dynamics and their behavioural correlates..."

p. 20 "First, we extracted periods when the animals' speed was < 4 cm/s, preceded by 60 s with no movement > 4 cm/s." I have a hard time making sense of this sentence. The phrase "no movement > 4 cm/s" means "speed < 4 cm/s". If that was indeed the condition, it's much more understandable to just say "we extracted periods when the animals' speed was < 4 cm/s for longer than 60s, but the first 60s were not included in the analysis" or something like that.

Thank you for the suggestion. We updated to (Line 630):

"...First, we extracted periods when the animals' speed was < 4 cm/s for longer than 60 s..."

Fig. S9: In the caption, the reference should be to Stella et al (2019), not Steele and Morris (1999).

Updated.

Reviewer #2 (Remarks on code availability):

There are instructions for running the code and the code seems to be complete. I have not tried running the code.

Reviewer #3 (Remarks to the Author):

This manuscript extends the group's 2024 Elife article where they explained hippocampal theta sequences and theta cycling with the continuous attractor network (CAN) model. In this manuscript, the authors studied the properties of the CAN model with a slow damping effect, which they called adaptation. The authors found that with different adaptation strengths and input noise strengths, the model can generate stationary, diffusive, or super-diffusive replay-like dynamic patterns. Based on this model, they made three predictions related to the fine features of hippocampal replay and verified those predictions with experimental data. Combined with their previous work on theta sequences, the authors aimed to provide a comprehensive theoretical framework that incorporates two to three major hippocampal compressed sequential coding across brain states. They can explain many interesting phenomena which were previously observed but less understood in terms of their cause. The study provides rigid derivation for the replay features based on the model parameters. The analytical derivation is elegant, and this is an important step to enhance our understanding of replay quantitatively.

The major issue here is that the concept of adaptation was not well-explained, which restricts the reader's understanding of these findings and makes the audience feel some observed phenomena may be caused by other technical effects. To enhance the impact of their work in the field, especially to audiences without strong modeling background, the authors need to provide more biological explanations of the model and adaptation.

Thank you for the suggestion. In response, we have added a paragraph in the introduction to provide more background on firing rate adaptation, as also recommended by Reviewer 1 (Line 55-65):

"Recent studies have proposed that the generation of sequential activity may arise from single-cell firing rate adaptation, which penalises repeated activation of the same firing patterns and therefore causes different cells with lateral connections to fire sequentially (Romani and Tsodyks, 2015; Chu et al., 2024; Vollan et al., 2024). Firing rate adaptation is a widespread

neurobiological phenomenon, exhibited by almost any type of neuron that generates action potentials, such as in rodent motoneurons (Granit et al, 1963), hippocampal CA1 pyramidal cells (Lancaster and Nicoll, 1987), pyramidal cells of the piriform cortex (Barkai and Hasselmo, 1994), and in most pyramidal neurons in rodent neocortex (Connors and Gutnick, 1990). Biologically, firing rate adaptation is a complex and multifaceted phenomenon that arises from multiple mechanisms, including: intrinsic neuronal properties such as afterhyperpolarization (AHP) (Madison and Nicoll, 1984); synaptic processes such as short-term depression (Zucker, 1989); and neuromodulatory influences such as acetylcholine (Ach) (Liljenström and Hasselmo, 1993)”

To provide a stronger biological explanation of firing rate adaptation and its connection to our model, we also have included patch-clamp data analysis (thanks to Michael Hasselmo and Motoharu Yoshida for kindly sharing their data with us). This addition helps illustrate the biological basis of firing rate adaptation and how it can be modelled as a slow negative feedback process in our framework (see Fig. S1 which we attached below).

Specifically:

- (a) illustrates the slow negative feedback modulation of firing rate adaptation in a single neuron within the model (upper panel) and shows how cell firing frequency changes over time when a constant input current is applied, with variations under different adaptation strengths.
- (b) presents experimental data from a real cell with weak firing rate adaptation, where the spike train (blue) of action potentials was recorded during a 1-second current pulse injection (orange) in an in-vitro whole-cell patch-clamp experiment from Yoshida et al., 2013.
- (c) shows experimental data from a real cell exhibiting strong firing rate adaptation.

Finally, we also included a paragraph in Discussion (Line 363-373) about the functions of firing rate adaptation (see the reply to Major concern 1).

We hope this comparison provides a clearer understanding of how firing rate adaptation operates biologically and how it is captured in our model.

Major concerns:

1. Adaptation is the key part in the model and requires a more detailed introduction or discussion. In the right hand side of equation 1, the restoring term is split into $-U$ and $-V$ which are assumed to have different time scales. I can imagine the fast restoring $-U$ might be related to refractory period, then what is the biological mechanism that supports the slow adaptation ($-V$)? Does the slow adaptation actually exist in the hippocampus? Wouldn't that lead to a more uniform firing rate after a long duration while in fact hippocampal cells' firing rates are highly skewed? Is being able to trigger replay the only purpose of having adaptation? The authors should address all these issues.

Thank you for pointing this out. We address these questions point-by-point below:

1: "what is the biological mechanism that supports the slow adaptation ($-V$)?":

Firing rate adaptation (whether reflecting e.g., spike frequency adaptation or short-term depression) is considered as a slow dynamic process because it involves mechanisms that operate on timescales longer than the action potentials themselves. For instance, spike frequency adaptation is mediated by ion channels that open or close more slowly than the voltage-gated sodium and potassium channels responsible for the rapid upstroke and repolarization of action potentials. Examples include calcium-activated potassium channels and slow voltage-gated potassium channels. Additionally, calcium accumulation and subsequent clearance (via pumps or buffers) occur on much slower timescales compared to the millisecond duration of individual spikes, further contributing to the slow nature of adaptation. Similarly, short-term depression, another form of firing rate adaptation, is caused by the depletion of neurotransmitter vesicles at the presynaptic terminal during high-frequency activity. The replenishment of vesicle pools depends on processes such as vesicle docking, priming, and mobilization from reserve pools, which typically take tens to hundreds of milliseconds or longer.

We have added detailed discussion about this point in Methods (Line 751-762).

2: "Does the slow adaptation actually exist in the hippocampus?"

Numerous studies on hippocampal brain slices have demonstrated firing rate adaptation, particularly in pyramidal cells, which are well-distributed in the pyramidal layer of the hippocampus, making them accessible for recording (see example studies by Pedarzani and Storm (1993), Gu et al. (2007), and Ruth et al. (2014) etc).

Additionally, we provided evidence of adaptation in both place cells and grid cells in behaving animals. Specifically, we observed asymmetrical firing, where cells fired more spikes as the animal traversed the first half of the firing field compared to the second half (see the general reply for more details).

"Wouldn't that lead to a more uniform firing rate after a long duration while in fact hippocampal cells' firing rates are highly skewed?"

In the current model, adaptation does not result in a more uniform firing rate, either during active running or immobile periods (see Fig. 4a). This is because the continuous attractor model incorporates global inhibition

and symmetric synaptic connections, ensuring that the activity bump consistently maintains a Gaussian-like profile which is localized and symmetric.

The observed skewed firing is more likely due to long-term potentiation (LTP), which occurs when animals repeatedly traverse a linear track from one end to the other over multiple trials. Experimental evidence supporting this comes from Ekstrom et al. (2001), who demonstrated that applying the NMDA receptor antagonist CPP (which prevents LTP) blocks the experience-dependent skewness. This suggests that the skewness arises from mechanisms distinct from firing rate adaptation.

“Is being able to trigger replay the only purpose of having adaptation?”

This is a good question and we have added this discussion into the paper (Line 363):

“Firing rate adaptation in biological systems is hypothesised to serve important roles beyond simply triggering replay, potentially contributing to several cognitive and computational processes. For instance, it has been suggested to facilitate mental exploration (Hopfield, 2010), support theta sequences during active running in rodents (see our recent eLife paper, Chu et al., 2024), and enable efficient sampling-based Bayesian inference of sensory information (Dong et al., 2022). Additionally, it may act as a high-pass filter, separating transient signals from slower oscillatory signals to improve signal processing (Benda et al., 2005).

These potential functions suggest that firing rate adaptation might not merely be a by-product of intrinsic dynamics but instead could play an important role in supporting specific neural computations and energy-efficient coding strategies. While firing rate adaptation is ubiquitous in neurons, possibly as an evolutionary mechanism to conserve energy and prevent over-firing, its hypothesised functional roles remain to be directly validated experimentally. Furthermore, whether cognitive processes exert control over firing rate adaptation is an open question that requires further exploration.”

We hope these explanations provided meet the reviewers' expectations.

2. By making three predictions in the manuscript, the authors assume a hierarchy related to the adaptation stability, where the stability increases from the level of replay events, to across brain states, and then across days and animals. Because the authors predict the diffusive replay is correlated with longer theta sequences even within the animals, they assumed the adaptation strength should be more stable across these brain states than across days during a familiar task. I don't think everyone will expect this to happen, especially without an understanding of what is adaptation and what is the underlying biological mechanism. The authors should provide additional evidence and discussion to support their claims.

Thank you for raising this point. We believe there may have been a misunderstanding regarding the role of adaptation in our model. To clarify, we did not assume a hierarchy related to the stability of adaptation.

First, if adaptation strength varies both across recording sessions (within the same animal) and across animals, we would then observe a correlation between replay diffusivity and theta sequence length because the model predicts that both features co-vary with adaptation strength. Second, if adaptation strength becomes weaker during subsequent sleep compared to the preceding running session, we would observe a difference in replay diffusivity between awake and sleep states.

However, these two phenomena (correlation between theta and replay within the same animal, and differences of replay diffusivity across brain states) do not imply that one requires greater stability of firing rate adaptation than the other.

Additionally, the third phenomenon, i.e., negative correlation between neural activity and replay diffusivity (now framed as a theoretical explanation for prior empirical findings instead of a prediction) does not rely on the so-called "stability" of firing rate adaptation. Instead, it happens because while oscillations increase the neural activity, it will naturally reduce the effect of adaptation, and hence lead to shorter step size and smaller replay diffusivity.

Previously, we aimed to provide a cohesive narrative by discussing firing rate adaptation from multiple perspectives and levels of granularity. However, as Reviewer 2 pointed out, not all of these aspects represent true predictions, nor is adaptation necessarily the sole mechanism underlying the observed phenomena (like your next comment about sleep and awake replay). While we agree with this observation, we believe that the proposed model offers a simple and elegant framework for understanding many empirical observations, both from previous literature and from our own analysis of empirical data.

In response, we have carefully reorganized the manuscript to make these distinctions clearer. We hope these revisions address the reviewer's concerns and improve the clarity and precision of our claims.

3. I don't think showing waking rest replays are more diffusive than sleep replay is a valid prediction and validation process, unless the authors can show direct or indirect evidence explaining why waking rest should have stronger adaptation strength compared with sleep. Many technical issues may also lead to this phenomenon, such as cells having different firing rates and spike amplitudes across states, and during waking rest, the recoding may suffer stronger from the EMG noise, which could impact the decoding. These issues will need to be addressed thoroughly.

Thank you for pointing this out. We agree with the reviewer that direct evidence of dynamic changes in firing rate adaptation during awake and subsequent sleep states is currently lacking. In response, we have reorganized the paper as follows:

- Retaining the correlation between replay diffusivity and theta sequences as a **prediction**.
- Reframing the "anti-phase locking between step size and neural activity to slow-gamma phase" as a **theoretical explanation** of empirical observations. This phenomenon was previously observed but remained less understood from a mechanistic view.

- Describing the “difference in replay diffusivity between awake and sleep states” as **an interesting finding**. While this may not be directly explained by our model, it is consistent with the simple framework the model provides.

We believe it is valuable to retain the empirical results on awake and sleep replay after rephrasing them (Line 267-288), as they represent a meaningful finding despite the need for further thorough checks to address potential technical issues. Another reason to report these results is the supporting evidence from prior studies, which have independently demonstrated super-diffusive awake replay (Pfeiffer and Foster, 2015) and Brownian-diffusive sleep replay (Stella et al., 2019). Additionally, related discussions on the differences between awake and sleep replay can be found in follow-up studies by Krause and Drugowitsch (2022) and McNamee et al. (2021), which used different datasets to explore similar distinctions. Importantly, our study performs a pairwise comparison of awake and subsequent sleep replay within individual animals, an approach that has not been explored in previous research.

4. My feeling is this model is an (abnormal) diffusion equation with damping. For a traditional diffusive process, we will observe a Gaussian distribution with increasing standard deviation over time. The reason why we observe a traveling front in this model seems to be because the center of this Gaussian distribution is strongly suppressed and what is left is the tail of the distribution. The fact that the bump traveling speed depends on adaptation strength is also consistent with this hypothesis, where the strong damping suppresses all the activities except for the extreme tail of the distribution. This leads to two issues: 1) the tail should be very small in terms of amplitude, the reason why we can see a prominent traveling bump is because of the normalization step in equation 8. Then this step needs to be rationalized. What does this mean, and does it actually happen in the hippocampus? 2) The diffusion process doesn't have a preferred direction, meaning we could observe two bumps travel to opposite directions simultaneously. The authors state “Without loss of generality, we assume that the intrinsic movement of the bump is from left to right on the linear track”, this is true when the propagation has a preferred direction. In reality, replay does have preferred directions, but I cannot see why it should happen in the model based on equations 1-2. The authors should talk about the potential mechanism that could break the symmetry of the propagation. Also, hippocampal replay can start from the middle of the track rather than the end of the track. When authors run numerical simulations at the middle of the track, does the probability bump propagate in one direction rather than two?

Thank you for raising these concerns. We understand the reviewer's points and will address them step by step. Before doing so, it is important to clarify the following:

In our study, **super-diffusive dynamics in the network** refers to the activity bump travelling distances that follow a heavy-tailed distribution. Importantly, the “bump” we discuss throughout the paper represents the shape of the population activity of sorted place cells (based on their firing fields in the environment), not the distribution of distances that the trajectory moved. To measure the distance travelled by the bump, we used the **center of mass of the bump** (i.e., the location of the bump center assuming the bump shape remains the same). This is analogous to using the corresponding location of the maximum posterior probability as the decoded location when analyzing replay in empirical data. This distinction is critical before addressing the reviewer's specific concerns.

1. “For a traditional diffusive process, we will observe a Gaussian distribution with increasing standard deviation over time...”

This is correct if the reviewer is referring to the distribution of the bump center in a diffusive process, which would indeed follow a Gaussian distribution with increasing standard deviation over time.

2. “The reason why we observe a traveling front in this model seems to be because the center of this Gaussian distribution is strongly suppressed and what is left is the tail of the distribution...”

This is not accurate. If the center of the Gaussian distribution were strongly suppressed, the bump would only exhibit large jumps without transferring to nearby locations. This would imply replay sequences that are entirely jumpy, lacking sequential dynamics, which then are not quantified as replay.

To provide an intuitive explanation: In the model, **super-diffusive replay-like dynamics** emerge when the adaptation strength is set near the travelling boundary (see Fig. 2c). Network noise sometimes drives the adaptation effect below the boundary, causing the activity bump to travel only short distances (local movements). At other times, noise drives the adaptation above the boundary, resulting in the bump travelling large distances (jumpy movements). Thus, the dynamics are a mix of local movements and large jumps, rather than being dominated by extreme tails (see Fig. 3a,f&i).

3. “The strong damping suppresses all the activities except for the extreme tail of the distribution...”

This scenario does not occur under the adaptation strength regime used in our model. For this to happen, the adaptation would need to be exceptionally strong, effectively eliminating the activity bump at location A and allowing it to grow only at location B (far from A). This would leave only the tail of the distribution. However, such a case was not considered in our model.

We hope this intuitive explanation clarifies the concern. For further details, we refer the reviewer to Eqs. 3 and 4, where we theoretically demonstrate that the distances travelled follow a power-law distribution, not one where only the tail remains.

Next, we address the questions regarding the “preferred direction” of the bump::

1. “...meaning we could observe two bumps travel to opposite directions simultaneously...”

This will not occur in the model due to global inhibition and the symmetric synaptic connections, which decay with the distance between the firing fields of two place cells with no directional preference. At any point during the simulation, the network will only support a single activity bump.

2. “...In reality, replay does have preferred directions, but I cannot see why it should happen in the model based on equations 1-2...”

In the current model, simulated replay does not exhibit a preferred travel direction because of its design. Specifically, the synaptic connections are translationally invariant and symmetric.

- **Linear Track:** On a linear track, the activity bump can travel in either direction with equal probability (50% chance of travelling left or right).
- **Open Field:** In an open field, the activity bump can propagate in any direction with equal probability.

We recognise that in real scenarios on linear tracks, place fields and synaptic connections can become asymmetric due to experience-dependent processes such as LTP-like plasticity (e.g., Mehta et al., 2000). This can be incorporated into the model by modifying the symmetric synaptic connections to asymmetric ones. Under these conditions, the activity bump would exhibit a preferred travel direction. Importantly, this modification would not alter the overall modelling results, as the firing rate adaptation can still apply, with stronger adaptation leading to more diffusive replay-like dynamics, even with a preferred direction. The situation becomes more complex in 2D environments, where animals do not travel linearly (e.g., left to right). For instance, in a 2D model with synaptic connections biased towards a home location or a fixed goal location, the replay trajectories could be simulated to head toward the goal or home location (as in Pfeiffer and Foster, 2013). In this case, adding adaptation would still generate replay sequences exhibiting various diffusive dynamics.

We have discussed this scenario of asymmetric synaptic connections in Line 380-387.

3. “...When authors run numerical simulations at the middle of the track, does the probability bump propagate in one direction rather than two...”

First, we would like to clarify that the bump is an **activity bump**, not a probability bump. Second, regardless of the starting point of the simulation, the bump propagates in both directions with equal probability under symmetric synaptic connections, but each time point will only have one bump, not two. The initial direction is determined by network noise and, once established, the bump continues propagating in that direction.

5. The authors use a cluster less method to decode neural activity from combined spikes from areas CA1, CA2, and CA3. These 3 brain areas are known to behave differently across theta phases (CA1 versus CA3) and during sharp-wave ripples (CA2). Notably, CA2 can show stationarity during ripples. Additionally, decoded stationarity was shown for whole detected events during development in CA1 (Farooq and Dragoi, 2019 and Muessig et al., 2019, both should be referred to). An additional confound could be brought up by pyramidal spike-like activity that may not pass the criterion of a well clustered pyramidal neuronal activity, specifically spikes from putative inhibitory neurons (that escaped the waveshape criterion) and EMG activity (or other non-neuronal activity). These activities could increase diffusivity as well as stationarity in an uncontrollable fashion. The authors should cross-validate their biological claims using clustered CA1 putative pyramidal neurons from this or other datasets available to them.

Thank you for pointing this out. The empirical data we used primarily consists of recordings from CA1 and CA3 tetrodes, with only a few tetrodes from CA2 (from three animals). The animals were adult rats at the time of recording. Interneurons were excluded based on waveform characteristics, specifically those with widths less than 0.3 ms (Fox and Ranck, 1975).

Regarding EMG activity, it occurs across the environment regardless of the animal's location and can be treated as a noise term independent of the decoded spatial location. The clusterless method effectively handles such noise, as demonstrated in previous studies (e.g., Deng et al., 2015; Denovellis et al., 2021). This method has been successfully applied to identify both theta and replay sequences in the hippocampus (Chen et al., 2012b; Kloosterman et al., 2014; Deng et al., 2016; Kay et al., 2020) and is particularly useful for real-time decoding.

Compared to clustered spike methods, which discard spike events that cannot be uniquely assigned to individual neurons, the clusterless method retains more information, enhancing the decoding quality. Additionally, spike sorting is not essential for recovering underlying neural population dynamics (Trautmann et al., 2019) and often involves subjective inclusion decisions that vary across experimenters.

Regarding cross-validation with clustered CA1 cells:

Unfortunately, the dataset we obtained contains only few well-sorted cells. The best session included only 22 sorted CA1 cells, resulting in poor decoding outcomes. However, we re-ran the analysis using clusterless methods applied only to CA1 tetrodes. The decoded theta sequences were visually comparable to those obtained using all tetrodes, and the replay sequences also showed similar patterns (Panel a: example from one animal). Across all sessions, replay diffusivity was consistent with results obtained using CA1–CA3 tetrodes (Panel b). Additionally, the correlation between replay diffusivity and theta sweep length was preserved (Panel c), although its statistical significance diminished, likely due to the reduced number of tetrodes used for decoding.

While CA1 and CA3 neurons may exhibit differences in theta phase behaviours during running, we visually inspected the decoded theta sequences and found them comparable to those obtained using all tetrodes. Furthermore, CA1 and CA3 spiking is known to be tightly coordinated within and across hemispheres during SWR replay (Carr et al., 2012). Multiple prior studies (e.g., Diba and Buzsáki, 2007; Karlsson and Frank, 2009) have combined recordings across hippocampal subfields and reported results consistent with those obtained using only CA1. Therefore, we used all available tetrodes for both theta and replay decoding.

Finally, we acknowledge that no single decoding algorithm is universally “best”; each has its strengths and limitations depending on the context. As the number of recorded neurons increases, the performance differences among decoding algorithms diminish.

Regarding cross-validation of clustered data on other datasets:

We have tested the decoder on clustered CA1 data (Carey et al., 2019) and MEC grid cell data (Gardner et al., 2022) in another paper (Ji&Chu et. al., 2025, Curr. Biol.), and it performed well in decoding theta sequences. Specifically, it successfully identified forward-backward sweeps in the Carey dataset and left-right sweeps in the Gardner dataset.

We hope these responses address the reviewer’s concerns.

6. The decoding error during animal exploration of the maze has a median of 9 cm for 1-2 m-long segments being decoded. This error is very large and will most likely impact the reliability and interpretation of the posterior probabilities. The authors compared this error with errors reported in 2 previous studies. However, one of those studies used a 10 m-long track and large bins and the other used immature place cells during development, both of which increase the decoding error. The reported decoding error in 1-3 m-long track of adult rats is usually less than half of that reported here. The large error here could be contributed by the clusterless decoding where ‘noise’ combined with genuine spiking activity of pyramidal (CA1) neurons. The authors should also compute the decoding error obtained from CA1 clustered putative pyramidal neurons from this or other available datasets (see comment #6).

Thank you for raising this point. While the decoding error appears relatively large (median of 9 cm with a 95% CI of 8.2–9.8 cm), we believe this is an accurate reflection of the underlying data. Through visual inspection of multiple running periods across different animals, we observed that the decoded positions closely track the animals' actual positions (see Fig. S4 for examples of online decoding from various trials and animals).

The larger decoding error is likely attributable to theta sweeps within the decoded position. During a theta cycle, the look-ahead or look-back distances can extend up to 20 cm. This characteristic inherently contributes to larger decoding errors, as opposed to smaller errors that would imply a lack of theta sweeps, where the decoded position would unrealistically align precisely with the animal's position throughout.

We also computed the decoding error using only CA1 tetrodes, though still applying the clusterless method (as the dataset did not include well-sorted cells). In this case, the median decoding error was 9.4 cm with a 95% CI of 8.4–10.2 cm. This result is expected, as using only CA1 tetrodes provides less information compared to including all tetrodes (with approximately half in CA1 and the remainder in CA3, and only a few in CA2).

Minor concerns

1. If I understand the model (equation 1) correctly, cells are arranged based on their place field in the environment. In that way, we have a one-to-one mapping between space and cells. Then the assumption that

the recurrent connection strength diminishes with distance can be very interesting. In other words, we can say that cells with stronger functional connectivity are more likely to represent adjacent locations. Does it mean that by this arrangement the model indicates a very strong pre-configured spatial representation (preplay)?

Thank you for raising this interesting point. The observed stronger connectivity does not necessarily indicate preplay, as suggested by a study from the Foster lab (Silva et al., 2015). Instead, it could be a result of LTP-like synaptic plasticity driven by prior experience. For example, sequential activities during immobile periods can be disrupted by administering an NMDA receptor antagonist during the preceding experience, which supports this plasticity-based interpretation.

In the model, the assumption of stronger functional connectivity among cells coding for adjacent locations can be attributed to LTP-like plasticity. Initially, cells may not exhibit recurrent connectivity, but over time, theta sequences develop during learning (Feng et al., 2015). This process gradually strengthens connections between neurons have overlapping firing fields that participate in successive theta sweeps.

We acknowledge the importance of investigating this learning process in greater detail and plan to explore it in future work.

2. If there is a one-to-one mapping, I would suggest changing the xlabel in Fig. 2b top panel into position. We care more about activity strength across spatial bins. Also, what are those off-diagonal high values in Fig. 2b bottom panel? Are they related to some boundary conditions?

We have updated the x-axis label in Fig. 2b (top panel) for clarity.

The high off-diagonal values are a result of the periodic boundary conditions implemented in the model, which simplify the simulation process. Importantly, removing the periodic boundary effect would not alter the modeling results, as the underlying dynamics remain consistent.

3. I don't understand this statement in section 2.2.2 "This type of eigenvalue rescaling amplifies transition probabilities associated with distant states...". I assume the spectral analysis decomposes the time series into spectral components, and, in the absence of nonlinearity, Fourier modes are mutually independent, with their temporal dynamics illustrated by eigenvalues. Large (>1) eigenvalues indicate their strength will grow over time while small (<1) eigenvalues indicate they will be reduced. In that case, at large offset, some modes (large or small) will grow, and some will decay, those modes will change in their power/amplitude rather than become other modes. So, what does "amplifies transition probabilities associated with distant states" mean?

Thank you for pointing this out. We offer an intuitive metaphor followed by a mathematical explanation:

Imagine eigenvectors with longer periods as being analogous to large cities in the US—big, but farther apart from one another—whereas eigenvectors with shorter periods are like small towns—smaller, but located closer together. Travelling between large cities results in "jumpy" transitions, while travelling between small towns is

smoother and less jumpy. Rescaling large eigenvalues can be thought of as preferring to travel between big cities rather than small towns, resulting in a trajectory that is more jumpy.

Mathematical Explanation:

Consider a replay sequence propagating at a constant speed under a transition matrix O governed by the dynamics: $\tau \dot{U} = O \cdot U$. Decomposing O yields a set of eigenvalues and eigenvectors. Small eigenvalues correspond to eigenvectors with short periods, while large eigenvalues correspond to eigenvectors with longer periods.

If we scale up the large eigenvalues and scale down the small ones, then reconstruct the transition matrix and let the dynamics evolve, the resulting sequence becomes more diffusive. This occurs because increasing the weight of eigenvectors with large periods enhances their influence, and these large periods correspond to larger spatial displacements. Essentially, when the sequence "jumps," it moves further away from its current position due to the dominance of large-period eigenvectors.

For a more detailed theoretical analysis, we refer the reviewer to McNamee et al. (2021).

4. The cluster less decoding method (Denovellis et al., 2021) assumes three states (stationary, continuous, and fragmented), and the initial transition matrix is heavily concentrated on the diagonal. I don't know how this strong tendency of keeping the current state (inertia) will impact the result as the concepts of stationary, diffusive, or super-diffusive seem to overlap with these states in the cluster less decoding method. How about using a memoryless Bayesian decoding method with all the spiking events before clustering?

This issue has been thoroughly investigated in Denovellis et al., 2021. First, it is important to note that the transition matrix is updated every 2 ms (the step size in decoding), with a probability of staying in a given dynamic state set at 0.98. Over a 100 ms window—the typical duration of a sharp-wave ripple (SWR)—this translates to a 64% chance of switching to another dynamic state (calculated as $1 - 0.98^{50}$).

Second, the state-space decoder differs from standard decoding methods by integrating a data model (the same as the standard Bayesian decoder) with a movement dynamic model. Unlike memoryless Bayesian decoders, which use larger time bins (e.g., 20 ms), the state-space model allows for smaller time bins (2 ms) by incorporating prior knowledge about the movement dynamics. This enables the detection of changes on finer time scales and accounts for periods without spiking activity (Deng et al., 2015).

Finally, calculating replay diffusivity (see Fig. 4d inset) requires detecting changes at small time scales, such as 2 ms. Consequently, the memoryless Bayesian decoding method would not be suitable for this analysis in this study.

5. The discussion of the relationship between theta sequences and replay needs to be re-evaluated. Firstly, Drieu et al., 2018 showed a correlation between lack of theta sequences in a non-natural condition and a degraded replay, not a definitive causal relationship between them. Secondly, Farooq and Dragoi, 2019 also showed a correlation between the emergence of theta sequences and replay during development (should be referred to together with Drieu et al. 2018 and Muessig et al., 2019). Thirdly, Dragoi and Buzsaki, 2006 showed the temporally-compressed organization of place cells sequences during theta oscillations (theta sequences) and discussed their relationship to replay (should be cited). Figure 1a in the current study seems to be adapted after the Dragoi and Buzsaki study and this should also be acknowledged.

Thank you for the suggestions. We have now added these references in the text accordingly and updated the discussion about the relationship between theta sequences and replays.

6. I think in Fig. 5c maybe a 2d joint distribution showing density/counts can better illustrate the relation between theta sequence length and replay step size. Original plots can be still included as marginal distributions.

Thank you for pointing this out. The purpose of this plot (now Fig. 4d) was to present an example of the distributions of theta sequence lengths and replay step sizes from a typical recording session. Since there is no direct one-to-one correspondence between theta sequences and replay step sizes (e.g., one theta sequence followed by one replay or vice versa does not occur), we cannot explicitly illustrate their relationship in this context. For an analysis of the relationship between these two features, please refer to Fig. 4e in the revised paper.

7. Fig. 5e seems to be bad examples, the differences between animals are not obvious. Why in Fig. 5d rat 3 has an average sweep length >50 cm while in Fig. 5e the mean value is clearly <50cm?

Thank you for pointing this out. When we generated Fig. 5e, we excluded look-ahead and look-back distances greater than 50 cm. This exclusion resulted in the mean value of theta sequence lengths appearing smaller than 50 cm in Fig. 5e, even though the true average exceeds 50 cm, as shown in Fig. 5d. To avoid any potential confusion, we have removed the misleading example (now Fig. 4 in the revised paper), as Fig. 5d (now Fig. 4d) sufficiently conveys the intended message.

8. I don't agree with the statement at the last line of page 18 "awake replay might directly reflect the animal's movements." Even though waking rest replay has higher ratio than sleep replay, only less than 15% are replay of current trajectories by the traditional measure (with a 5% chance level). Especially in Pfeiffer and Foster, 2013, less than 1% of replays are goal-directed.

Good to know about these stats! We have updated it to (Line 347-351):

'...this observation suggests that awake replay may share more similarities with the animal's movement dynamics than sleep replay does...'

9. There is a typo in 4.3, " τ_u is the neuronal time constant, and τ_v is the adaptation time constant which is much larger than τ_v ". Last one should be τ_u .

Updated.

10. Adding line numbers can make the reviewing process easier.

Updated.

Dear Editors and Referees,

We thank the reviewers for their time and constructive feedback in this second round of review of our manuscript, "*Dynamical Modulation of Hippocampal Replay Sequences through Firing Rate Adaptation*" (NCOMMS-24-50968A-Z). Below, we respond to the comments point by point, with reviewers' remarks in green and our replies in plain text.

Yours sincerely,

Dr. Zilong Ji and Prof. Neil Burgess
UCL Institute of Cognitive Neuroscience & UCL Queen Square Institute of Neurology,
University College London, London, UK

Reviewer #1 (Remarks to the Author):

Thanks to the authors for their extensive revisions and comments. I have a few remaining queries/comments which I hope will be useful.

The authors performed a fresh set of analyses in order to provide evidence for adaptation mechanisms in awake behavior animals. This seems to me to be a significant contribution in this manuscript and I would suggest adding this figure and potentially even referencing it in the abstract.

However, the authors say "In addition, we have highlighted a very recent paper from the Foster lab (Mallory et al, Science, 2025) showed that firing rate adaptation is a key factor controlling hippocampal replay. This is a direct support of our firing rate adaptation model which we have added to the references. " As far as I can see, Mallory et al just model the predominance of forward replay post-stop as a neural adaptation/fatigue mechanism in a CAN model. They don't provide empirical evidence that that is how the brain is generating such replay dynamics which I interpret is what "direct support" means. Its the similar issue that was highlighted in the previous round of reviews from my pov. There are many CAN-type models invoking neural adaptation in neural replay. Here is another study along the same lines from the same authors: [https://www.cell.com/cell-reports/fulltext/S2211-1247\(25\)00246-3](https://www.cell.com/cell-reports/fulltext/S2211-1247(25)00246-3)

Thank you for this comment. I recently discussed this with John Widloski when he was in London, and we reached a general agreement on the point. In their replay results, the key evidence for adaptation is shown in Fig. 2D and 2K — namely, forward replay occurs before reverse replay. The idea is that neurons along the

animal's recent path have already been active and are therefore adapted, whereas neurons ahead of the path have not yet adapted. This leads the replay sequence to run forward first, then backward. This is what I meant by "direct evidence," although it is not direct in the sense of experimentally manipulating adaptation and observing its effect on replay.

We are familiar with the paper the referee mentioned ([https://www.cell.com/cell-reports/fulltext/S2211-1247\(25\)00246-3](https://www.cell.com/cell-reports/fulltext/S2211-1247(25)00246-3)), and we have also published related work here: [https://www.cell.com/current-biology/fulltext/S0960-9822\(24\)01174-6?uuid=uuid%3A47a8abd3-bd6d-4c23-a93c-e7f26e5af58e](https://www.cell.com/current-biology/fulltext/S0960-9822(24)01174-6?uuid=uuid%3A47a8abd3-bd6d-4c23-a93c-e7f26e5af58e), which focused more on left-right theta sweeps in the entorhinal cortex.

"Our focus is on illustrating how diverse replay dynamics observed in the hippocampus can be reconciled through a simple and effective neural mechanism of firing rate adaptation, rather than on whether controlling adaptation strength can directly induce a replay regime switch. These are two distinct questions."

I don't understand. It seems the model is that you get diverse replay dynamics if the adaptation strength is changed. Changing the adaptation strength gives you diversity so I don't see how "diverse replay dynamics" can be divorced from changing the adaptation strength. Maybe I'm misinterpreting the word "diverse" but I understand it to mean diffusive vs superdiffusive for example.

Thank you for the comment. What we meant is that different adaptation strengths could lead to diverse replay dynamics. However, we are cautious about making claims regarding what controls the change in adaptation strength that could cause replay statistics to switch from diffusive to super-diffusive, or vice versa — as we currently have no direct evidence for this.

In the McNamee et al. paper (which we have cited and credited as an elegant and influential model), such a replay switch is achieved through systematic modulation along the DV axis in MEC (implemented mathematically via eigenvalue rescaling). In our own framework, we face a related open question — namely, what biological mechanisms might control changes in adaptation strength. While we do not yet have a definitive answer, we see this as an interesting area for future work and would be glad to explore possible links with the DV-axis modulation described in McNamee et al.

In the abstract, it is still claimed that this model predicts that "replay diffusivity varies within an animal across behavioural states" As discussed in the previous round of reviews, this is not a (novel) prediction, since it was already normatively predicted and evidenced in data in McNamee et al 2021 and also forms a key element of

Krause & Drugowitsch 2022. The contribution here is to repeat such analyses on a per-animal basis rather than at the group-level. Furthermore, I don't see how it is a prediction at all in this ms. What is shown here is that it is possible for adaptation mechanisms to generate both diffusive and superdiffusive sequencing but doesn't give any reason why these might be associated with e.g. wake vs sleep. Why not superdiffusive during sleep? This seems beyond the scope of this study which focuses more on circuit mechanisms.

Thank you for pointing this out. We have now revised the text to read: "...our framework is consistent with previous work showing that replay diffusivity can vary within an animal across behavioural states..." We agree that this is not a novel prediction, and appreciate the clarification. We also agree that the contribution here is to demonstrate this effect at the per-animal level rather than at the group level (as reflected in line 284 of the main text).

Regarding point 6 - the distinction between Levy flights and Levy walks. Thanks for clarifying, it makes sense to me. Interesting comment from Brad Pfeiffer, I think the control analyses in Pfeiffer & Foster should account for such concerns but no matter. In order to sharpen the predictive distinction between models, I suggest explicitly citing McNamee et al as predicting Levy flights in the superdiffusive regime (based on cortical input) in contrast to the Levy walks of the model considered here.

Thanks for pointing this out. Following your suggestion, we have now explicitly cited McNamee et al. in Line 892:

"...This power-law distribution corresponds to Lévy walks, where the activity bump traverses the intervening positions before stabilizing at the final position, rather than Lévy flights, where the activity bump 'jumps' directly to the final position without traversing the intervening positions (see McNamee et al. for more details)...."

Reviewer #2 (Remarks to the Author):

The authors have comprehensively and successfully addressed all concerns that I raised in the first round.

We thank the reviewer for their positive feedback and are pleased that our revisions have addressed the concerns from the first round. We also appreciate their assessment of the code availability.

Reviewer #2 (Remarks on code availability):

There are instructions for running the code and the code seems to be complete. I

have not tried running the code.

Reviewer #3 (Remarks to the Author):

In the revised manuscript, the authors have addressed most of our previous concerns. By incorporating patch-clamp data analysis and discussing potential neural mechanisms underlying hippocampal adaptation, the manuscript has improved in quality and now has stronger biological support. I do not have concerns requiring further analysis, but I have two minor questions that need clarification. I believe there may have been some misunderstanding, possibly due to a lack of clarity in my previous comments, so I will attempt to clarify them here.

1. Related to previous major concern 2: The authors support their model with the following arguments:

1.1 Their model predicts that adaptation strength is positively correlated with theta sweep extent during running.

1.2 Their model predicts that adaptation strength is positively correlated with replay diffusivity during waking rest.

1.3 Experimental data show that theta sweep extent is positively correlated with replay diffusivity.

For this argument to be valid, do we need to assume that the hippocampal circuit exhibits similar adaptation strength between running and waking rest states? The authors demonstrated that adaptation strength can differ between rest and sleep states. If adaptation strength also differs between waking rest and running, could that undermine their argument? My intuition is that a common factor invariant across both waking rest and running states (such as recording quality or place map length) may better explain the observed correlation between theta sequences and replay.

Thanks for pointing this out. First, in response to your question: we need to assume similar adaptation strength — that is, a common factor invariant across both running and waking rest states for the same animal. If adaptation strength differed between these states, it would indeed undermine the argument in the model, as the referee noted. However, because running and waking rest are interleaved within the same recording session, we would not expect adaptation strength to change so frequently and rapidly. That said, it is possible that firing rate adaptation could differ between wake and sleep states, and we agree that more empirical evidence is needed to address this.

Regarding the comment on “recording quality or place map length as a better/easier explanation of the correlation between theta sweeps and replay,” we performed two

control analyses in the manuscript (Lines 219–234, and Fig.S7, S8 & S9). First, we tested whether place field size was a confounding factor for the correlation and found it was not. Second, we tested whether decoding accuracy was a confounding factor and also found it was not. We believe the first analysis relates to the “place map length” mentioned by the referee, while the second relates to recording quality.

2. Related to previous major concern 4: The authors have addressed most of my questions, and I appreciate their patience. My remaining question pertains to Eqn 9 (global inhibition), particularly with the large k value used in their study. To me, this term looks like a normalization term that makes sure overall firing rates remain stable over time. While this may be a commonly used technique in CAN models, I would like further biological explanation on this. My question consists of two parts:

2.1 With a small k value, the firing rate of neurons would be more directly dependent on their own presynaptic inputs, which seems more biologically plausible to me. In this case, would the overall activity level of the circuit (or the amplitude of the traveling bump) experience rapid growth or decay over time?

2.2 If so, then a relatively large k value is crucial to reconcile the model's predictions with experimental data. In that case, what is the biological interpretation of this strong global inhibition (or alternatively, global excitation, if the overall activity tends to decay over time)? This equation implies that at each time step, a neuron's firing rate depends not only on its own presynaptic inputs but also on the presynaptic inputs of all neurons in the network. How could neurons instantaneously know presynaptic inputs of other neurons in the network? I would expect neuronal homeostasis to occur on a relatively longer timescale, making this instantaneous adjustment seem biologically implausible. A discussion or explanation of this mechanism would be valuable.

Thanks for pointing this out.

Re Q2.1: From our theoretical analysis of the network dynamics, to ensure that the CAN holds bump-like activity, k must satisfy the condition:

$$k < \rho J_0^2 / (8\sqrt{2\pi}a)$$

where ρ is the density of neurons (number of neurons arranged per unit area on the x and y coordinates), J_0 is the baseline recurrent connection strength, and a is the place field size. If k is larger than this value (i.e., global inhibition is too strong), the bump-like activity cannot be maintained and the network will be silent. In other words, k cannot be too large.

The reason the referee thought k was relatively large is because we set ρ as 20 in table 1, which makes k as 20. If we set ρ as 1, then k would also be set to 1 (to keep them balanced).

If we choose a small value of k , the overall activity level will increase, but not substantially, because of global normalization — that is, the activity will be normalized to a lower level when the overall activity increases.

Re Q2.2: We hope the reply to Q2.1 has partially answered the question here, that is, k cannot be relatively large, which needs to be bounded by $\rho J_0^2 / (8\sqrt{2\pi}a)$.

Re: "This equation implies that at each time step, a neuron's firing rate depends not only on its own presynaptic inputs but also on the presynaptic inputs of all neurons in the network. How could neurons instantaneously know presynaptic inputs of other neurons in the network?"

The referee is correct. We also recognise that this is a strong assumption in our model and in many previous CAN models. Without such global inhibition, the activity cannot form a single bump; instead, there could be multiple bumps in the network (as in some models of grid cells). However, to model place cell activity and/or head direction cell activity, global inhibition (or a similar mechanism) is required.

An alternative is to explicitly model interneurons, where each excitatory neuron projects randomly to several interneurons, and each interneuron projects randomly to several excitatory neurons. The average effect of such connectivity would approximate global inhibition. Simply put, each excitatory neuron can "see" all other excitatory neurons through inhibitory neurons, rather than directly through excitatory connections.

Finally, our model is a rate-based model simplified from a spiking model via mean-field approximation, and therefore omits certain details present in spiking models. This simplification allows us to make the analysis tractable and to gain a clearer analytical understanding of the network dynamics. In this context, the global inhibition term in the rate-based model can be interpreted as the averaged effect of multiple underlying inhibitory processes, which ultimately produce a similar form of global inhibition.

Dear Editors and Referees,

We thank the reviewers for their time and constructive feedback in this final round of review of our manuscript, "*Dynamical Modulation of Hippocampal Replay Sequences through Firing Rate Adaptation*". Below, we respond to the comments point by point, with reviewers' remarks in green and our replies in plain text.

Yours sincerely,

Dr. Zilong Ji
UCL Institute of Cognitive Neuroscience & UCL Queen Square Institute of
Neurology,
University College London, London, UK

Reviewer #1 (Remarks to the Author):

Thanks for your work on this revision and clear descriptions of the updates.

Regarding the Mallory et al work, I also spoke with John on his recent European tour :)
some take-aways:

I remain steadfast in my point that there is no "direct evidence" for the neural adaptation model in the context of the Mallory et al data. As you say "...it is not direct in the sense of experimentally manipulating adaptation and observing its effect on replay." I'm quite confident that "direct evidence", in general parlance, does indeed correspond to the experimental manipulation of the cause in order to test for the predicted effect. One could say that your model (which is essentially the same concept of John's in modeling retrospective replay suppression in the Mallory paper) is consistent with this data. However, it is also consistent with replay suppression via MEC input to HC as modeled here (Figure 2) albeit in a different scenario with an explicit cognitive function (<https://www.mdpi.com/1099-4300/24/12/1791>).

It is also notable that a major part of the Mallory work is to demonstrate the manipulation of hippocampal replay as a function of MEC input, thus I think it worth emphasizing how the Mallory work points to some form of cooperative interaction between HC-endogenous neural adaptation and MEC input as two distinct mechanisms controlling the timecourse of replay (i.e. adaptation suppresses retrospective replay initially then MEC input subsequently enhances it) e.g. towards the end of the second paragraph in your Discussion (fyi, an attempt at such a point was made here in Figure 2 <https://www.sciencedirect.com/science/article/pii/S0959438824000175> but not as eloquently elaborated as you have it in your Discussion points).

Thank you for pointing this out. We have revised the Discussion accordingly. We now refer to the Mallory paper only in the following context: “FRA in biological systems is hypothesised to serve roles extending beyond the initiation of hippocampal replay...” We hope this phrasing provides a more accurate representation than before.

In the discussion, you have the line "This idea is supported by empirical data showing that MEC input controls the temporal organization of hippocampal activity [Schlesiger et al., 2015, Yamamoto and Tonegawa, 2017], but possibly not by Ormond and McNaughton [2015]."

Can you elaborate re the Ormond/McNaughton point? The point is that the Ormond/McNaughton data is inconsistent with the MEC theory? I always thought it quite consistent in the sense that the grid code in MEC constitutes a spectral decomposition. In the McNamee et al paper, there is some modeling where if you suppress high-frequency components (via tau modulation) then the place receptive fields in HC expand/spread.

Thank you for pointing this out. I see this point now and have deleted “but possibly not by Ormond and McNaughton [2015].” in the Discussion.

Really comprehensive and interesting work, thank you.

Thank you for the thorough and engaging discussion — this is what peer review should be. Much appreciated.

Reviewer #3 (Remarks to the Author):

In this improved revised manuscript, the authors have comprehensively addressed all my questions and I have no additional queries. I recommend this interesting study for publication.

We thank the reviewer for their positive feedback and are pleased that our revisions have addressed the concerns.